# Post-extinction recovery of the Phanerozoic oceans and biodiversity hotspots

Pedro Cermeño[1,9 ✉], Carmen García-Comas[1,9 ✉], Alexandre Pohl[2,3], Simon Williams[4,5], Michael J. Benton[6], Chhaya Chaudhary[7], Guillaume Le Gland[1], R. Dietmar Müller[5], Andy Ridgwell[2] & Sergio M. Vallina[8]

The fossil record of marine invertebrates has long fuelled the debate as to whether or not there are limits to global diversity in the sea[1–5]. Ecological theory states that, as diversity grows and ecological niches are filled, the strengthening of biological interactions imposes limits on diversity[6,7]. However, the extent to which biological interactions have constrained the growth of diversity over evolutionary time remains an open question[1–5,8–11]. Here we present a regional diversification model that reproduces the main Phanerozoic eon trends in the global diversity of marine invertebrates after imposing mass extinctions. We find that the dynamics of global diversity are best described by a diversification model that operates widely within the exponential growth regime of a logistic function. A spatially resolved analysis of the ratio of diversity to carrying capacity reveals that less than 2% of the global flooded continental area throughout the Phanerozoic exhibits diversity levels approaching ecological saturation. We attribute the overall increase in global diversity during the Late Mesozoic and Cenozoic eras to the development of diversity hotspots under prolonged conditions of Earth system stability and maximum continental fragmentation. We call this the 'diversity hotspots hypothesis', which we propose as a non-mutually exclusive alternative to the hypothesis that the Mesozoic marine revolution led this macroevolutionary trend[12,13].

The question of whether or not there is an equilibrium diversity that the biota, or portions of the biota, cannot exceed has led to decades of debate between those who think that there is a limit to the global diversity that the Earth can carry[2,3,10] (that is, a carrying capacity) and those who think that the biosphere is so far from the equilibrium diversity (that is, its carrying capacity) that we can ignore the existence of any limit[4,5,11]. This question has traditionally been addressed by examining the shape of global fossil diversity curves[3,14]. For example, the Palaeozoic era plateau in marine invertebrate diversity is generally taken as strong evidence for the existence of ecological limits to further diversification[3,15]. However, as diversity varies considerably among geographical regions, and each geographical region has its own geological and environmental history, addressing this question requires simultaneously reconstructing the dynamics of regional diversity in both space and time[16,17]. If diversity dynamics were governed by diversity-dependent feedbacks on speciation and extinction rates, then regional diversity should remain stable regardless of time once the carrying capacity has been reached (that is, the logistic model). By contrast, if evolutionary rates were independent of standing diversities, then we should observe positive relationships between evolutionary time-within-regions (or time-for-speciation) and diversity; the older the habitat, the longer the lineages have had to diversify and fill empty

niches or explore new ones (that is, the exponential model). Determining which diversification model best describes the dynamics of regional diversity is key to understanding the mechanisms that underlie biogeographical patterns and macroevolutionary trends. However, the fossil record is biased by uneven geographical and stratigraphic sampling efforts[17,18] and variation in the rock record available for sampling[19], hindering our ability to investigate the effect of geographical variability in evolutionary time and diversification rate.

To overcome this limitation, we coupled two alternative models of diversification—logistic and exponential—to a global model of palaeogeography and plate-motion that constrains evolutionary time-within-regions. In both diversification models, the net diversification rate varies within a fixed range of values as a function of seawater temperature and food supply, which are reconstructed using a spatially explicit palaeo–Earth system model (Methods and Extended Data Table 1). In the logistic model, the spatially resolved effective carrying capacities ($K_{eff}$) are allowed to vary within a fixed range of values (from $K_{min}$ to $K_{max}$) as a positive linear function of food availability in each ocean region and time. We set relatively low $K_{min}$ and $K_{max}$ values to enforce diversity saturation, hereafter referred to as the 'saturated' logistic model. Mass extinctions are imposed by imputing negative diversification rates to regional communities and assuming

[1]Institut de Ciències del Mar, Consejo Superior de Investigaciones Científicas, Barcelona, Spain. [2]Department of Earth and Planetary Sciences, University of California, Riverside, Riverside, CA, USA. [3]Biogéosciences, UMR 6282, UBFC/CNRS, Université Bourgogne Franche-Comté, Dijon, France. [4]State Key Laboratory of Continental Dynamics, Department of Geology, Northwest University, Xi'an, China. [5]EarthByte Group, School of Geosciences, University of Sydney, Sydney, New South Wales, Australia. [6]School of Earth Sciences, University of Bristol, Bristol, UK. [7]Alfred Wegener Institute, Helmholtz Centre for Polar and Marine Research, Bremerhaven, Germany. [8]Instituto Español de Oceanografía, Consejo Superior de Investigaciones Científicas, Gijón, Spain. [9]These authors contributed equally: Pedro Cermeño, Carmen García-Comas. ✉e-mail: pedrocermeno@icm.csic.es; cgcomas@icm.csic.es

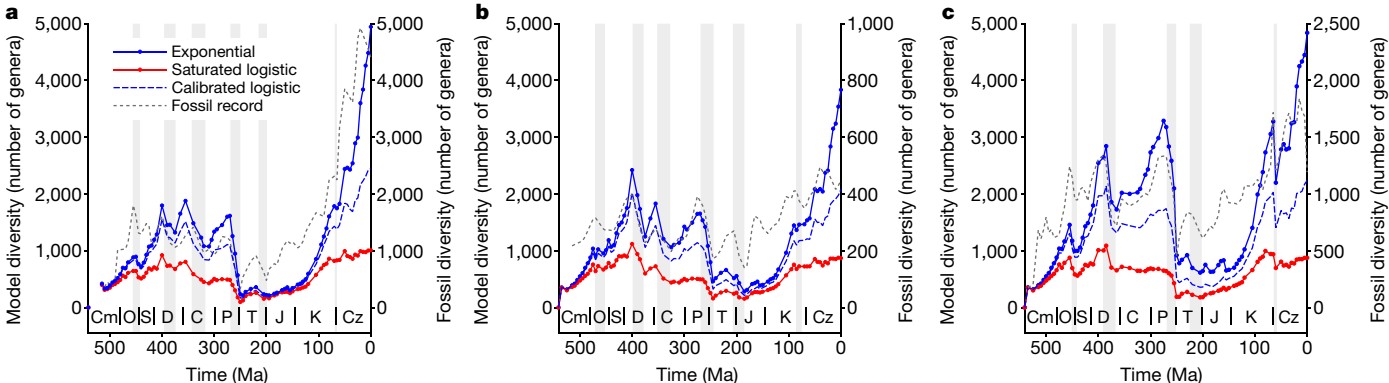

**Fig. 1 | Global diversity dynamics during the Phanerozoic. a–c**, Global diversity dynamics reconstructed from the saturated logistic model (red), the exponential model (blue) and the calibrated logistic model (blue dashed line) after imposing the pattern of mass extinctions extracted from the fossil diversity curve of ref. [20] (**a**), ref. [21] (**b**) and ref. [22]. (**c**). In each panel, the corresponding fossil diversity curve is superimposed (grey). The shaded areas represent mass extinctions. C, Carboniferous; Cm, Cambrian; Cz, Cenozoic; D, Devonian; J, Jurassic; K, Cretaceous; O, Ordovician; P, Permain; S, Silurian; T, Triassic.

non-selective extinction. The percentage of diversity loss as well as the starting time and duration of mass extinctions were extracted from three fossil diversity curves of reference[20–22].

## Reconstructing global diversity dynamics

Each of the two diversification models tested here produces a total of 82 spatially explicit reconstructions of diversity spanning from the Cambrian period (541 million years ago (Ma)) to the present (Supplementary Videos 1 and 2). On each of the 82 diversity distribution maps, we traced hundreds of line transects from diversity peaks to their nearest diversity troughs and integrated the total diversity in each transect by assuming a decay function in taxonomic similarity with geographical distance (Methods and Extended Data Fig. 1). For each of the 82 time intervals, all integrated diversities along transects were then reintegrated stepwise, from the transect with the greatest diversity to the transect with the lowest one, assuming the same distance-decay function applied to individual transects. The resulting global diversity estimates were plotted against the midpoint value of the corresponding time interval to generate a synthetic global diversity curve. Both the saturated logistic model and the exponential model produced relatively similar global diversity dynamics (Fig. 1). This is to be expected as the global diversity curves produced by both models were equally influenced by mass extinctions and long-term variations in the degree of continental fragmentation (Extended Data Fig. 2). However, although both models show similar diversity dynamics, the amplitudes of global diversity variations differ markedly between models (Fig. 1). The exponential model gives rise to conspicuous increases in global diversity from the Cambrian to Late Ordovician period, the Silurian period to Early Devonian period, the Carboniferous period (Early to Late Pennsylvanian subperiod), the Early to Late Cretaceous period and the Palaeocene epoch to the present. The Permian–Triassic period mass-extinction event lowered global diversity to Early Palaeozoic levels, but later diversification led Late Cretaceous and Neogene period faunas to exceed the mid-Palaeozoic global diversity peak. These trends emerge consistently regardless of which mass extinction pattern was imposed[20–22] (Fig. 1). The logistic model also reproduces the initial increase in diversity, from the Cambrian to Late Ordovician and from the Silurian to Early Devonian (Fig. 1). However, in contrast to the exponential model, in the saturated logistic model, this initial upwards trend is followed by a convex diversity pattern that is interrupted by a modest increase during the Cretaceous, which rarely exceeds the mid-Palaeozoic global diversity peak in our set of simulations. A sensitivity analysis revealed that these global dynamics are barely affected by (1) a sea level rise/fall of 100 m relative to the sea level of the original palaeo–digital elevation model and (2) halving

or doubling the mean ocean phosphate concentration, to decrease/increase food availability (Extended Data Fig. 3).

## The calibrated logistic model

All other things being equal, the higher the $K_{min}$ and $K_{max}$ values of the logistic model, the longer the time required to reach diversity saturation. Consequently, the choice of $K_{min}$ and $K_{max}$ critically influences the extent to which regional biotas reach saturation. To calibrate the $K_{min}$ and $K_{max}$ parameters, we run simulations of pairwise $K_{min}$ and $K_{max}$ combinations in a geometric sequence of base 2, from 2 to 256 genera, and tested the effect of changing the $K_{min}$ and $K_{max}$ values on the concordance between the normalized global diversities generated by the model and those estimated from the fossil record. We focused the analysis on the time-series data between the end of one mass extinction and the beginning of the next, that is, considering those time intervals dominated by rising diversity trajectories. Lin's concordance correlation coefficient (CCC) increases with increasing $K_{min}$ and $K_{max}$ until reaching a plateau, except for the mass extinction pattern of Sepkoski[20], for which it continues to increase even at the highest $K_{min}$ and $K_{max}$ values (Extended Data Fig. 4). These results were consistently replicated using alternative values for the parameters of the model that define the temperature and food dependence of the net diversification rate (Extended Data Fig. 4 (insets, grey lines) and Extended Data Table 2).

We next re-run the logistic model using the average of all $K_{min}$ and $K_{max}$ combinations giving a CCC greater than 0.70, hereafter referred to as the 'calibrated' logistic model. The calibrated model generates global diversity curves half way between the two end-member diversification models—the saturated logistic and the exponential (Fig. 1). Most of the diversity is concentrated in shallow marine environments, in which high temperatures and abundant food supplies increase the rates of diversification compared with the deep-sea habitats (Fig. 2a–f and Extended Data Fig. 5). Diversity hotspots occur in tropical shelf seas of the Early Devonian, Permian, Late Cretaceous and Cenozoic (Fig. 2c–f and Supplementary Video 3). During the Early Devonian, diversity hotspots developed on the western continental margins of Laurentia and Siberia as well as on the tropical shelves of Gondwana. The recovery of Laurentian diversity hotspots after the Late Devonian mass extinction led to the onset of Permian hotspots, which eventually disappeared during the Permian–Triassic mass extinction. Diversity hotspots became particularly prominent during the Late Cretaceous and Cenozoic in the western basins of the Tethys Ocean, the Arabian Peninsula, the Atlantic Caribbean-East Pacific and the Indo-West Pacific provinces (Fig. 2e,f). This temporal trend in the prominence of diversity hotspots cannot be explained by a secular increase in the maximum lifetime of

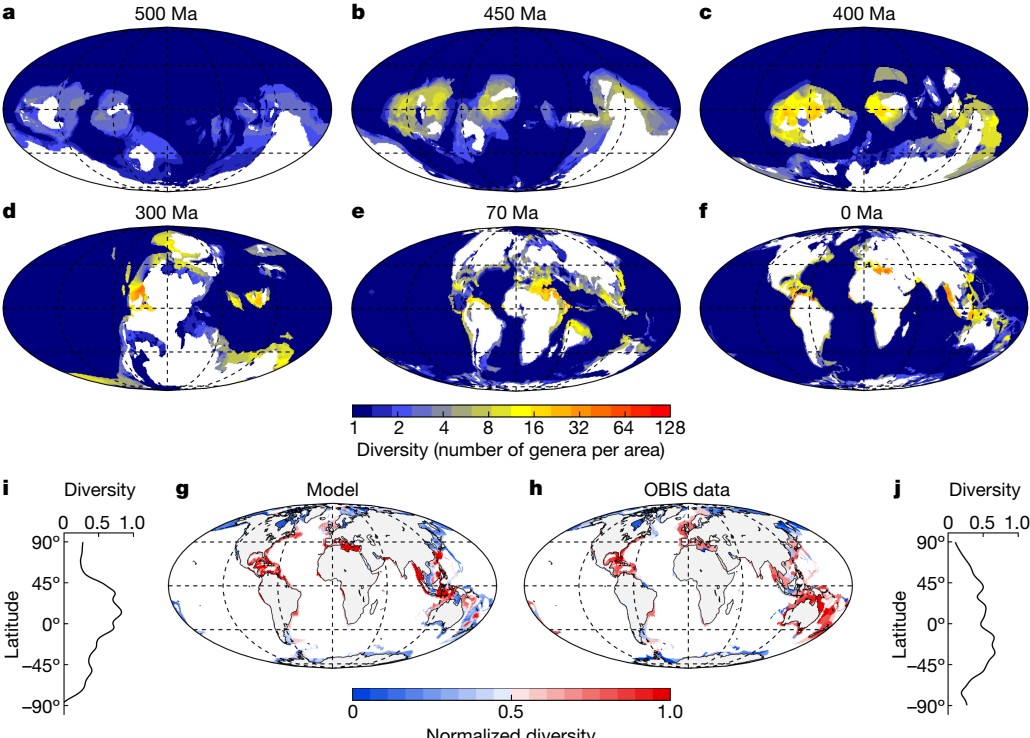

**Fig. 2 | Re-diversifying the Phanerozoic oceans. a–f,** Spatial distribution of marine animal diversity (number of genera per area) in the Cambrian (Guzhangian, 500 million years ago (Ma); **a**), Late Ordovician (Katian, 450 Ma; **b**), Early Devonian (Emsian, 400 Ma; **c**), Late Carboniferous (Pennsylvanian, 300 Ma; **d**), Late Cretaceous (Maastrichtian, 70 Ma; **e**) and the present day (**f**) generated by the calibrated logistic model after imposing the pattern of mass extinctions extracted from the fossil diversity curve of ref. [20]. This model run uses the following parameters: $Q_{10} = 1.75$, $K_{food} = 0.5$ mol C m$^{-2}$ yr$^{-1}$, net diversification rate limits ($\rho_{min} - \rho_{max}$) = 0.001–0.035 Myr$^{-1}$ (per capita), and a $K_{min}$ to $K_{max}$ range between 12 and 123 genera per area according to the calibration analysis presented in Extended Data Fig. 4. The same plots but for the mass extinction patterns extracted from the fossil diversity curves of refs. [21,22] are shown in Extended Data Fig. 5. The full Phanerozoic sequences are shown in Supplementary Video 3. **g,h,** Current spatial distributions of diversity along the continental margins from model simulations (**g**) and field observations extracted from the OBIS database (genera belonging to subphylum Crustacea and phylum Mollusca) (**h**). For the purpose of comparison, normalized diversities (0–1) bounded between quantiles 0.05 and 0.95 are represented. **i,j,** Zonal mean diversity for 20° latitudinal bands from model simulations (**i**) and field observations extracted from the OBIS database (**j**).

shelf seas. Geological data from ancient continental margins trapped within orogenic belts[23] and global tectonic reconstructions[24] show no evidence of an increase in the lifespan of passive continental margins or in the maximum ages of the seafloor over the Phanerozoic. Rather, we argue that the temporal proximity between the Ordovician–Silurian (Hirnantian stage), Late Devonian (Frasnian–Famennian stage) and Permian–Triassic mass extinctions, coinciding with a long-lived phase of marine shelves destruction during the assembly of Pangaea, interrupted the development of diversity hotspots during the Palaeozoic. In fact, by deactivating the Late Devonian mass extinction in the model, we found that the development of diversity hotspots before the end of the Permian leads to global diversities that are two to three times greater than those generated by the same calibrated logistic model with all mass extinctions enabled (Extended Data Fig. 6a–c). By contrast, the comparatively long expanse of time that separated the mass extinctions of the end-Triassic and end-Cretaceous extended the time-for-speciation under conditions of increasing continental fragmentation, giving rise to exceptionally high-diversity regions before the Cretaceous–Palaeogene mass extinction. The extraordinary diversity of Late Cretaceous hotspots ensured the continuity of relatively high diversity levels in the aftermath of the end-Cretaceous mass extinction, facilitating the subsequent development of diversity hotspots during the Cenozoic.

To evaluate the performance of the model in reconstructing the spatial distributions of diversity, we compared the results of the present-day calibrated logistic model (that is, 0 Ma) with observations of two of the most relevant groups of marine invertebrates, crustaceans and molluscs, extracted from the Ocean Biodiversity Information System (OBIS)—a global database of occurrence records of marine taxa (Methods). The regional diversity map generated by the model shows reasonable similarities to the observed diversity distributions along the continental margins (Fig. 2g,h). The main discrepancies between the model and the OBIS data occur in the surroundings of Australia and New Zealand, where the model underestimates diversity. The model lacks long-distance dispersal, which precludes a more detailed reconstruction of the spatial structuring of diversity in such a highly interconnected ocean region. On the other hand, the OBIS data are not homogeneously distributed over the global ocean and, although our analysis attempts to minimize this bias (Methods), some of the discrepancies between model and observations may be due to database limitations. Despite these discrepancies, both observed and modelled diversity decline from the equator towards the poles (Fig. 2i,j), with most diversity concentrated in the Indo-West Pacific, the Atlantic Caribbean-East Pacific and the Mediterranean (Fig. 2g,h).

Using the outputs of the calibrated logistic model, we analysed the spatial and temporal variability of the diversity-to-$K_{eff}$ ratio. This ratio provides a quantitative index of how far (ratios close to zero) or how close (ratios close to one) the regional faunas are from noticing the effect of diversity-dependent ecological factors (that is, the proximity to diversity saturation). The diversity-to-$K_{eff}$ ratio falls below 0.25 in most of the ocean and throughout the Phanerozoic (Fig. 3a–l and Supplementary Video 4), supporting the idea that the dynamics of regional diversity have been systematically operating below $K_{eff}$ and, therefore, far from saturation.

Finally, we calculated the diversity-to-$K_{eff}$ ratio along the flooded continental regions using the combinations of $K_{min}$ and $K_{max}$ that resulted from

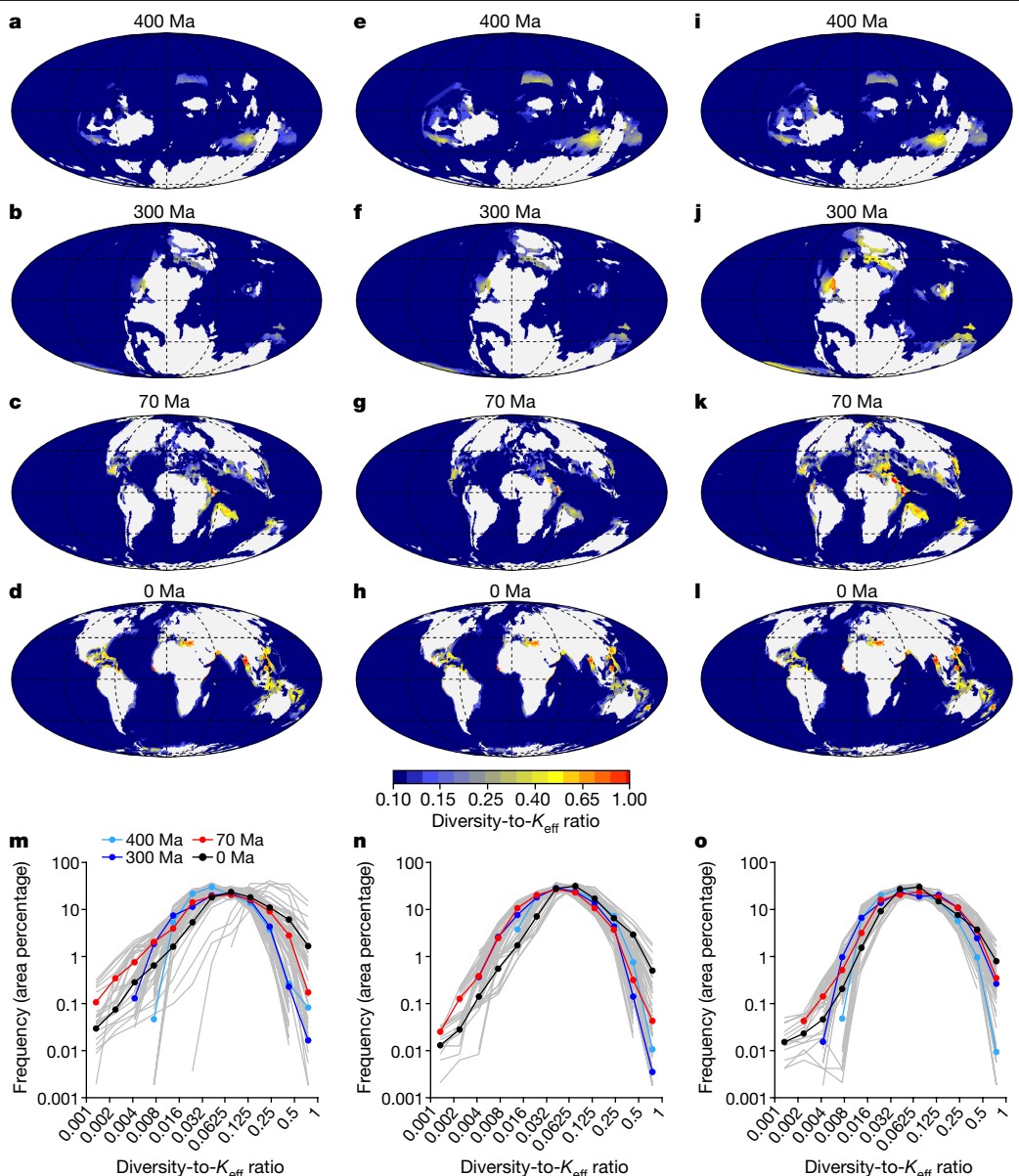

**Fig. 3 | The pervasiveness of ecological unsaturation. a–l**, Spatial distribution of the diversity-to-$K_{eff}$ ratio in deep-sea habitats and flooded continental regions of the Early Devonian (Emsian, 400 Ma; **a,e,i**), Late Carboniferous (Pennsylvanian, 300 Ma; **b,f,j**), Late Cretaceous (Maastrichtian, 70 Ma; **c,g,k**) and the present day (**d,h,l**) generated by the calibrated logistic model after imposing the pattern of mass extinctions extracted from the fossil diversity curve of ref. [20] (**a–d**), ref. [21] (**e–h**) and ref. [22] (**i–l**). The full Phanerozoic sequences are shown in Supplementary Video 4. **m–o**, Frequency distributions (area percentage) of the diversity-to-$K_{eff}$ ratio for the flooded continental regions generated by the calibrated logistic model after imposing the pattern of mass extinctions extracted from the fossil diversity curve of ref. [20] (**m**), ref. [21] (**n**) and ref. [22] (**o**). The grey lines are the frequency distributions generated from model simulations using the first 15 parameter settings listed in Extended Data Table 2. The coloured dots are average values for different Phanerozoic times.

simulations with different parameter values (Extended Data Fig. 4 (insets, grey lines) and Extended Data Table 2), and represent its frequency distributions (Fig. 3m–o). Most of the estimates fall within the exponential growth regime of the logistic function (that is, diversity-to-$K_{eff}$ ratio < 0.25). On average, less than 10% of the estimates exceed the threshold of 0.25, and less than only 2% of the estimates, those associated with well-developed diversity hotspots, exceed the threshold of 0.5.

## Discussion

Our model corroborates earlier claims that Earth's environmental history[5,25,26] and the patterns of continental fragmentation and reassembly[22,27,28] have been major determinants of marine animal diversification.

The use of a mechanistic model also enables examination of the probable causes of particular patterns in the fossil record that cannot be deduced from inspection of the fossil diversity curve alone. For example, we found that, in the absence of progressive continental reconfiguration that allows continental shelf habitats to reposition along the latitudinal temperature gradient, diversity grows disproportionately in shelf seas lying permanently within the tropical belt (Methods and Extended Data Fig. 7). We can also test the role of temperature and food dependence of the net diversification rate, and find that, by disabling these relationships in the model, highly unrealistic biogeographical distributions arise, such as the occurrence of diversity hotspots in high latitudes, leaving the growth of diversity solely as a function of habitat age (Extended Data Fig. 8). However, there are also limits to the

potential factors that can be accounted for in our modelling. For example, here we do not account for variations through time of seawater oxygenation, owing to uncertainty in both how oxygenation might set limits on maximum diversity as well as in the Phanerozoic history of the atmospheric composition itself[29]. The finite, approximately 5 million year temporal resolution of our time slices also precludes more rapid changes—such as cycles in sea level and continental flooding—from being explicitly accounted for, creating a potential temporal error in the loss or gain of shelf habitat and associated changes in biodiversity.

It has been hypothesized that the Mesozoic marine revolution[12,13], that is, the emergence of shell-crushing predators and the consequent ecological restructuring of marine ecosystems, was primarily responsible for the increase in global diversity over the last 150 million years. The fact that our model can reproduce such an increase in diversity without the need to invoke evolutionary innovations like the emergence of new modes of predation[12,13], defence[12,30], mobility[30] or reproduction[31], among others, raises a new hypothesis based on how Earth's environmental history and palaeogeographical evolution interacted in concert to enable the development of diversity hotspots. We call this the 'diversity hotspots hypothesis', which is proposed as a non-mutually exclusive alternative to the hypothesis that evolutionary innovation and new ecospace occupation led this macroevolutionary trend.

With the possible exception of diversity hotspots, our results indicate that the diversity of marine invertebrates has remained below saturation throughout their evolutionary history, shedding light on one of the most controversial topics in evolutionary ecology[1–5,8–11]. A model of taxonomic diversification operating within the exponential growth regime of a logistic function implies that diversity will recover faster than it would if it operated near saturation, with important consequences for post-extinction recovery dynamics. We envision that our spatially explicit reconstructions of diversity could shed light on other long-standing questions in biogeography and macroevolution, as well as provide a synthetic, spatially resolved history of biodiversity, that can be sampled in different ways to explore sampling biases in the rock record.

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

## Methods

### Palaeogeographical model

We use palaeogeographical reconstructions describing Earth's palaeotopography and palaeobathymetry for a series of time slices from 541 Ma to the present day. The reconstructions merge existing models from two published global reconstruction datasets—those of ref. [32] and ref. [33] (https://doi.org/10.5281/zenodo.5348492), which themselves are syntheses of a wealth of previous work.

For continental regions, estimates of palaeoelevation and continental flooding rely on a diverse range of geological evidence, such as sedimentary depositional environments and the spatiotemporal distribution of volcanic activity. For a full description, see a recent review[34]. Together, these data can be used to define the past locations of mountain ranges and palaeoshorelines[34]. For this part of our reconstruction, we used the compilation of ref. [33] with updated palaeoshorelines based on depositional environment information in current fossil databases[35]. This compilation comprises 82 palaeotopography maps covering the entire Phanerozoic. Note that each palaeogeographical map is a time slice representing the concatenation of geological data over several million years[36].

We quantified the impact of using the original compilation of ref. [33] on our model results and found only small changes with respect to using the reconstructions with updated palaeoshorelines (Extended Data Fig. 3a–c). Similarly, eustatic sea level is thought to have varied by around 100 m at timescales much shorter than the duration of the time-slices throughout the Phanerozoic[37], such that the extent of continental flooding could have varied within each time slice by an amount significant for our analysis. For this reason, and to assess the uncertainty of our results to continental palaeogeography in general, we computed additional maps of continental flooding in which the sea level is raised or lowered by 100 m compared with the original palaeo–digital elevation model grids of ref. [33] (Extended Data Fig. 3d–f).

For deep-ocean regions, the primary control on seafloor depth is the age of the seafloor, so reconstructing palaeobathymetry relies on constructing maps of seafloor age back in time[38]. As a consequence, we rely on reconstruction models that incorporate a continuous network of plate boundaries. For this part, we used the reconstruction of ref. [32] and derived maps of seafloor age from the plate tectonic model using the method of ref. [39] for which source code is available at GitHub (https://github.com/siwill22/agegrid-0.1). Palaeobathymetry is derived from the seafloor age maps following the steps outlined in ref. [38]. It is important to note that seafloor age maps for most of the Phanerozoic (that is, pre-Pangaea times) are not directly constrained by data due to recycling of oceanic crust at subduction zones. Rather, they are model predictions generated by constructing plate motions and plate boundary configurations from the geological and palaeomagnetic record of the continents. Nonetheless, the first-order trends in ocean-basin volume and mean seafloor age are consistent with independent estimates for at least the last 410 million years (Myr)[39].

The reconstructions of refs. [32,33] differ in the precise locations of the continents through time. To resolve this discrepancy, we reverse reconstructed the continental palaeoelevation model of ref. [33] to present-day coordinates using their rotation parameters, then reconstructed them back in time using the rotations of ref. [32]. Owing to the differences in how the continents are divided into different tectonic units, this process leads to some gaps and overlaps in the results[40], which we resolved primarily through a combination of data interpolation and averaging. Manual adjustments were made to ensure that the flooding history remained consistent with the original palaeotopography in areas in which interpolation gives a noticeably different history of seafloor ages. The resulting palaeotopography maps are therefore defined in palaeomagnetic reference frame[32] appropriate for use in Earth system models.

For the biodiversity modelling, we generate estimates of the age of the seafloor for discrete points within the oceans and flooded continents, and track these ages through the lifetime of each point (Supplementary Video 5). For the oceans, this is achieved using the method described in ref. [39] in which the seafloor is represented by points that are incrementally generated at the mid-ocean ridges for a series of time steps 1 Myr apart, with each point tracked through subsequent time steps based on Euler poles of rotation until either the present-day is reached, or they arrive at a subduction zone and are considered to be destroyed.

For the continents, tracking the location of discrete points is generally simpler as most crust is conserved throughout the timespan of the reconstruction. In contrast to the deep oceans (where we assume that crust is at all times submerged), we model the 'age' of the seafloor from the history of continental flooding and emergence within the palaeogeographical interpretation[33]. The continents are seeded with uniformly distributed points at the oldest timeslice (541 Ma) at which they are assigned an age of zero. These points are tracked to subsequent time slices of which the palaeogeography is used to determine whether the point lies within a flooded or emergent region. Points within flooded regions of continents are considered to be seafloor, and the age of this seafloor is accumulated across consecutive time slices where a given point lies within a flooded region. When a point is within an emergent region, the seafloor age is reset to zero. Following this approach, individual points within stable continents may undergo several cycles of seafloor age increasing from zero before being reset. At the continental margins formed during the Pangaea breakup, the age of the seafloor continuously grows from the onset of rifting. Intraoceanic island arcs represent an additional case, which can appear as new tectonic units with the reconstructions at various times. In these cases, we assume that the seafloor has a zero-age at the time at which the intraoceanic arc first develops, then remains predominantly underwater for the rest of its lifetime.

Thus, for each of the 82 palaeogeographical reconstructions, we annotate 0.5° by 0.5° grids as continental, flooded continental shelf or oceanic for later use in model coupling and production of regional diversity maps.

### Palaeoenvironmental conditions under the cGENIE Earth system model

We use cGENIE[41], an Earth system model of intermediate complexity, to simulate palaeoenvironmental conditions of seawater temperature and organic carbon export production (as a surrogate for food supply) throughout the Phanerozoic (from 541 Ma to the present day).

cGENIE is based on a three-dimensional (3D) ocean circulation model coupled to a 2D energy–moisture balance atmospheric component and a sea-ice module. We configured the model on a 36 × 36 (latitude, longitude) equal area grid with 17 unevenly spaced vertical levels in depth, down to a maximum ocean depth of 5,900 m. The cycling of carbon and associated tracers in the ocean is based on a size-structured plankton ecosystem model with a single (phosphate) nutrient[42,43], and adopts an Arrhenius-type temperature-dependent scheme for the remineralization of organic matter exported to the ocean interior[44].

cGENIE provides a spatially resolved representation of ocean physics and biogeochemistry, which is a prerequisite for the present study to be able to reconstruct the spatial patterns of biodiversity in deep time. However, owing to the computational impracticality of generating a single transient simulation of physics (that is, temperature) and biogeochemistry (that is, export production) over the entire Phanerozoic, we therefore generate 30 model equilibria at regular time intervals throughout the Phanerozoic that are subsequently used as inputs for the regional diversification model (see the 'Model coupling' section).

We used 30 Phanerozoic palaeogeographical reconstructions through time (~20 Myr evenly spaced time intervals) to represent key time periods. For each continental configuration corresponding to a given age in Earth history, we generate idealized 2D (but zonally averaged) wind speed and wind stress, and 1D zonally averaged albedo forcing fields[45] required by the cGENIE model using the 'muffingen'

open-source software (see the 'Code availability' section). For each palaeogeographical reconstruction, the climatic forcing (that is, solar irradiance and carbon dioxide concentration) is adapted to match the corresponding geological time interval. The partial pressure of $CO_2$ is taken from the recent update of the GEOCARB model[46]. Solar luminosity is calculated using the model of stellar physics of ref. [47]. We impose modern-day orbital parameters (obliquity, eccentricity and precession). The simulations are initialized with a sea-ice-free ocean, homogeneous oceanic temperature (5 °C) and salinity (34.9‰). As variations in the oceanic concentration of bio-available phosphate remain challenging to reconstruct in the geological past[48,49], we impose a present-day mean ocean phosphate concentration (2.159 µmol kg$^{-1}$) in our baseline simulations. We quantify the impact of this uncertainty on our model results by conducting additional simulations using half and twice the present-day ocean phosphate concentration (Extended Data Fig. 3g–i). For each ocean phosphate scenario (that is, 0.5×, 1× and 2× the present-day value), each of the 30 model simulations is then run for 20,000 years, a duration ensuring that deep-ocean temperature and geochemistry reach equilibrium. For each model simulation, the results of the mean annual values of the last simulated year are used for the analysis. Note that, although cGENIE makes projections of the distribution of dissolved oxygen ([$O_2$]) in the ocean, our diversification model does not currently consider oxygenation to be a limit on diversity. Thus, we assumed a modern atmospheric partial pressure of $O_2$ in all 30 palaeo simulations and did not use the resulting projected [$O_2$] fields.

**Regional diversification model**

We tested two models of diversification—the logistic model and the exponential model—describing the dynamics of regional diversity over time. In both models, the net diversification rate ($\rho$), with units of inverse time (Myr$^{-1}$), varies within a pre-fixed range of values as a function of seawater temperature and food availability. The net diversification rate is then calculated for a given location and time according to the following equation:

$$\rho = \rho_{max} - (\rho_{max} - \rho_{min})(1 - Q_{temp}Q_{food}) \qquad (1)$$

where $\rho_{min}$ and $\rho_{max}$ set the lower and upper net diversification rate limits within which $\rho$ is allowed to vary, and $Q_{temp}$ and $Q_{food}$ are non-dimensional limitation terms with values between 0 and 1 that define the dependence of $\rho$ on temperature and food, respectively (Extended Data Table 1).

The model considers a direct relationship between seawater temperature, food supply and the rate of net diversification on the basis of the theoretical control that temperature and food supply exert on the rates of origination and extinction (Supplementary Fig. 1). Temperature rise is expected to accelerate the biochemical kinetics of metabolism[50] and shorten the development times of individuals[51], leading to higher rates of mutation and origination. Greater food availability increases population sizes, which increases the rates of mutation and reduces the probability of extinction[52]. Furthermore, a large body of observations shows the existence of a positive relationship between resource availability (that is, food supply) and the standing stock of species in marine and terrestrial communities[53,54]. A larger food supply would support a greater number of individuals. A greater diversity of food resources could also lead to a finer partitioning of available resources[55].

The temperature dependence of $\rho$ is calculated using the following equation:

$$Q_{temp} = \frac{Q_{10}^{\frac{T-T_{min}}{10}}}{Q_{10}^{\frac{T_{max}-T_{min}}{10}}} \qquad (2)$$

where the $Q_{10}$ coefficient measures the temperature sensitivity of the origination rate. In equation (2) above, $T$ is the seawater temperature (in °C)

at a given location and time, and $T_{min}$ and $T_{max}$ are the 0.01 percentile and the 0.99 percentile, respectively, of the temperature frequency distribution in each time interval. In the model, the values of $T_{min}$ and $T_{max}$ used to calculate $Q_{temp}$ are therefore recomputed for every time interval (~5 Myr) according to the temperature frequency distribution of the corresponding time interval. This enables us to use updated $T_{min}$ and $T_{max}$ values in each Phanerozoic time interval and to account for the thermal adaptation of organisms to ever changing climate conditions.

The food limitation term is parameterized using a Michaelis–Menten formulation as follows:

$$Q_{food} = \frac{POC\ flux}{(K_{food} + POC\ flux)} \qquad (3)$$

where POC flux (mol m$^{-2}$ yr$^{-1}$) is the particulate organic carbon export flux, which is used as a surrogate for food availability, at a given location and time of the simulated seafloor. The parameter $K_{food}$ (mol m$^{-2}$ yr$^{-1}$) in equation (3) is the half-saturation constant, that is, the POC flux at which the diversification rate is half its maximum value, provided that other factors were not limiting. These temperature and food supply limitation terms vary in space and time as a result of changes in seawater temperature and particulate organic carbon export rate, respectively, thereby controlling the spatial and temporal variability of $\rho$ (Supplementary Video 6).

The net diversification rate becomes negative (1) in the event of mass extinctions or (2) in response to regional-scale processes, such as sea-level fall and/or seafloor deformation along convergent plate boundaries. Mass extinction events are imposed as external perturbations to the diversification model by imputing negative net diversification rates to all active seafloor points (ocean points and flooded continental points) and assuming non-selective extinction. The percentage of diversity loss as well as the starting time and duration of mass extinctions are extracted from three fossil diversity curves of reference[20–22] (Source Data for Fig. 1). Each of these fossil diversity curves provides different insights into the Phanerozoic history of marine animal diversity based on uncorrected range-through genus richness estimates[20,22] and sampling standardized estimates[21]. Regional-scale processes—such as sea level fall during marine regressions and/or seafloor destruction at plate boundaries, either by subduction or uplift—are simulated by the combined plate tectonic–palaeoelevation model, and constrain the time that seafloor habitats have to accumulate diversity.

The model assumes non-selective extinction during mass extinction events (that is, the field of bullets model of extinction; everything is equally likely to die, no matter the age of the clade and regardless of adaptation)[56]. However, there is much fossil evidence supporting extinction selectivity[57,58]. It could be argued that higher extinction rates at diversity hotspots would have delayed their subsequent recovery, flattening global diversity trends. This argument is difficult to reconcile with Sepkoski's genus-level global diversity curve but could be consistent with the standardized diversity curve of ref. [21]. Similarly, the model is also not suitable for reproducing the explosive radiations of certain taxonomic groups after mass extinctions, which could explain the offset between the model and fossil observations in the early Mesozoic (Fig. 1).

Letting $D$ represent regional diversity (number of genera within a given seafloor point) and $t$ represent time, the logistic model is formalized by the following differential equation:

$$\frac{\partial D(t)}{\partial t} = \rho D\left[1 - \frac{D}{K_{eff}}\right] \qquad (4)$$

where $D(t)$ is the number of genera at time $t$ and $K_{eff}$ is the effective carrying capacity or maximum number of genera that a given seafloor point (that is, grid cell area after gridding) can carry at that time, $t$. In our logistic model, $K_{eff}$ is allowed to vary within a fixed range of values

(from $K_{min}$ to $K_{max}$) as a positive linear function of the POC flux at a given location and time as follows:

$$K_{eff} = K_{max} - (K_{max} - K_{min}) \frac{POC\ flux_{max} - POC\ flux}{POC\ flux_{max} - POC\ flux_{min}} \quad (5)$$

where POC flux$_{min}$ and POC flux$_{max}$ correspond to the 0.01 and 0.99 quantiles of the POC flux range in the whole Phanerozoic dataset.

In the logistic model, the net diversification rate decreases as regional diversity approaches its $K_{eff}$. The exponential model is a particular case of the logistic model when $K_{eff}$ approaches infinity and, therefore, neither the origination rate nor the extinction rate depend on the standing diversities. In this scenario, diversity grows in an unlimited manner over time only truncated by the effect of mass extinctions and/or by the dynamics of the seafloor (creation versus destruction). Thus, the exponential model is as follows:

$$\frac{\partial D(t)}{\partial t} = \rho D \quad (6)$$

where the rate of change of diversity (the time derivative) is proportional to the standing diversity $D$ such that the regional diversity will follow an exponential increase in time at a speed controlled by the temperature- and food-dependent net diversification rate. Even if analytical solutions exist for the steady-state equilibrium of the logistic and exponential functions, we solved the ordinary differential equations (4) and (6) using numerical methods with a time lag of 1 Myr to account for the spatially and temporally varying environmental constraints, seafloor dynamics and mass extinction events.

As the analysis of global fossil diversity curves is unable to discern the causes of diversity loss during mass extinctions, our imputation of negative diversification rates could have overestimated the loss of diversity in those cases in which sea level fall, a factor already accounted for by our model, contributed to mass extinction. This effect was particularly recognizable across the Permian–Triassic mass extinction (Extended Data Fig. 6d–f), and supports previous claims that the decline in the global area of the shallow water shelf exacerbated the severity of the end-Permian mass extinction[34].

## Model coupling

As stated above, the coupled plate tectonic–palaeoelevation (palaeogeographical) model corresponds to a tracer-based model (a Lagrangian-based approach) that simulates and tracks the spatiotemporal dynamics of ocean and flooded continental points. The diversification models start at time 541 Ma with all active points having a $D_0 = 1$ (one single genus everywhere) and we let points accumulate diversity heterogeneously with time according to seafloor age distributions (for ocean points) and the time that continents have been underwater (for flooded continental points). The ocean points are created at mid-ocean ridges and disappear primarily at subduction zones. Between their origin and demise, the points move following plate tectonic motions and we trace their positions while accumulating diversity. The flooded continental points begin to accumulate diversity from the moment that they are submerged, starting with a $D$ value equal to the nearest neighbour flooded continental point with $D > 1$, thereby simulating a process of coastal recolonization (or immigration). The diversification process remains active while the seafloor points remain underwater, but it is interrupted, and $D$ set to 0, in those continental points that emerge above sea level. Similarly, seafloor points corresponding to ocean domains disappear in subduction zones, and their diversity is lost. We track the geographical position of the ocean and flooded continental points approximately every 5 Myr, from 541 Ma to the present. Each and every one of the tracked points accumulates diversity over time at a different rate, which is modulated by the environmental history (seawater temperature and food availability) of each point, as

described in equations (1)–(3). When a point arrives in an environment with a carrying capacity lower than the diversity it has accumulated through time, we reset the diversity of the point to the value of the carrying capacity, thereby simulating local extinction.

Seawater temperature ($T$) and food availability (POC flux) are provided by the cGENIE model, which has a spatial and temporal resolution coarser than the palaeogeographical model. The cGENIE model provides average seawater $T$ and POC flux values in a 36 × 36 equal area grid (grid cell area equivalent to 2° latitude by 10° longitude at the equator) and 30 time slices or snapshots (from 541 Ma to the present: each ~20 Myr time intervals). To have environmental inputs for the 82 time slices of the plate tectonic–palaeoelevation model, we first interpolate the cGENIE original model output data on a 0.5° by 0.5° grid to match the annotated grids provided by the plate tectonic–palaeoelevation model. As the relatively coarse spatial resolution of the cGENIE model prevents rendering the coast–ocean gradients, we assign surface $T$ and POC flux at the base of the euphotic zone to the flooded continental shelf grid cells, and deep ocean $T$ and POC flux at the bottom of the ocean to the ocean grid cells. As there are time slices without input data of seawater $T$ and POC flux, we interpolate/extrapolate seawater $T$ and POC flux values into the 0.5° by 0.5° flooded continental shelf and ocean grids independently. Finally, we interpolate values from these 0.5° by 0.5° flooded continental shelf and ocean grids into the exact point locations in each time frame. Thus, each active point is tracked with its associated time-varying $T$ and POC flux values throughout its lifetime. On average, 6,000 flooded continental points and 44,000 oceanic points were actively accumulating diversity in each time frame. The model cannot simulate the singularities of relatively small enclosed seas for which the spatial resolution of the palaeogeographical and Earth system models is insufficient to capture relevant features (such as palaeobathymetry, seawater temperature) in detail. The method is also likely to underestimate the diversity of epeiric (inland) seas due to the difficulty of simulating immigration, a process that is strongly influenced by the effect of surface ocean currents and is not considered here. However, as stated above, the model considers recolonization of recently submerged areas by marine biota from nearby coastal environments, which partially explains coastal immigration.

## Estimation of global diversity from regional diversity

Our regional diversity maps are generated by separately interpolating ocean point diversity and flooded continental point diversity into the 0.5° by 0.5° annotated grids provided by the palaeogeographical model. We calculate global diversity at each time step from each of the regional diversity maps following a series of steps to integrate diversity along line transects from diversity peaks (maxima) to diversity troughs (minima) (Extended Data Fig. 1). To select the transects, first, we identify on each of the regional diversity maps the geographical position of the diversity peaks. We identify local maxima (that is, grid cells with diversity greater than their neighbour cells), and define the peaks as those local maxima with diversity greater than the 0.75 quantile of diversity values in all local maxima in the map. In the case of grid cells with equal neighbour diversity, the peak is assigned to the grid cell in the middle. We subsequently identify the geographical position of the diversity troughs, which are defined as newly formed ocean grid cells (age = 0 Myr) and, therefore, with diversities equal to one. The troughs are mostly located at mid-ocean ridges.

On each of the 82 spatial diversity maps, we trace a line transect from each diversity peak to its closest trough, provided that the transect does not cross land in more than 20% of the grid cells along the linear path (Supplementary Video 7). On average, for each spatial diversity map, we trace 400 ($\sigma = \pm 75$) linear transects. This sampling design gives rise to transects of different lengths, which may bias the estimates of global diversity. To minimize this bias, we cut the tail of the transects to have a length of 555 km equivalent to 5° at the equator. We tested an alternative cut-off threshold, 1,110 km, and the results do not alter the study's conclusions.

We apply Bresenham's line algorithm[59] to detect the grid cells crossed by the transects and annotate their diversity. To integrate regional diversity along the transects, we developed a method to simplify the scenario of peaks and troughs heterogeneously distributed on the 2D diversity maps. The method requires (1) a vector (the transect) of genus richness ($\alpha_n$) at $n$ different locations (grids) arranged in a line (1D) of $L$ grids, and (2) a coefficient of similarity ($V_{n,n+1}$) between each two neighbouring locations, $n$ and $n+1$. $V_{n,n+1}$, the coefficient of similarity, follows a decreasing exponential function with distance between locations. The number of shared genera between $n$ and $n+1$ is $V_{n,n+1} \times \min(\alpha_n; \alpha_{n+1})$. We integrate diversity from peaks to troughs and assume that, along the transect, $\alpha_{n+1}$ is lower than $\alpha_n$. We further assume that the genera present in $n$ and $n+2$ cannot be absent from $n+1$. Using this method, we integrate the transect's diversity ($\gamma_i$) using the following equation:

$$\gamma_i = \alpha_1 + \sum_{n=1}^{L-1} (1 - V_{n,n+1}) \alpha_{n+1} \qquad (7)$$

To integrate the diversity of all transects ($\gamma_i$) on each 2D diversity map (or time slice), we apply the same procedure as described above (Extended Data Fig. 1). We first sort the transects in descending order from the highest to the lowest diversity. We then assume that the number of shared genera between transect $i$ and the rest of the transects with greater diversity $\{1, 2, ..., i-1\}$ is given by the distance of its peak to the nearest neighbour peak (NN($i$)) of those already integrated $\{1, 2, ..., i-1\}$. Thus, we perform a zigzag integration of transects' diversities down gradient, from the greatest to the poorest, weighted by the nearest neighbour distance among the peaks already integrated. As a result, the contribution of each transect to global diversity will depend on its diversity and its distance to the closest transect out of all those transects already integrated. Using this method, we linearize the problem to simplify the cumbersome procedure of passing from a 2D regional diversity map to a global diversity estimate without knowing the identity (taxonomic affiliation) of the genera. If $\gamma_{total}$ is the global diversity at time $t$:

$$\gamma_{total} = \gamma_1 + \sum_{i=2}^{j} (1 - V_{NN(i),i}) \gamma_i \qquad (8)$$

Finally, the resulting global estimates are plotted against the midpoint value of the corresponding time interval to generate a synthetic global diversity curve. To compare the global diversity curves produced by the diversification models with those composed from the fossil record, Lin's CCC[60] is applied to the data normalized to the min–max values of each time series (that is, rescaled within the range 0–1). Lin's CCC combines measures of both precision and accuracy to determine how far the observed data deviate from the line of perfect concordance (that is, the 1:1 line). Lin's CCC increases in value as a function of the nearness of the data's reduced major axis to the line of perfect concordance (the accuracy of the data) and of the tightness of the data around its reduced major axis (the precision of the data).

The time series of global diversity generated from the fossil record and from the diversification model exhibit serial correlation and the resulting CCCs are therefore inflated. The use of methods for analysing non-zero autocorrelation time series data, such as first differencing or generalised least squares regression, enables high-frequency variations along the time series to be taken into account. However, the relative simplicity of our model, which was designed to reproduce the main Phanerozoic trends in global diversity, coupled with the fact that biases in the fossil data would introduce uncertainty into the analysis, leads us to focus our analysis on the long-term trends, obviating the effect of autocorrelation.

## Model parameterization and calibration

The diversification models are parameterized assuming a range of values that constrain the lower and upper limits of the genus-level net diversification rate ($\rho_{min}$ and $\rho_{max}$, respectively) (Extended Data Table 1) according to previously reported estimates from fossil records (figures 8 and 11 of ref. [5]). A range of realistic values is assigned for the parameters $Q_{10}$ and $K_{food}$, determining, respectively, the thermal sensitivity and food dependence of the net diversification rate. We test a total of 40 different parameter combinations (Extended Data Table 2). The resulting estimates of diversity are then compared against the fossil diversity curves of ref. [20], ref. [21] or ref. [22], and the 15 parameter combinations providing the highest CCCs are selected.

The results of the logistic diversification model rely on the values of the minimum and maximum carrying capacities ($K_{min}$ and $K_{max}$, respectively) within which the spatially resolved effective carrying capacities ($K_{eff}$) are allowed to vary. The values of $K_{min}$ and $K_{max}$ are therefore calibrated by running 28 simulations of pair-wise $K_{min}$ and $K_{max}$ combinations increasing in a geometric sequence of base 2, from 2 to 256 genera (Extended Data Fig. 4). We perform these simulations independently for each of the 15 parameter settings selected previously (Extended Data Fig. 4 and Extended Data Table 2). Each combination of $K_{min}$ and $K_{max}$ produces a global diversity curve, which is evaluated as described above using Lin's CCC.

Calculating estimates of global diversity from regional diversity maps in the absence of information on genus-level taxonomic identities requires that we assume a spatial turnover of taxa with geographical distance (Extended Data Fig. 1). Distance-decay curves are routinely fitted by calculating the ecological similarity (for example, the Jaccard similarity index) between each pair of sampling sites, and fitting an exponential decay function to the points on a scatter plot of similarity ($y$ axis) versus distance ($x$ axis). Following this method, we fit an exponential decay function to the distance–decay curves reported in ref. [61], depicting the decrease in the Jaccard similarity index ($J$) of fossil genera with geographical distance (great circle distance) at different Phanerozoic time intervals:

$$J = J_{off} + (J_{max} - J_{off}) e^{-\lambda \times distance} \qquad (9)$$

where $J_{off} = 0.06$ (n.d.) is a small offset, $J_{max} = 1.0$ (n.d.) is the maximum value of the genus-based Jaccard similarity index and $\lambda = 0.0024$ (km$^{-1}$) is the distance-decay rate.

The Jaccard similarity index ($J$) between consecutive points $n$ and $n+1$ is bounded between 0 and $\min(\alpha_n; \alpha_{n+1})/\max(\alpha_n; \alpha_{n+1})$. A larger value for $J$ would mean that there are more shared genera between the two communities than there are genera within the least diverse community, which is ecologically absurd. However, using a single similarity decay function can lead the computed value of $J$ to be locally larger than $\min(\alpha_n; \alpha_{n+1})/\max(\alpha_n; \alpha_{n+1})$. To prevent this artefact, we use the Simpson similarity index or 'overlap coefficient' ($V$) instead of $J$. $V$ corresponds to the percentage of shared genera with respect to the least diverse community ($\min(\alpha_n; \alpha_{n+1})$). $V$ is bounded between 0 and 1, whatever the ratio of diversities. As the pre-existing estimates of similarity are expressed using $J$ (ref. [61]), we perform the conversion from $J$ to $V$ using the algebraic expression $V = (1 + R) \times J/(1 + J)$ where $R = \max(\alpha_n; \alpha_{n+1})/\min(\alpha_n; \alpha_{n+1})$ (Supplementary Note 1). In the cases in which $J$ exceeds the $\min(\alpha_n; \alpha_{n+1})/\max(\alpha_n; \alpha_{n+1})$, $V$ becomes >1 and, in those cases, we force $V$ to be <1 by assuming $R = 1$, that is $\alpha_n = \alpha_{n+1}$.

The model considers a single distance-decay function for the spatial turnover of taxonomic composition. However, the degree of provinciality (that is, the partitioning of life into distinct biogeographical units) varies in space and time as a result of environmental gradients[62] and plate tectonics[63]. In fact, the increase in provinciality has been invoked as the main driver of the increase in global diversity, especially in the Late Cretaceous and Cenozoic[22,62,63]. This is a deficiency of the model. Unfortunately, information on the extent to which marine provinciality has varied in space and time throughout the Phanerozoic is limited[61,62], and there is no simple (mechanistic) way to implement different distance-decay functions of taxonomic similarity in the model. There

is a clear difference between longitudinal and latitudinal distance, the latter being a more significant source of taxonomic turnover[62]. This effect would add to the observation that tropical diversity hotspots became more prominent towards the end of the Phanerozoic, offering two complementary explanations for the increase in diversity in the Mesozoic: (1) favourable conditions for the development of diversity hotspots and (2) a higher degree of provinciality.

## Fossil data

We used three fossil diversity curves of reference[20–22] to (1) extract the patterns of mass extinctions (starting time, duration and magnitude) imposed on the model, and (2) compare the global diversity curves produced by the model with those generated from fossil data. Sepkoski's global diversity curve corresponds to marine invertebrates listed in Sepkoski's published marine genus compendium[20] (data downloaded from Sepkoski's Online Genus Database; http://strata.geology.wisc.edu/jack/). The global diversity curves of refs. [21,22] are digitized from the original sources, that is, figure 3 of ref. [21] and figure 2a of ref. [22], respectively, using the free software XYscan. The curve reported by ref. [21] corresponds to genus-richness estimates obtained after correcting for sampling effort using the shareholder quorum subsampling technique. This curve is binned at approximately 11 Myr time intervals and includes non-tetrapod marine animals of which Anthozoa, Trilobita, Ostracoda, Linguliformea, Articulata, Bryozoa, Crinoidea, Echinoidea, Graptolithina, Conodonta, Chondrichthyes, Cephalopoda, Gastropoda and Bivalvia are the major taxonomic groups. The curve reported in ref. [22] corresponds to 1 Myr range-through richness estimates of marine skeletonized invertebrate genera including Brachiopoda, Bivalvia, Anthozoa, Trilobita, Gastropoda, Crinoidea, Blastoidea, Edrioasteroidea, Ammonoidea, Nautiloidea and Bryozoa. All digitized (and interpolated) diversity data and the net diversification rate data imputed by the model to simulate mass extinctions are provided as Source Data for Fig. 1.

The choice to analyse the data for all animals is somewhat in conflict with the fact that, in the open ocean, photosynthetic primary production and the flux of organic matter at the bottom of the water column are decoupled. This decoupling leads to contrasting differences in the amount of food available to the planktonic/nektonic and benthic communities yet, in the model, we assume that the amount of organic carbon reaching the seafloor is a proxy for food supply. Given that most of the diversity is concentrated in shallow shelf seas, this assumption is likely to be of only relatively minor importance in a global context.

## OBIS data

We use the occurrence records of genera belonging to two of the most diverse marine invertebrate groups: subphylum Crustacea and phylum Mollusca, as downloaded from the Ocean Biodiversity Information System (OBIS) on 22 October 2021 (www.obis.org). The list of genera is validated with the genera names in WoRMS (https://www.marinespecies.org) and only the accepted, extant and marine names were selected for the analysis. This corresponds to a total of 10,018,142 records of 9,750 genera (6,540,489 records and 5,533 genera of crustaceans and 3,477,653 records and 4,217 genera of molluscs) collected from 1920 to 2021. The records are gridded into hexagons (800,000 $km^2$ at the equator) to account for different gamma (regional) diversity across latitudes, otherwise, a bias would occur in the resulting estimates. To ensure sufficient sampling size, we selected only those hexagons with more than or equal to 10 occurrence records and with more than three genera. Furthermore, we use the frequencies of the genera to estimate the number of unobserved genera per hexagon. We do so by extrapolating the number of genera on the basis of a bias-corrected Chao estimate according to the tail of rare genera (that is, those genera that have only one or two occurrence records in a hexagon)[64,65]. The final number of genera per hexagon is the sum of the observed and unobserved estimates of genera. The analysis is performed with the package vegan[66] in R v.4.1.2. Finally, we spatially overlap the hexagons

and 0.5° × 0.5° square grid to match the map of the palaeo analysis and extract the value of the diversity index per coastal grid in QGIS v.3.22.0. The comparison between model and observations is made on the normalized diversities (0–1) bounded between the 0.05 and 0.95 quantiles to minimize the effect of outliers in the observed pattern. These diversity data are provided as Source Data for Fig. 2.

## Testing a static (null) palaeogeographical model

To evaluate the effect of palaeogeography on global diversity dynamics, we carry out simulations for three static palaeogeographical configurations: the Devonian (400 Ma), the Carboniferous (300 Ma) and the present. For each of these three configurations, the model runs for 541 Myr in a 'static mode', that is, diversity accumulates steadily at a pace determined by the temperature and food assigned to each grid at the selected static configuration. Mass extinctions are imposed the same way we do in the default model with variable palaeogeography. The test is performed for the exponential diversification model and the calibrated logistic model and for each of the three mass extinction patterns[20–22]. Extended Data Fig. 7a–i shows the differences between the log-transformed normalized diversities (between 0 and 1) produced by the calibrated logistic model with static palaeogeography (nDiv tectonics OFF) and with variable palaeogeography (nDiv tectonics ON). Red and blue colours denote, respectively, the extent to which the static model produces diversity estimates above or below those produced by the model with plate tectonics. Tropical regions are dominated by reddish colours indicating that the static model particularly overestimates diversity in these regions, where high temperatures accelerate diversification. In the absence of plate tectonics, the model leads to a scenario of uncontrolled diversity growth (mainly in the tropical shelf seas; reddish areas on maps) such that even mass extinctions cannot dampen diversity growth (Extended Data Fig. 7j–l) and only effective carrying capacities prevent diversity from running away (Extended Data Fig. 7m–o). A static geographical configuration also prevents diversity hotspots from disappearing at convergent plate boundaries. These results support the idea that Earth's palaeogeographical evolution and sea level changes, by creating, positioning and destroying seafloor habitats, have had a key role in constraining the growth of diversity throughout the Phanerozoic.

## Reporting summary

Further information on research design is available in the Nature Research Reporting Summary linked to this paper.

## Data availability

Fossil diversity digitized data and those data downloaded from the Sepkoski's Online Genus Database (http://strata.geology.wisc.edu/jack/) are provided as source data with this paper. Diversity data downloaded from the Ocean Biodiversity Information System (OBIS) database (www.obis.org) are also provided as source data with this paper. Source data are provided with this paper.

## Code availability

The coupled palaeogeographical–diversification model presented here uses input data of seafloor age distributions and palaeoenvironmental conditions from the siwill22/agegrid-0.1 v1-alpha palaeogeographical model and the cGENIE Earth system model, respectively. The code used to reconstruct seafloor age distributions from GPlates full-plate tectonic reconstructions is available at Zenodo (https://doi.org/10.5281/zenodo.3271360). The code for the version of the 'muffin' release of the cGENIE Earth system model used in this paper is tagged as v0.9.30, and is available at Zenodo (https://doi.org/10.5281/zenodo.6676435). Configuration files for the specific experiments presented in the paper can be found in the directory: genie-userconfigs/PUBS/published/

Cermeno_et_al.2022. Details of the experiments, plus the command line needed to run each one, are given in the readme.txt file in that directory. The code for the muffingen boundary file generator, is available at Zenodo (https://doi.org/10.5281/zenodo.6676451). The code for the coupled palaeogeographical–diversification model is available at Zenodo (https://doi.org/10.5281/zenodo.6535496). The code is also available at GitHub (https://github.com/CarmenGarciaComas/INDITEK); this is the recommended mode of access as it will contain any updates and clarifications.

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

**Acknowledgements** This work was funded by research grant CGL2017-91489-EXP from the Spanish government. A.P. acknowledges funding from the European Union's Horizon research and innovation programme under the Marie Skłodowska-Curie grant agreement no. 838373. S.W. was supported by NSFC grant no. 41972237 and ARC grant nos DP180102280 and DP200100966. M.J.B. acknowledges support through ERC Advanced Grant 'Innovation' (ERC 788203). P.C. and S.M.V. acknowledge support through grant no. CTM2017-87227-P. A.R. acknowledges NSF grant EAR-2121165, as well as support from the Heising–Simons Foundation. The Institut de Ciències del Mar (CSIC) received funding from the Spanish government through the 'Severo Ochoa Centre of Excellence' accreditation (CEX2019-000928-S).

**Author contributions** P.C. and C.G.-C. proposed and designed the study. P.C., C.G.-C. and S.M.V. developed the diversification model. A.P. and A.R. performed the cGENIE model simulations. S.W. and R.D.M. prepared the plate tectonic/palaeoelevation model reconstructions. C.G.-C. performed the coupling of the diversification model to the plate tectonic/palaeoelevation model. C.G.-C. and G.L.G. developed the method to estimate global diversity from regional diversity. C.C. analysed OBIS data for marine invertebrates. P.C., C.G.-C., A.P., S.W., M.J.B., C.C, G.L.G., R.D.M., A.R. and S.M.V. contributed to data analysis and discussion of results. P.C., C.G.-C., A.P. and S.W. wrote the manuscript with inputs from all of the authors.

**Competing interests** The authors declare no competing interests.

**Additional information**
**Correspondence and requests for materials** should be addressed to Pedro Cermeño or Carmen García-Comas.

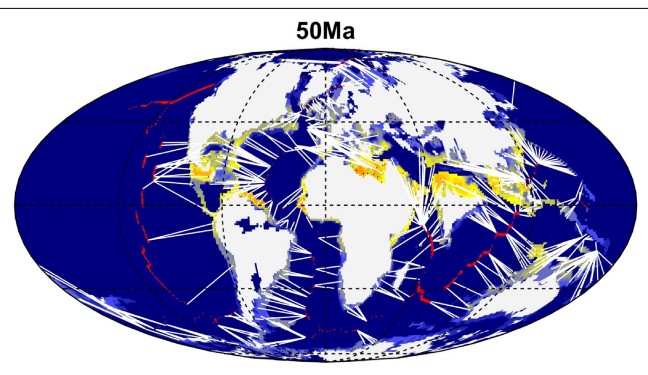

**50Ma**

**(1) Tracing transects from diversity peaks to diversity troughs at time bin X**

## (2) Integrating diversity along transects

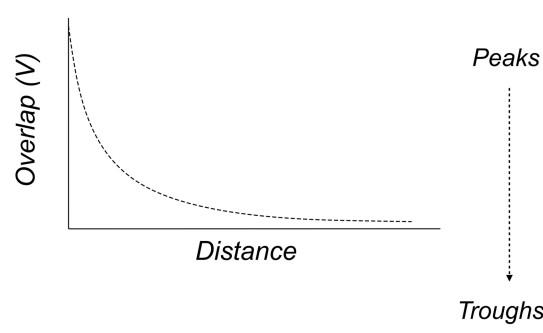

*Overlap (V)* vs *Distance*

*Transects*

| Peaks | | | | | |
|---|---|---|---|---|---|
| 50 | 15 | 80 | 140 | 20 | $\alpha_1$ |
| 40 | 8 | 60 | 80 | 16 | $\alpha_n$ |
| 30 | 5 | 45 | 60 | 13 | $\alpha_{n+1}$ |
| 20 | 3 | 28 | 40 | 10 | . |
| 10 | 1 | 15 | 20 | 7 | . |
| 1 | | 7 | 1 | 3 | . |
| | | 1 | | 1 | $\alpha_L$ |

*Troughs*

$$\gamma_i = \alpha_1 + \sum_{n=1}^{L-1} \left(1 - V_{n,n+1}\right) \alpha_{n+1}$$

| 56 | 16 | 88 | 151 | 22 | $\gamma_i$ |
|---|---|---|---|---|---|
| $\gamma_3$ | $\gamma_5$ | $\gamma_2$ | $\gamma_1$ | $\gamma_4$ | |

## (3) Integrating transects' diversity

| $\gamma_1$ | 151 |
|---|---|
| $\gamma_2$ | 88 |
| $\gamma_3$ | 56 |
| $\gamma_4$ | 22 |
| $\gamma_5$ | 16 |

Pair-wise distance between transects (nearest-neighbours, NN)

| | $\gamma_2$ | $\gamma_3$ | $\gamma_4$ | $\gamma_5$ |
|---|---|---|---|---|
| $\gamma_1$ | 10000 | 10500 | 1800 | 6000 |
| $\gamma_2$ | | 1500 | 2500 | 2500 |
| $\gamma_3$ | | | 2000 | 900 |
| $\gamma_4$ | | | | 18000 |

V between pair of NN transects

| | $\gamma_2$ | $\gamma_3$ | $\gamma_4$ | $\gamma_5$ |
|---|---|---|---|---|
| $\gamma_1$ | 0.6 | | 0.88 | |
| $\gamma_2$ | | 0.9 | | |
| $\gamma_3$ | | | | 0.95 |
| $\gamma_4$ | | | | |

$$\gamma_{total} = \gamma_1 + \sum_{i=2}^{j} \left(1 - V_{NN(i),i}\right) \gamma_i$$

**Extended Data Fig. 1 | Computing global diversity from diversity maps.** For each time interval or regional diversity map, (1) we plot hundreds of transects (white lines) from diversity peaks to their nearest troughs (Supplementary Video 7 for an example), (2) we integrate diversity along the transects (from $\alpha_{0'}$ to $\alpha_L$) according to the distance between pairs of grids using the overlap coefficient (V), which gives the proportion of shared genera with respect to the grid with the least diversity, i.e. $V_{n,n+1} \times \min(\alpha_n; \alpha_{n+1})$ [**see** Supplementary Note 1 **in SI**], (3) we order the resulting transects' diversity ($\gamma_i$) from maximum diversity ($\gamma_{max}$) to minimum diversity ($\gamma_{min}$), calculate the pair-wise distance between transects (the distance between their peaks), and integrate the diversity of transects from the greatest to lowest according to the nearest-neighbour distance of the corresponding transect to those transects already integrated ($\gamma_{total}$).

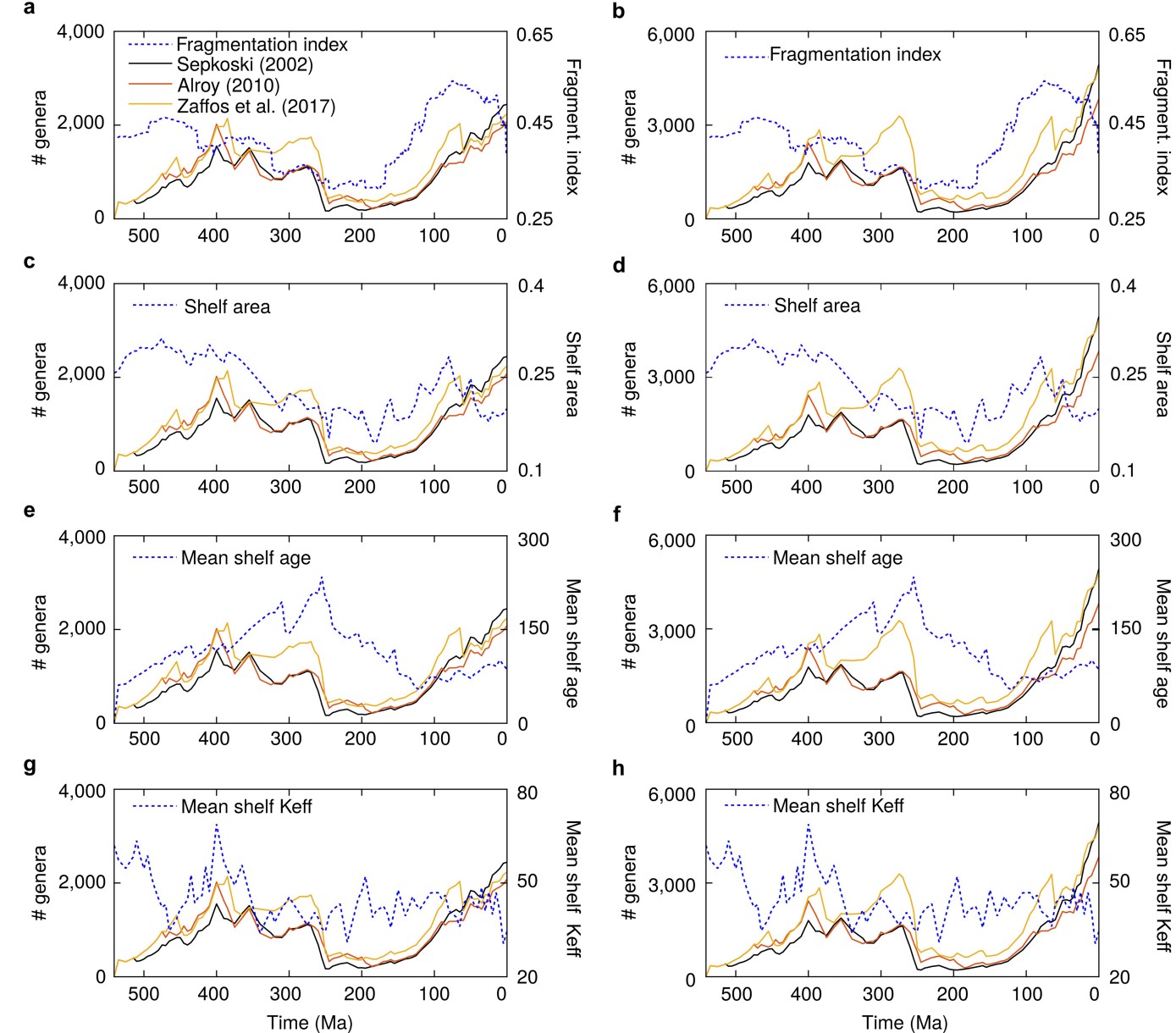

**Extended Data Fig. 2 | Global diversity dynamics and continental configuration.** Global diversity dynamics (# genera) reconstructed by using the calibrated logistic model (**a**, **c**, **e**, **g**) and the exponential model (**b**, **d**, **f**, **h**) and overimposed on the temporal variability of the fragmentation index **a**, (**a**, **b**), global mean shelf area (**c**, **d**), mean shelf age (**e**, **f**) and mean shelf effective carrying capacity ($K_{eff}$) (**g**, **h**).

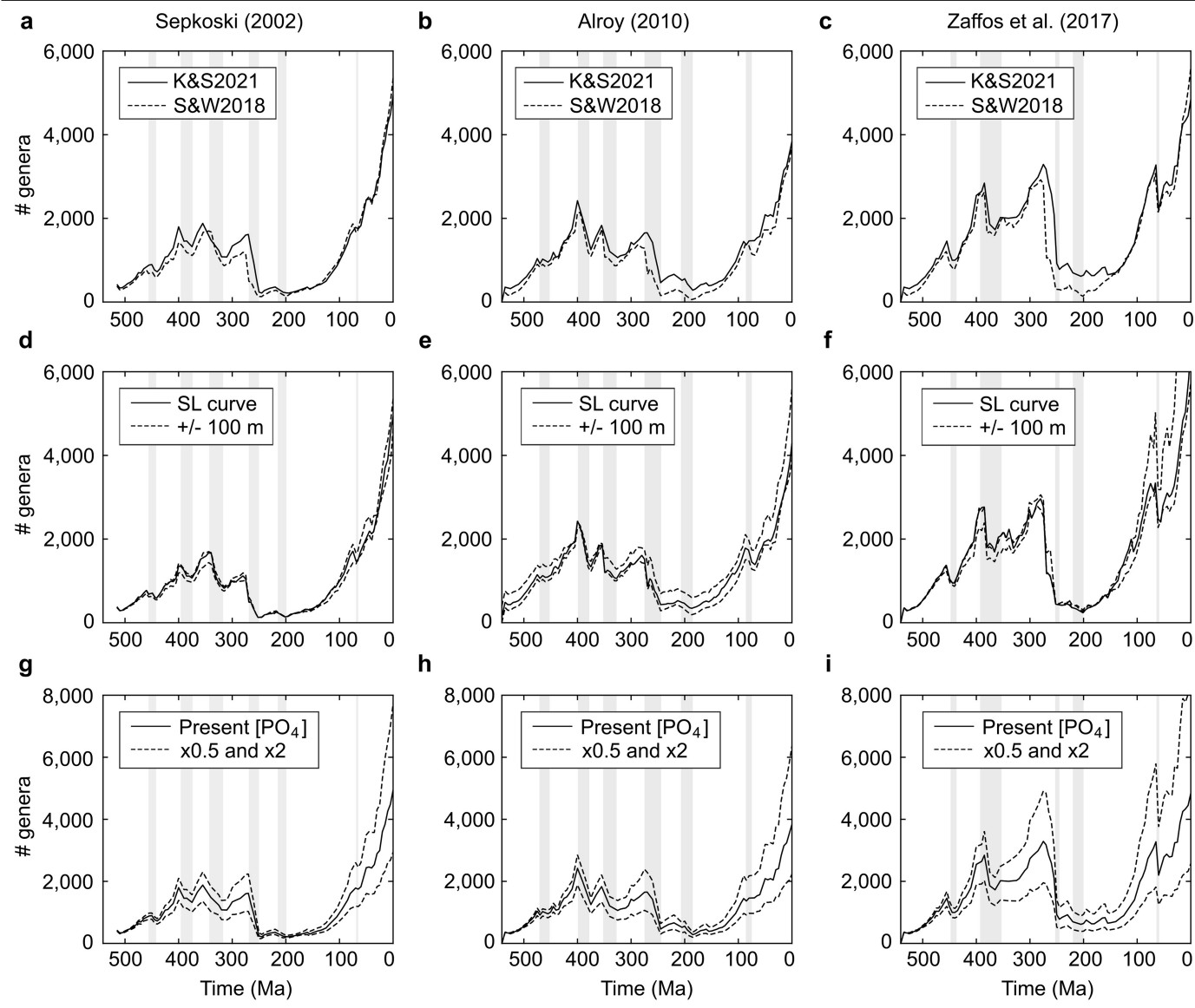

**Extended Data Fig. 3 | Sensitivity analysis. a-c**, Global diversity dynamics reconstructed by the exponential diversification model using two alternative palaeogeographic models[33,35]. **d-f**, Effect of changing the sea level +/− 100 m compared to the original palaeo-digital elevation model of Scotese and Wright[33]. **g-i**, Effect of changing the mean ocean phosphate concentration (x2 and x0.5 with respect to present day concentrations, i.e., default scenario). All simulations were run for each of the three mass extinction patterns, that is, Sepkoski[20] (**a**, **d**, **g**), Alroy[21] (**b**, **e**, **h**), and Zaffos et al[22]. (**c**, **f**, **i**).

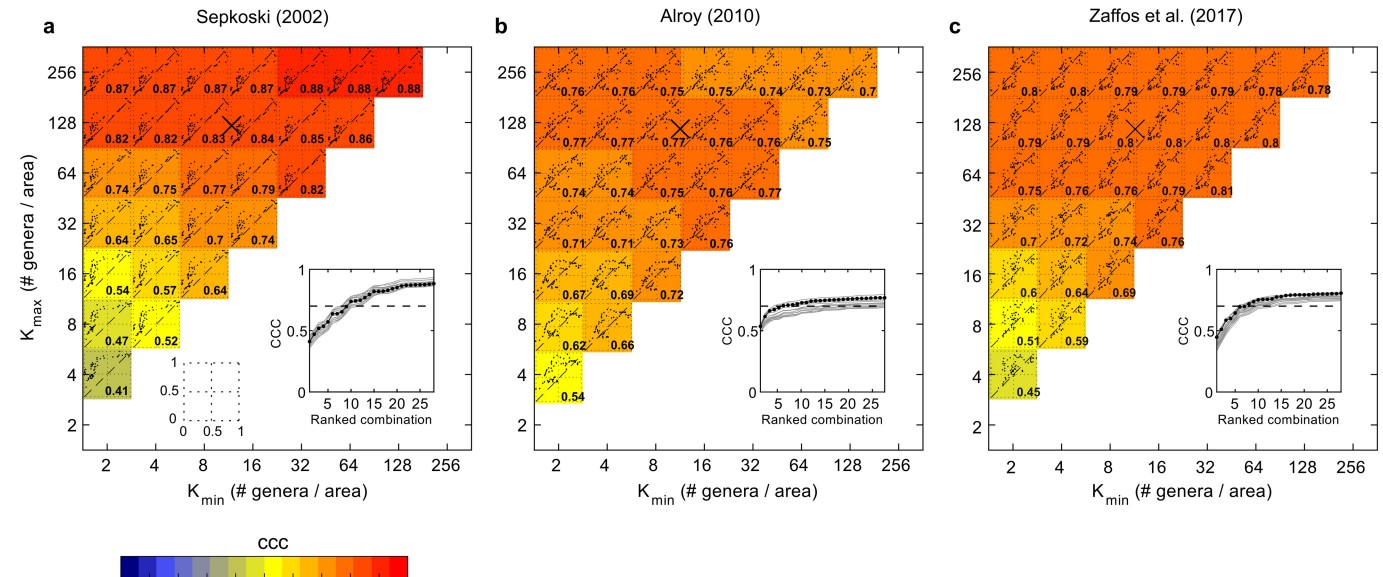

**Extended Data Fig. 4 | Calibrating the logistic model's carrying capacities.**
**a-c**, Lin's concordance correlation coefficient (CCC) for the relationship
between the global diversities resulting from the model and the fossil diversity
estimates of Sepkoski[20] (n = 66 data points) (**a**), Alroy[21] (n = 60) (**b**), and Zaffos
et al[22] (n = 69) (**c**) using different combinations of $K_{min}$ and $K_{max}$ in the model.
Only the time series data between the end of one mass extinction and the
beginning of the next were included in the analysis .Superimposed on each
combination of $K_{min}$ and $K_{max}$ are the scatterplots of the normalized (0-1)
diversity data (model vs fossil). The inset in each panel shows the CCCs in

ascending order for the different combinations of $K_{min}$ and $K_{max}$. The black curve
in the insets is for the simulation run using the selected parameters [i.e.,
$Q_{10} = 1.75$, $K_{food} = 0.5$ molC m$^{-2}$y$^{-1}$, net diversification rate limits ($\rho_{min} - \rho_{max}$) =
0.001-0.035 Myr$^{-1}$]. The grey curves are for each of the first 15 combinations of
parameters listed in Extended Data Table 2. The dashed line denotes the CCC
value of 0.7 and the cross in each panel is the average of all $K_{min}$ and $K_{max}$
combinations giving a CCC greater than 0.7. These are the values of $K_{min}$ and
$K_{max}$ used to run the calibrated logistic model.

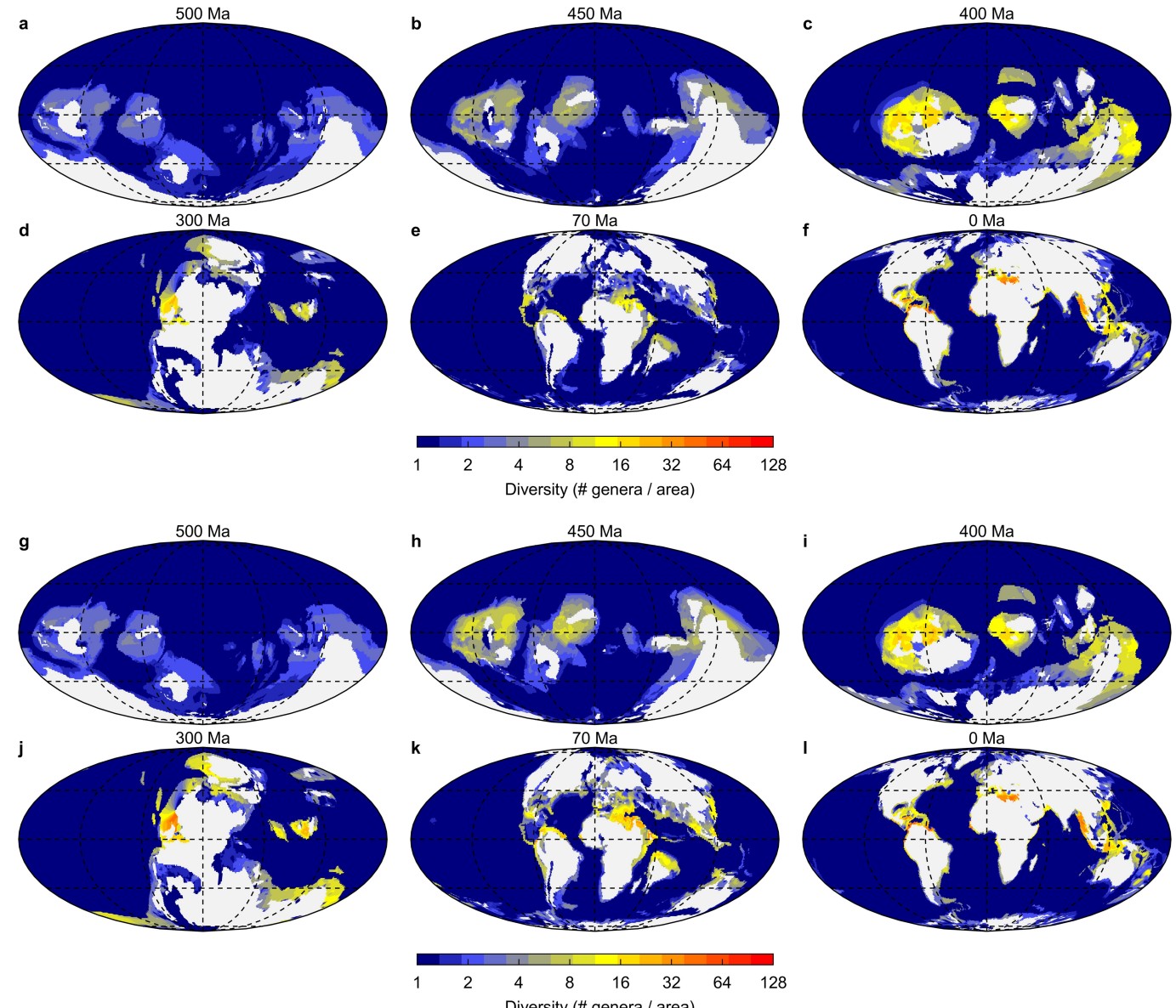

**Extended Data Fig. 5 | Re-diversifying the Phanerozoic oceans. a-f**, Spatial distribution of marine animal diversity (# genera/area) in the Cambrian (Guzhangian, 500 Ma), Late Ordovician (Katian, 450 Ma), Early Devonian (Emsian, 400 Ma), Late Carboniferous (Pennsylvanian, 300 Ma), Late Cretaceous (Maastrichtian, 70 Ma) and present generated by the calibrated logistic model after imposing the pattern of mass extinctions extracted from the fossil diversity curve of Alroy[21]. **g-l**, as in a-f but for the pattern of mass extinctions extracted from the fossil diversity curve of Zaffos et al[22]. These model runs use the following parameters: $Q_{10} = 1.75$, $K_{food} = 0.5$ molC m$^{-2}$y$^{-1}$, net diversification rate limits ($\rho_{min} - \rho_{max}$) = 0.001-0.035 Myr$^{-1}$ (per capita), and a $K_{min}$ to $K_{max}$ range between 11 and 119 genera per unit area for Alroy[21] and between 11 and 117 genera for Zaffos et al[22]. See also Supplementary Video 3 for the full Phanerozoic sequences.

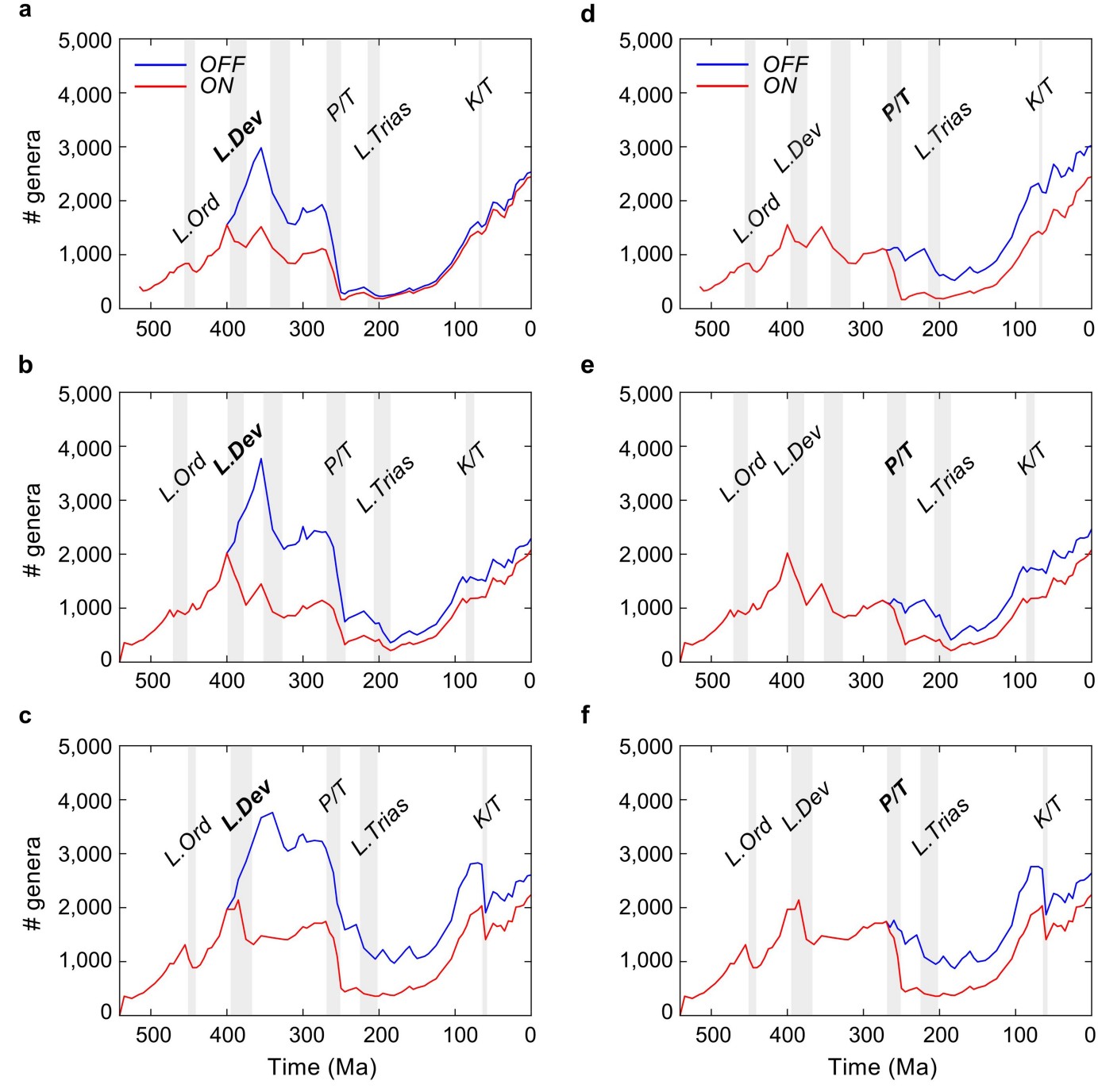

**Extended Data Fig. 6 | Testing the effect of mass extinctions.** Global diversity dynamics (# genera) reconstructed by the calibrated logistic model after disabling (blue) and enabling (red) the Late Devonian (**a-c**) and Permo-Triassic (**d-f**) mass extinctions from the mass extinction pattern extracted from the fossil diversity curve of Sepkoski[20] (**a**, **d**), Alroy[21] (**a**, **e**) and Zaffos et al[22] (**a**, **f**), respectively.

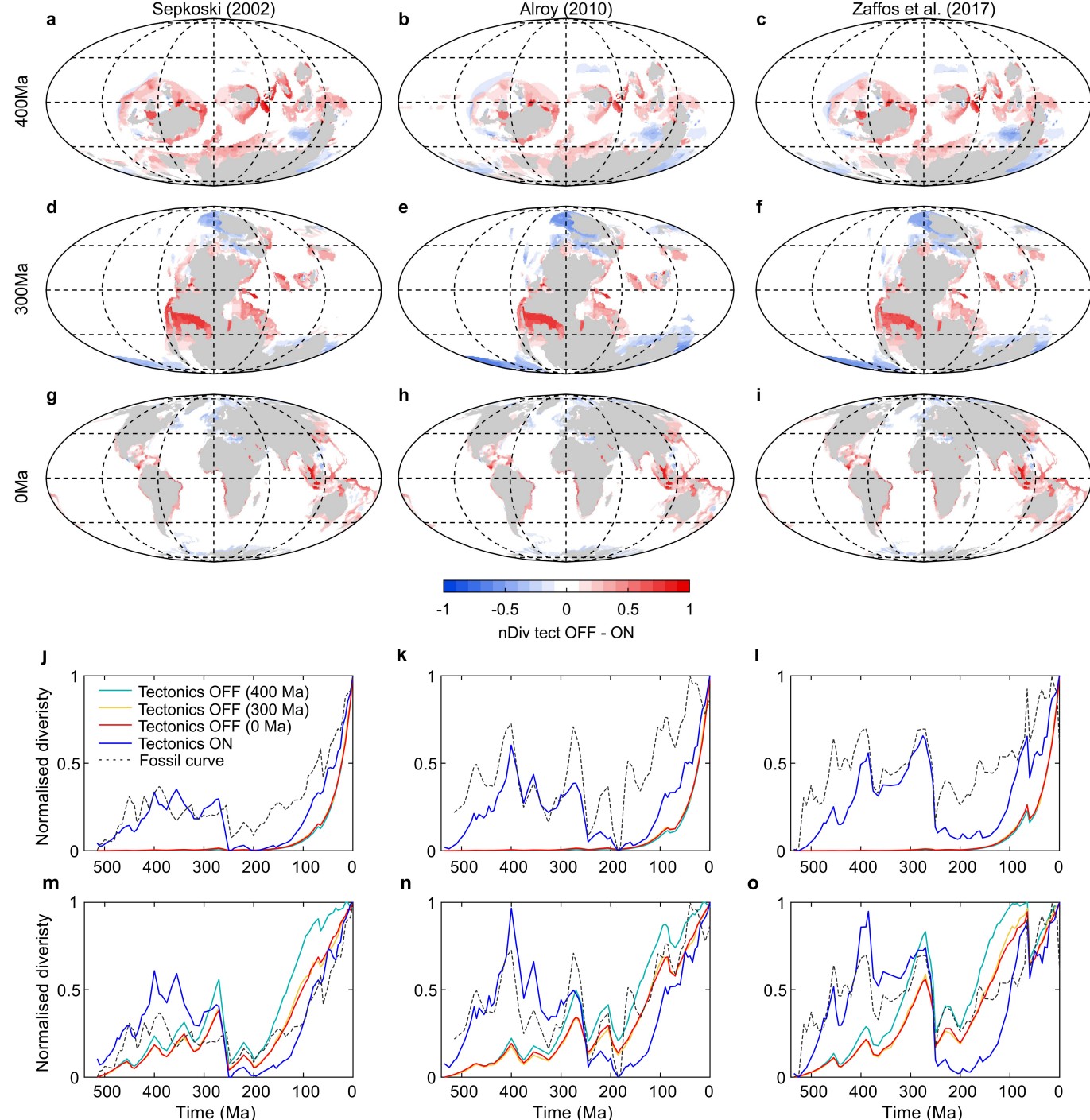

**Extended Data Fig. 7 | Testing the effect of plate tectonics. a-i**, Comparison of the results of the calibrated logistic model without plate tectonics and with plate tectonics. The color code represents the difference between the log-transformed normalized diversities (0-1) produced by the model with static palaeogeography (nDiv tectonics OFF) and the model with variable palaeogeography (nDiv tectonics ON) for three time frames (panels row-wise 400 Ma, 300 Ma and 0 Ma) and three mass extinction patterns (panels column-wise Sepkoski[20], Alroy[21] and Zaffos et al[22]). **j-l**, Global diversity dynamics produced by the exponential diversification model with static palaeogeography (light blue, yellow and red colour lines correspond to palaeogeographic configurations of 400 Ma, 300 Ma and 0 Ma, respectively) and with variable palaeogeography (blue line) for each of the three mass extinction patterns (panels column-wise Sepkoski[20], Alroy[21] and Zaffos et al[22]). The corresponding fossil diversity curve is superimposed on each panel (grey dashed line). **m-o**, As in panels j-l but for the calibrated logistic model.

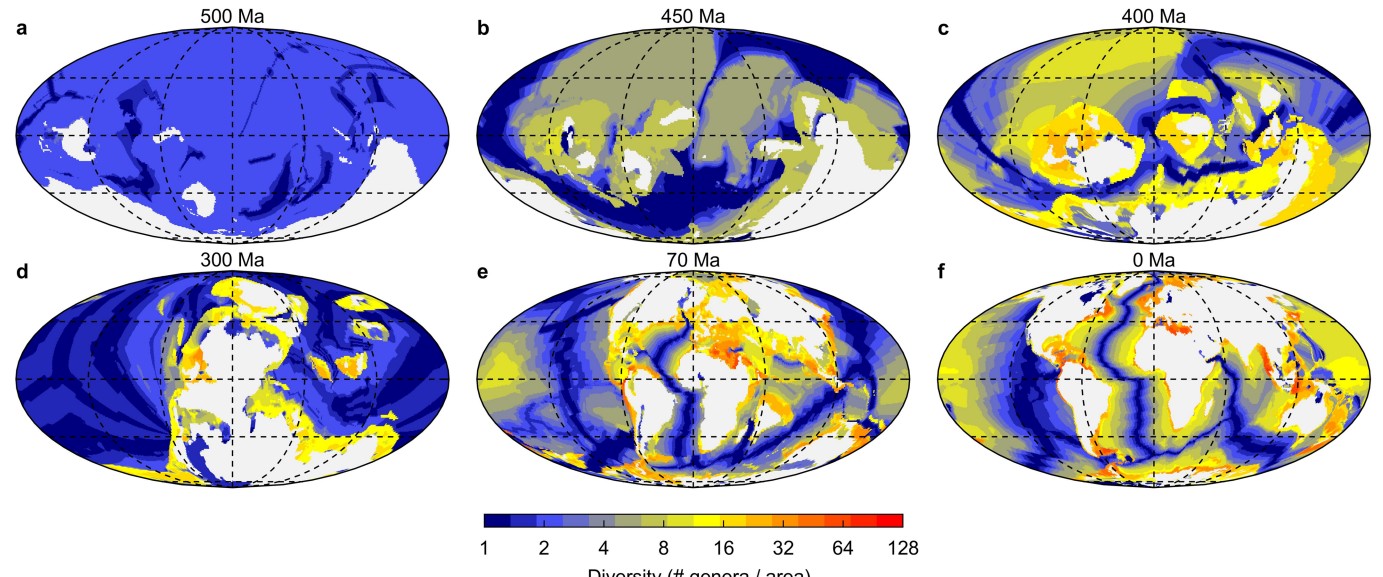

**Extended Data Fig. 8 | Testing the effect of environmental forcings.** Spatial distribution of marine animal diversity (# genera / area) in the Cambrian (Guzhangian, 500 Ma), Late Ordovician (Katian, 450 Ma), Early Devonian (Emsian, 400 Ma), Late Carboniferous (Pennsylvanian, 300 Ma), Late Cretaceous (Maastrichtian, 70 Ma) and present generated by the calibrated logistic model after imposing the pattern of mass extinctions extracted from the fossil diversity curve of Sepkoski[20]. This model run uses the following parameters: $Q_{10} = 1$ (no temperature dependence), $K_{food} = 0$ molC m$^{-2}$y$^{-1}$ (no food dependence), net diversification rate limits ($\rho_{min} - \rho_{max}$) = 0.001-0.035 Myr$^{-1}$, and a $K_{min}$ to $K_{max}$ range between 12 and 123 genera.

**Extended Data Table 1 | Model parameters and range of parameter values tested**

| Symbol | Description | Value | Range tested | units |
|---|---|---|---|---|
| $\rho_{min}$ | Minimum net diversification rate | 0.001 | -- | $Myr^{-1}$ |
| $\rho_{max}$ | Maximum net diversification rate | 0.035 | $0.03 - 0.04$ | $Myr^{-1}$ |
| $Q_{10}$ | Thermal sensitivity | 1.75 | $1.5 - 2.5$ | n.u. |
| $K_{food}$ | Half-saturation constant for food | 0.5 | $0.25 - 1$ | $mol\ m^{-2}\ yr^{-1}$ |
| lat-lon | Radius of search for immigration | 278 | $0 - 278$ | Km |

**Extended Data Table 2 | Lin's Concordance Correlation Coefficient (CCC) using different parameter settings in the model**

| Sepkoski (2002) | | | | | Alroy (2010) | | | | | Zaffos et al. (2017) | | | | |
|---|---|---|---|---|---|---|---|---|---|---|---|---|---|---|
| Model parameters | | | Lin's CCC | | Model parameters | | | Lin's CCC | | Model parameters | | | Lin's CCC | |
| Q10 | Kfood | ρmax | Exp | Log | Q10 | Kfood | ρmax | Exp | Log | Q10 | Kfood | ρmax | Exp | Log |
| 1.5 | 1 | 0.035 | 0.91 | 0.60 | 1.5 | 1 | 0.035 | 0.73 | 0.64 | 1.5 | 1 | 0.035 | 0.78 | 0.64 |
| 1.5 | 0.75 | 0.035 | 0.91 | 0.61 | 1.5 | 0.75 | 0.035 | 0.71 | 0.66 | 1.75 | 1 | 0.035 | 0.77 | 0.60 |
| 1.75 | 1 | 0.035 | 0.91 | 0.55 | 1.75 | 1 | 0.035 | 0.70 | 0.60 | 1.75 | 0.75 | 0.035 | 0.76 | 0.63 |
| 1.75 | 0.75 | 0.035 | 0.91 | 0.56 | 1.75 | 0.75 | 0.035 | 0.70 | 0.62 | 2 | 1 | 0.035 | 0.75 | 0.57 |
| 2 | 0.75 | 0.035 | 0.91 | 0.53 | 2 | 0.75 | 0.035 | 0.70 | 0.60 | 2 | 0.75 | 0.035 | 0.75 | 0.59 |
| 2 | 1 | 0.035 | 0.91 | 0.52 | 1.5 | 1 | 0.04 | 0.68 | 0.64 | 1.5 | 0.75 | 0.035 | 0.75 | 0.66 |
| 2.25 | 0.75 | 0.035 | 0.91 | 0.51 | 2 | 1 | 0.035 | 0.68 | 0.58 | 2.25 | 1 | 0.035 | 0.74 | 0.55 |
| 2.5 | 0.75 | 0.035 | 0.91 | 0.49 | 1.5 | 0.5 | 0.035 | 0.68 | 0.69 | 2.25 | 0.75 | 0.035 | 0.74 | 0.57 |
| 2.25 | 1 | 0.035 | 0.90 | 0.50 | **1.75** | **0.5** | **0.035** | **0.68** | **0.65** | 2.5 | 1 | 0.035 | 0.74 | 0.53 |
| 2.5 | 1 | 0.035 | 0.90 | 0.49 | 2 | 0.5 | 0.035 | 0.67 | 0.62 | 2.5 | 0.75 | 0.035 | 0.73 | 0.56 |
| 1.5 | 0.5 | 0.035 | 0.90 | 0.61 | 2.25 | 0.75 | 0.035 | 0.68 | 0.60 | **1.75** | **0.5** | **0.035** | **0.72** | **0.63** |
| **1.75** | **0.5** | **0.035** | **0.90** | **0.57** | 2.5 | 0.75 | 0.035 | 0.68 | 0.59 | 2.5 | 0.5 | 0.035 | 0.72 | 0.58 |
| 2.5 | 0.5 | 0.035 | 0.90 | 0.50 | 2.5 | 1 | 0.035 | 0.68 | 0.56 | 2.25 | 0.5 | 0.035 | 0.72 | 0.59 |
| 1.5 | 1 | 0.04 | 0.90 | 0.59 | 2 | 1 | 0.04 | 0.67 | 0.58 | 1.75 | 1 | 0.04 | 0.72 | 0.61 |
| 2.25 | 0.5 | 0.035 | 0.90 | 0.52 | 2.25 | 1 | 0.035 | 0.66 | 0.57 | 1.5 | 1 | 0.04 | 0.72 | 0.65 |
| 2 | 0.5 | 0.035 | 0.90 | 0.54 | 2.5 | 1 | 0.04 | 0.66 | 0.56 | 2 | 0.5 | 0.035 | 0.71 | 0.61 |
| 2 | 1 | 0.04 | 0.89 | 0.51 | 1.75 | 1 | 0.04 | 0.66 | 0.60 | 2 | 1 | 0.04 | 0.71 | 0.58 |
| 2.5 | 1 | 0.04 | 0.89 | 0.49 | 2.5 | 0.5 | 0.035 | 0.65 | 0.62 | 2.25 | 1 | 0.04 | 0.70 | 0.56 |
| 1.75 | 1 | 0.04 | 0.89 | 0.55 | 2.25 | 1 | 0.04 | 0.65 | 0.57 | 2.5 | 1 | 0.04 | 0.70 | 0.55 |
| 2.25 | 1 | 0.04 | 0.89 | 0.50 | 2.25 | 0.5 | 0.035 | 0.65 | 0.61 | 1.5 | 0.5 | 0.035 | 0.69 | 0.67 |
| 1.5 | 0.75 | 0.04 | 0.88 | 0.59 | 1.5 | 0.75 | 0.04 | 0.64 | 0.66 | 1.75 | 0.75 | 0.04 | 0.68 | 0.62 |
| 1.75 | 0.75 | 0.04 | 0.88 | 0.54 | 1.75 | 0.75 | 0.04 | 0.64 | 0.62 | 2.5 | 0.75 | 0.04 | 0.67 | 0.56 |
| 1.75 | 0.25 | 0.035 | 0.88 | 0.57 | 2 | 0.75 | 0.04 | 0.63 | 0.59 | 2 | 0.75 | 0.04 | 0.67 | 0.59 |
| 1.5 | 0.25 | 0.035 | 0.87 | 0.60 | 2 | 0.25 | 0.035 | 0.62 | 0.65 | 2.25 | 0.75 | 0.04 | 0.67 | 0.57 |
| 2 | 0.75 | 0.04 | 0.87 | 0.51 | 1.75 | 0.25 | 0.035 | 0.62 | 0.67 | 2.5 | 0.25 | 0.035 | 0.66 | 0.60 |
| 2.25 | 0.75 | 0.04 | 0.87 | 0.51 | 2.5 | 0.25 | 0.035 | 0.62 | 0.63 | 1.5 | 0.75 | 0.04 | 0.65 | 0.67 |
| 2 | 0.25 | 0.035 | 0.87 | 0.54 | 2.25 | 0.75 | 0.04 | 0.61 | 0.57 | 2.25 | 0.25 | 0.035 | 0.65 | 0.62 |
| 2.5 | 0.75 | 0.04 | 0.87 | 0.50 | 2.5 | 0.75 | 0.04 | 0.60 | 0.58 | 2 | 0.25 | 0.035 | 0.64 | 0.63 |
| 2.5 | 0.25 | 0.035 | 0.87 | 0.51 | 2.25 | 0.25 | 0.035 | 0.60 | 0.65 | 2.5 | 0.5 | 0.04 | 0.63 | 0.57 |
| 2.25 | 0.25 | 0.035 | 0.87 | 0.53 | 1.5 | 0.25 | 0.035 | 0.60 | 0.69 | 2.25 | 0.5 | 0.04 | 0.62 | 0.59 |
| 1.5 | 0.5 | 0.04 | 0.86 | 0.58 | 1.75 | 0.5 | 0.04 | 0.59 | 0.65 | 1.75 | 0.25 | 0.035 | 0.62 | 0.65 |
| 1.75 | 0.5 | 0.04 | 0.85 | 0.55 | 2 | 0.5 | 0.04 | 0.59 | 0.62 | 2 | 0.5 | 0.04 | 0.62 | 0.60 |
| 2 | 0.5 | 0.04 | 0.85 | 0.52 | 1.5 | 0.5 | 0.04 | 0.58 | 0.68 | 1.75 | 0.5 | 0.04 | 0.61 | 0.63 |
| 2.5 | 0.5 | 0.04 | 0.84 | 0.50 | 2.5 | 0.5 | 0.04 | 0.57 | 0.60 | 1.5 | 0.25 | 0.035 | 0.58 | 0.68 |
| 2.25 | 0.5 | 0.04 | 0.84 | 0.50 | 2.25 | 0.5 | 0.04 | 0.56 | 0.60 | 1.5 | 0.5 | 0.04 | 0.57 | 0.67 |
| 1.5 | 0.25 | 0.04 | 0.83 | 0.59 | 2 | 0.25 | 0.04 | 0.52 | 0.64 | 2.5 | 0.25 | 0.04 | 0.54 | 0.59 |
| 2.25 | 0.25 | 0.04 | 0.82 | 0.51 | 1.5 | 0.25 | 0.04 | 0.51 | 0.69 | 2.25 | 0.25 | 0.04 | 0.53 | 0.6 |
| 1.75 | 0.25 | 0.04 | 0.82 | 0.55 | 1.75 | 0.25 | 0.04 | 0.51 | 0.66 | 2 | 0.25 | 0.04 | 0.51 | 0.62 |
| 2 | 0.25 | 0.04 | 0.82 | 0.53 | 2.5 | 0.25 | 0.04 | 0.51 | 0.62 | 1.75 | 0.25 | 0.04 | 0.51 | 0.64 |
| 2.5 | 0.25 | 0.04 | 0.81 | 0.5 | 2.25 | 0.25 | 0.04 | 0.5 | 0.63 | 1.5 | 0.25 | 0.04 | 0.47 | 0.68 |

The CCCs are for the relationship between the normalized diversities estimated from the fossil record and those generated by the exponential (Exp) and the logistic (Log) models. The 15 combinations of model parameters that gave the highest CCC for each mass extinction pattern were selected. Of these, the combination that gave the highest CCC for the relationship between the fossil diversities and the diversities generated by the calibrated logistic model was selected as the best (marked in bold) and used by default to run the model.

# Reporting Summary

## Statistics

For all statistical analyses, confirm that the following items are present in the figure legend, table legend, main text, or Methods section.

| n/a | Confirmed | |
|---|---|---|
| ☐ | ☒ | The exact sample size (*n*) for each experimental group/condition, given as a discrete number and unit of measurement |
| ☐ | ☒ | A statement on whether measurements were taken from distinct samples or whether the same sample was measured repeatedly |
| ☒ | ☐ | The statistical test(s) used AND whether they are one- or two-sided <br> *Only common tests should be described solely by name; describe more complex techniques in the Methods section.* |
| ☒ | ☐ | A description of all covariates tested |
| ☐ | ☒ | A description of any assumptions or corrections, such as tests of normality and adjustment for multiple comparisons |
| ☐ | ☒ | A full description of the statistical parameters including central tendency (e.g. means) or other basic estimates (e.g. regression coefficient) AND variation (e.g. standard deviation) or associated estimates of uncertainty (e.g. confidence intervals) |
| ☒ | ☐ | For null hypothesis testing, the test statistic (e.g. *F*, *t*, *r*) with confidence intervals, effect sizes, degrees of freedom and *P* value noted <br> *Give P values as exact values whenever suitable.* |
| ☒ | ☐ | For Bayesian analysis, information on the choice of priors and Markov chain Monte Carlo settings |
| ☒ | ☐ | For hierarchical and complex designs, identification of the appropriate level for tests and full reporting of outcomes |
| ☒ | ☐ | Estimates of effect sizes (e.g. Cohen's *d*, Pearson's *r*), indicating how they were calculated |

*Our web collection on statistics for biologists contains articles on many of the points above.*

## Software and code

Policy information about availability of computer code

| | |
|---|---|
| Data collection | The coupled paleogeographic-diversification model presented here uses input data of seafloor age distributions and paleoenvironmental conditions from the siwill22/agegrid-0.1 v1-alpha paleogeographic model (DOI: 10.5281/zenodo.3271360) and the cGENIE Earth System Model (DOI: 10.5281/zenodo.4618023), respectively. Additionally, all data inputs required to run the coupled paleogeographic-diversification model are available along with the model code on GitHub (https://github.com/CarmenGarciaComas/INDITEK). |
| Data analysis | The model is written in MATLAB 2013b and tested with MATLAB 2021a in a MacOS 2.3 GHz 8-Core Intel Core i9, and with MATLAB 2020b on Windows with a 2.5 GHz Intel i5-3210M and on Linux Debian with a 2.6 GHz Intel Core 9th Gen i9-9980HK processor. The code for the coupled palaeogeographic-diversification model is assigned a DOI: 10.5281/zenodo.6535496. The code used to reconstruct seafloor age distributions from GPlates full-plate tectonic reconstructions is assigned a DOI: 10.5281/zenodo.3271360. The code for the version of the 'muffin' release of the cGENIE Earth System Model used in this study, is tagged as v0.9.20, and is assigned a DOI: 10.5281/zenodo.4618023. |

For manuscripts utilizing custom algorithms or software that are central to the research but not yet described in published literature, software must be made available to editors and reviewers. We strongly encourage code deposition in a community repository (e.g. GitHub). See the Nature Portfolio guidelines for submitting code & software for further information.

## Data

Policy information about availability of data

All manuscripts must include a data availability statement. This statement should provide the following information, where applicable:

- Accession codes, unique identifiers, or web links for publicly available datasets
- A description of any restrictions on data availability
- For clinical datasets or third party data, please ensure that the statement adheres to our policy

To test the model, we digitized fossil diversity data from 2 published articles [Alroy J. Science 329, 1191–1194 (2010); Zaffos et al. PNAS 114, 5653–5658 (2017)] using the free software XYscan under the terms of the GNU General Public License as published by the Free Software Foundation. Copyright 2002-2021 Thomas S. Ullrich (https://rhig.physics.yale.edu/~ullrich/software/xyscan/). In addition, the Sepkoski's fossil diversity data were downloaded from the Sepkoski's Online Genus Database at the following link: http://strata.geology.wisc.edu/jack/. All these data are supplied as Source_Data_File1_FossilTimeSeries.xlsx through the Nature online submission system. These data were also used to calculate the magnitude, time and duration of mass extinctions (net diversification rates). The negative net diversification rates during mass extinctions are supplied as Source_Data_File2_mass_extinctions.xlsx files. Finally, modern estimates of marine invertebrate diversity were obtained from occurrence records of genera belonging to two of the most diverse groups of marine invertebrates, crustaceans and molluscs, as downloaded from the Ocean Biodiversity Information System (OBIS) database on 22nd October 2021 (www.obis.org). These diversity data are suplied with this paper as Source_Data_File3_OBIS_data.xlsx file.

The coupled paleogeographic-diversification model presented here uses input data of seafloor age distributions and paleoenvironmental conditions from the siwill22/agegrid-0.1 v1-alpha paleogeographic model and the cGENIE Earth System Model, respectively. We provide code availability for each of these two models. Additionally, all data inputs required to run the coupled paleogeographic-diversification model are available along with the model code on GitHub (https://github.com/CarmenGarciaComas/INDITEK).

# Field-specific reporting

Please select the one below that is the best fit for your research. If you are not sure, read the appropriate sections before making your selection.

☐ Life sciences        ☐ Behavioural & social sciences        ☒ Ecological, evolutionary & environmental sciences

For a reference copy of the document with all sections, see nature.com/documents/nr-reporting-summary-flat.pdf

# Life sciences study design

All studies must disclose on these points even when the disclosure is negative.

| | |
|---|---|
| Sample size | Describe how sample size was determined, detailing any statistical methods used to predetermine sample size OR if no sample-size calculation was performed, describe how sample sizes were chosen and provide a rationale for why these sample sizes are sufficient. |
| Data exclusions | Describe any data exclusions. If no data were excluded from the analyses, state so OR if data were excluded, describe the exclusions and the rationale behind them, indicating whether exclusion criteria were pre-established. |
| Replication | Describe the measures taken to verify the reproducibility of the experimental findings. If all attempts at replication were successful, confirm this OR if there are any findings that were not replicated or cannot be reproduced, note this and describe why. |
| Randomization | Describe how samples/organisms/participants were allocated into experimental groups. If allocation was not random, describe how covariates were controlled OR if this is not relevant to your study, explain why. |
| Blinding | Describe whether the investigators were blinded to group allocation during data collection and/or analysis. If blinding was not possible, describe why OR explain why blinding was not relevant to your study. |

# Behavioural & social sciences study design

All studies must disclose on these points even when the disclosure is negative.

| | |
|---|---|
| Study description | Briefly describe the study type including whether data are quantitative, qualitative, or mixed-methods (e.g. qualitative cross-sectional, quantitative experimental, mixed-methods case study). |
| Research sample | State the research sample (e.g. Harvard university undergraduates, villagers in rural India) and provide relevant demographic information (e.g. age, sex) and indicate whether the sample is representative. Provide a rationale for the study sample chosen. For studies involving existing datasets, please describe the dataset and source. |
| Sampling strategy | Describe the sampling procedure (e.g. random, snowball, stratified, convenience). Describe the statistical methods that were used to predetermine sample size OR if no sample-size calculation was performed, describe how sample sizes were chosen and provide a rationale for why these sample sizes are sufficient. For qualitative data, please indicate whether data saturation was considered, and what criteria were used to decide that no further sampling was needed. |

| | |
|---|---|
| Data collection | *Provide details about the data collection procedure, including the instruments or devices used to record the data (e.g. pen and paper, computer, eye tracker, video or audio equipment) whether anyone was present besides the participant(s) and the researcher, and whether the researcher was blind to experimental condition and/or the study hypothesis during data collection.* |
| Timing | *Indicate the start and stop dates of data collection. If there is a gap between collection periods, state the dates for each sample cohort.* |
| Data exclusions | *If no data were excluded from the analyses, state so OR if data were excluded, provide the exact number of exclusions and the rationale behind them, indicating whether exclusion criteria were pre-established.* |
| Non-participation | *State how many participants dropped out/declined participation and the reason(s) given OR provide response rate OR state that no participants dropped out/declined participation.* |
| Randomization | *If participants were not allocated into experimental groups, state so OR describe how participants were allocated to groups, and if allocation was not random, describe how covariates were controlled.* |

# Ecological, evolutionary & environmental sciences study design

All studies must disclose on these points even when the disclosure is negative.

| | |
|---|---|
| Study description | We present a regional diversification model of marine animals for the Phanerozoic. The diversification model is coupled to a palaeogeographic model that constrains evolutionary time within regions (i.e. the age of the seafloor for deep ocean regions and the time underwater for flooded continental regions). An Earth System model provides palaeoenvironmental conditions (i.e. seawater temperature and organic C export fluxes, as a surrogate for food supply). The coupled model tracks the geographic position of ocean and flooded continental points every ca. 5 Myr, from 541 Ma to the present. Each and every one of the tracked points accumulate diversity over time at a different rate, which is modulated by the environmental history of each point. The model reproduces the present-day spatial distributions and fossil time-trajectories of marine invertebrate diversity after imposing mass extinctions. We find that the dynamics of global fossil diversity is best described by a diversification model that operates widely within the exponential growth regime of a logistic function. A spatially resolved analysis of the diversity-to-carrying capacity ratio reveals that <2% of the global flooded continental area throughout the Phanerozoic exhibits levels of diversity approaching ecological saturation. Our model corroborates earlier claims that Earth's environmental history and the patterns of continental fragmentation and reassembly have been major determinants of marine animal diversification. The analysis also shows that the development of diversity hotspots played a major role in the overall increase in global diversity during the late Mesozoic and Cenozoic. |
| Research sample | This is a modeling study. We test the model against three published fossil diversity curves. Fossil diversity curves are digitized from 2 published articles [Alroy J. Science 329, 1191–1194 (2010); Zaffos et al. PNAS 114, 5653–5658 (2017)] using the free software XYscan under the terms of the GNU General Public License as published by the Free Software Foundation. Copyright 2002-2021 Thomas S. Ullrich. In addition, the Sepkoski's fossil diversity data are downloaded from the Sepkoski's Online Genus Database at the following link: http://strata.geology.wisc.edu/jack/. Modern estimates of marine invertebrate diversity are downloaded from the Ocean Biodiversity Information System (OBIS) database on 22nd October 2021 (www.obis.org). |
| Sampling strategy | N/A - There is no sampling strategy as it is a model. |
| Data collection | The coupled paleogeographic-diversification model presented here uses input data of seafloor age distributions and paleoenvironmental conditions from the siwill22/agegrid-0.1 v1-alpha paleogeographic model (DOI: 10.5281/zenodo.3271360) and the cGENIE Earth System Model (DOI: 10.5281/zenodo.4618023), respectively. Additionally, all data inputs required to run the coupled paleogeographic-diversification model are available along with the model code on GitHub (https://github.com/CarmenGarciaComas/INDITEK).<br><br>To compare our data outputs with observations, we scanned published fossil curves and downloaded observations of taxa from a global dataset of marine diversity. Digitized fossil diversity data from the Alroy and Zaffos et al. published articles with the free software XYscan (https://rhig.physics.yale.edu/~ullrich/software/xyscan/) and those downloaded from the Sepkoski's Online Genus Database (http://strata.geology.wisc.edu/jack/) are provided as Source data with this paper: Source_Data_File1_FossilTimeSeries.xlsx. Diversity data downloaded from the Ocean Biodiversity Information System (OBIS) database on 22nd October 2021 (www.obis.org) are also provided as Source data with this paper: Source_Data_File3_OBIS_data.xlsx. |
| Timing and spatial scale | This is a modeling study and therefore there is no data collection. The model analysis covers the global ocean for the last 541 My. The palaeogeographic model mimics plate tectonics and traces the age and position of thousands of seafloor points (for the deep ocean and for the flooded continental regions) in 82 time frames about every 5 My (exact time frames: 541 535 525 515 510 505 500 495 490 485 480 475 470 465 455 450 445 440 435 430 425 420 415 410 400 390 385 375 365 355 340 325 320 310 305 300 295 285 275 270 265 260 255 250 245 240 230 220 205 200 195 185 180 170 165 160 155 150 140 135 130 125 120 105 95 90 85 80 70 65 60 50 45 40 35 30 25 20 15 10 5 0 Mya). Time frames were selected as representative of tectonic changes around those dates with computation limitation. The Earth system model provides paleoenvironmental conditions (seawater temperature and deep-time food supply) with a spatio-temporal resolution of 36x36 equal-area grids every 20 My (exact time frames: 541 530 520 510 496 436 415 400 366 327 301 265 234 196 178 168 149 131 116 107 97 87 75 69 61 52 36 26 15 0 Mya). Time frames and spatial resolution were selected as representative of tectonic changes around those dates with computation limitation. The paleoenvironmental conditions were interpolated into the seafloor points for the 82 time frames in a 2-step fashion (see Methods for details). Comparison of model outputs with fossil curves was done for the 82 time frames by scanning the fossil time series. Spatial comparison of 0 Mya with current diversity was done for 0.5x0.5 grid maps and mean latitudinal values of OBIS data after correcting for sampling effort (see Methods for details). |

| Data exclusions | This is a modeling study and we only use data (i.e. fossil diversity curves and observations of marine invertebrates extracted from the OBIS database) to test the global diversity time series and modern spatial diversity distributions generated by the model. To calculate Lin's concordance correlation coefficient, which provides an estimate of how much the global diversity curves in our model differ from the fossil diversity curves, we exclude data points that are within mass extinction intervals. The reasoning is that mass extinctions are imposed externally on the model and, therefore, leaving them would inflate the statistical fits. For the comparison with OBIS data, we focus on continental margins, which harbour the vast majority of the diversity of marine invertebrates. |
|---|---|
| Reproducibility | This is a modeling study. All the analysis and results can be replicated using the model code that is available on GitHub (https://github.com/CarmenGarciaComas/INDITEK). The results are consistently repeated using 40 different combinations of model parameters. Furthermore, we impose extinction patterns extracted from 3 different (published) fossil diversity curves and test the model against all of them. In addition, we test the robustness of the model against various configurations of the paleogeographic model and the earth system by conducting a series of sensitivity analyses. Sensitivities to changes in the palaeogeographic reconstruction models, sea level, or the oceanic inventory of phosphorus over the past 541 million years. In all cases, the results and conclusions of the study remain the same. |
| Randomization | N/A as data were not issued from a sampling strategy. This is a modeling study and the results are generated from the simulations that result from running the model. |
| Blinding | N/A as data were not issued from a sampling strategy. |

Did the study involve field work? ☐ Yes ☒ No

## Field work, collection and transport

| Field conditions | N/A |
|---|---|
| Location | N/A |
| Access & import/export | N/A |
| Disturbance | N/A |

# Reporting for specific materials, systems and methods

We require information from authors about some types of materials, experimental systems and methods used in many studies. Here, indicate whether each material, system or method listed is relevant to your study. If you are not sure if a list item applies to your research, read the appropriate section before selecting a response.

## Materials & experimental systems

| n/a | Involved in the study |
|---|---|
| ☒ | ☐ Antibodies |
| ☒ | ☐ Eukaryotic cell lines |
| ☒ | ☐ Palaeontology and archaeology |
| ☒ | ☐ Animals and other organisms |
| ☒ | ☐ Human research participants |
| ☒ | ☐ Clinical data |
| ☒ | ☐ Dual use research of concern |

## Methods

| n/a | Involved in the study |
|---|---|
| ☒ | ☐ ChIP-seq |
| ☒ | ☐ Flow cytometry |
| ☒ | ☐ MRI-based neuroimaging |

## Antibodies

| Antibodies used | *Describe all antibodies used in the study; as applicable, provide supplier name, catalog number, clone name, and lot number.* |
|---|---|
| Validation | *Describe the validation of each primary antibody for the species and application, noting any validation statements on the manufacturer's website, relevant citations, antibody profiles in online databases, or data provided in the manuscript.* |

## Eukaryotic cell lines

Policy information about cell lines

| Cell line source(s) | *State the source of each cell line used.* |
|---|---|
| Authentication | *Describe the authentication procedures for each cell line used OR declare that none of the cell lines used were authenticated.* |
| Mycoplasma contamination | *Confirm that all cell lines tested negative for mycoplasma contamination OR describe the results of the testing for mycoplasma contamination OR declare that the cell lines were not tested for mycoplasma contamination.* |

| Commonly misidentified lines<br>(See ICLAC register) | *Name any commonly misidentified cell lines used in the study and provide a rationale for their use.* |
|---|---|

## Palaeontology and Archaeology

| Specimen provenance | *Provide provenance information for specimens and describe permits that were obtained for the work (including the name of the issuing authority, the date of issue, and any identifying information). Permits should encompass collection and, where applicable, export.* |
|---|---|
| Specimen deposition | *Indicate where the specimens have been deposited to permit free access by other researchers.* |
| Dating methods | *If new dates are provided, describe how they were obtained (e.g. collection, storage, sample pretreatment and measurement), where they were obtained (i.e. lab name), the calibration program and the protocol for quality assurance OR state that no new dates are provided.* |

☐ Tick this box to confirm that the raw and calibrated dates are available in the paper or in Supplementary Information.

| Ethics oversight | *Identify the organization(s) that approved or provided guidance on the study protocol, OR state that no ethical approval or guidance was required and explain why not.* |
|---|---|

Note that full information on the approval of the study protocol must also be provided in the manuscript.

## Animals and other organisms

Policy information about studies involving animals; ARRIVE guidelines recommended for reporting animal research

| Laboratory animals | *For laboratory animals, report species, strain, sex and age OR state that the study did not involve laboratory animals.* |
|---|---|
| Wild animals | *Provide details on animals observed in or captured in the field; report species, sex and age where possible. Describe how animals were caught and transported and what happened to captive animals after the study (if killed, explain why and describe method; if released, say where and when) OR state that the study did not involve wild animals.* |
| Field-collected samples | *For laboratory work with field-collected samples, describe all relevant parameters such as housing, maintenance, temperature, photoperiod and end-of-experiment protocol OR state that the study did not involve samples collected from the field.* |
| Ethics oversight | *Identify the organization(s) that approved or provided guidance on the study protocol, OR state that no ethical approval or guidance was required and explain why not.* |

Note that full information on the approval of the study protocol must also be provided in the manuscript.

## Human research participants

Policy information about studies involving human research participants

| Population characteristics | *Describe the covariate-relevant population characteristics of the human research participants (e.g. age, gender, genotypic information, past and current diagnosis and treatment categories). If you filled out the behavioural & social sciences study design questions and have nothing to add here, write "See above."* |
|---|---|
| Recruitment | *Describe how participants were recruited. Outline any potential self-selection bias or other biases that may be present and how these are likely to impact results.* |
| Ethics oversight | *Identify the organization(s) that approved the study protocol.* |

Note that full information on the approval of the study protocol must also be provided in the manuscript.

## Clinical data

Policy information about clinical studies

All manuscripts should comply with the ICMJE guidelines for publication of clinical research and a completed CONSORT checklist must be included with all submissions.

| Clinical trial registration | *Provide the trial registration number from ClinicalTrials.gov or an equivalent agency.* |
|---|---|
| Study protocol | *Note where the full trial protocol can be accessed OR if not available, explain why.* |
| Data collection | *Describe the settings and locales of data collection, noting the time periods of recruitment and data collection.* |
| Outcomes | *Describe how you pre-defined primary and secondary outcome measures and how you assessed these measures.* |

# Dual use research of concern

Policy information about dual use research of concern

## Hazards

Could the accidental, deliberate or reckless misuse of agents or technologies generated in the work, or the application of information presented in the manuscript, pose a threat to:

No | Yes
☐ | ☐ Public health
☐ | ☐ National security
☐ | ☐ Crops and/or livestock
☐ | ☐ Ecosystems
☐ | ☐ Any other significant area

## Experiments of concern

Does the work involve any of these experiments of concern:

No | Yes
☐ | ☐ Demonstrate how to render a vaccine ineffective
☐ | ☐ Confer resistance to therapeutically useful antibiotics or antiviral agents
☐ | ☐ Enhance the virulence of a pathogen or render a nonpathogen virulent
☐ | ☐ Increase transmissibility of a pathogen
☐ | ☐ Alter the host range of a pathogen
☐ | ☐ Enable evasion of diagnostic/detection modalities
☐ | ☐ Enable the weaponization of a biological agent or toxin
☐ | ☐ Any other potentially harmful combination of experiments and agents

# ChIP-seq

## Data deposition

☐ Confirm that both raw and final processed data have been deposited in a public database such as GEO.

☐ Confirm that you have deposited or provided access to graph files (e.g. BED files) for the called peaks.

Data access links
*May remain private before publication.*
*For "Initial submission" or "Revised version" documents, provide reviewer access links. For your "Final submission" document, provide a link to the deposited data.*

Files in database submission
*Provide a list of all files available in the database submission.*

Genome browser session
(e.g. UCSC)
*Provide a link to an anonymized genome browser session for "Initial submission" and "Revised version" documents only, to enable peer review. Write "no longer applicable" for "Final submission" documents.*

## Methodology

Replicates
*Describe the experimental replicates, specifying number, type and replicate agreement.*

Sequencing depth
*Describe the sequencing depth for each experiment, providing the total number of reads, uniquely mapped reads, length of reads and whether they were paired- or single-end.*

Antibodies
*Describe the antibodies used for the ChIP-seq experiments; as applicable, provide supplier name, catalog number, clone name, and lot number.*

Peak calling parameters
*Specify the command line program and parameters used for read mapping and peak calling, including the ChIP, control and index files used.*

Data quality
*Describe the methods used to ensure data quality in full detail, including how many peaks are at FDR 5% and above 5-fold enrichment.*

Software
*Describe the software used to collect and analyze the ChIP-seq data. For custom code that has been deposited into a community repository, provide accession details.*

# Flow Cytometry

## Plots

Confirm that:

- [ ] The axis labels state the marker and fluorochrome used (e.g. CD4-FITC).
- [ ] The axis scales are clearly visible. Include numbers along axes only for bottom left plot of group (a 'group' is an analysis of identical markers).
- [ ] All plots are contour plots with outliers or pseudocolor plots.
- [ ] A numerical value for number of cells or percentage (with statistics) is provided.

## Methodology

| | |
|---|---|
| Sample preparation | *Describe the sample preparation, detailing the biological source of the cells and any tissue processing steps used.* |
| Instrument | *Identify the instrument used for data collection, specifying make and model number.* |
| Software | *Describe the software used to collect and analyze the flow cytometry data. For custom code that has been deposited into a community repository, provide accession details.* |
| Cell population abundance | *Describe the abundance of the relevant cell populations within post-sort fractions, providing details on the purity of the samples and how it was determined.* |
| Gating strategy | *Describe the gating strategy used for all relevant experiments, specifying the preliminary FSC/SSC gates of the starting cell population, indicating where boundaries between "positive" and "negative" staining cell populations are defined.* |

- [ ] Tick this box to confirm that a figure exemplifying the gating strategy is provided in the Supplementary Information.

# Magnetic resonance imaging

## Experimental design

| | |
|---|---|
| Design type | *Indicate task or resting state; event-related or block design.* |
| Design specifications | *Specify the number of blocks, trials or experimental units per session and/or subject, and specify the length of each trial or block (if trials are blocked) and interval between trials.* |
| Behavioral performance measures | *State number and/or type of variables recorded (e.g. correct button press, response time) and what statistics were used to establish that the subjects were performing the task as expected (e.g. mean, range, and/or standard deviation across subjects).* |

## Acquisition

| | |
|---|---|
| Imaging type(s) | *Specify: functional, structural, diffusion, perfusion.* |
| Field strength | *Specify in Tesla* |
| Sequence & imaging parameters | *Specify the pulse sequence type (gradient echo, spin echo, etc.), imaging type (EPI, spiral, etc.), field of view, matrix size, slice thickness, orientation and TE/TR/flip angle.* |
| Area of acquisition | *State whether a whole brain scan was used OR define the area of acquisition, describing how the region was determined.* |

Diffusion MRI  [ ] Used   [ ] Not used

## Preprocessing

| | |
|---|---|
| Preprocessing software | *Provide detail on software version and revision number and on specific parameters (model/functions, brain extraction, segmentation, smoothing kernel size, etc.).* |
| Normalization | *If data were normalized/standardized, describe the approach(es): specify linear or non-linear and define image types used for transformation OR indicate that data were not normalized and explain rationale for lack of normalization.* |
| Normalization template | *Describe the template used for normalization/transformation, specifying subject space or group standardized space (e.g. original Talairach, MNI305, ICBM152) OR indicate that the data were not normalized.* |
| Noise and artifact removal | *Describe your procedure(s) for artifact and structured noise removal, specifying motion parameters, tissue signals and physiological signals (heart rate, respiration).* |

| Volume censoring | *Define your software and/or method and criteria for volume censoring, and state the extent of such censoring.* |
|---|---|

## Statistical modeling & inference

| Model type and settings | *Specify type (mass univariate, multivariate, RSA, predictive, etc.) and describe essential details of the model at the first and second levels (e.g. fixed, random or mixed effects; drift or auto-correlation).* |
|---|---|
| Effect(s) tested | *Define precise effect in terms of the task or stimulus conditions instead of psychological concepts and indicate whether ANOVA or factorial designs were used.* |

Specify type of analysis:   ☐ Whole brain   ☐ ROI-based   ☐ Both

| Statistic type for inference (See Eklund et al. 2016) | *Specify voxel-wise or cluster-wise and report all relevant parameters for cluster-wise methods.* |
|---|---|
| Correction | *Describe the type of correction and how it is obtained for multiple comparisons (e.g. FWE, FDR, permutation or Monte Carlo).* |

## Models & analysis

| n/a | Involved in the study |
|---|---|
| ☐ | ☐ Functional and/or effective connectivity |
| ☐ | ☐ Graph analysis |
| ☐ | ☐ Multivariate modeling or predictive analysis |

| Functional and/or effective connectivity | *Report the measures of dependence used and the model details (e.g. Pearson correlation, partial correlation, mutual information).* |
|---|---|
| Graph analysis | *Report the dependent variable and connectivity measure, specifying weighted graph or binarized graph, subject- or group-level, and the global and/or node summaries used (e.g. clustering coefficient, efficiency, etc.).* |
| Multivariate modeling and predictive analysis | *Specify independent variables, features extraction and dimension reduction, model, training and evaluation metrics.* |

nature portfolio | reporting summary

March 2021

8