## [Peer Review File · Nature]

Manuscript Title: Post-extinction recovery of the Phanerozoic oceans and biodiversity hotspots

Reviewer Comments & Author Rebuttals

Reviewer Reports on the Initial Version:

Referees' comments:

Referee #1 (Remarks to the Author):

In principle, there are two approaches to understanding the history of biodiversity on Earth: 1) reconstructing that history based on global tabulations of first and last occurrences of fossil taxa and correlating it with various other records of change, and 2) forward modeling of biodiversity by parameterizing relevant boundary conditions and testing predictions with the fossil record. In this interesting study, the authors attempt the latter, using records of marine fossil biodiversity as benchmarks to compare and calibrate a forward model of biodiversity that is based principally on paleogeography, water mass properties, and persistence of marine environments. The empirical fossil record is also used to identify mass extinctions, perturbations that result in significant negative net diversification, which are also incorporated into the forward model. Two different diversification scenarios are considered: one in which there is unbridled exponential growth in all local geographic regions and one in which local carrying capacities determine local net diversification rates. In both cases, the model is seeded at 541 Ma with a uniform distribution of 1 genus assigned to all areas of sea floor. Diversity then accumulates on the deep sea floor according to its age distribution and water mass properties. On the shelf, where grid cells can oscillate between being emergent or submerged, new initial diversity values are seeded from the nearest neighbor flooded point. The diversification process for each grid remains active for as long as the area is underwater and is reset to 0 whenever it is emergent (shallow shelf) or removed entirely when it is destroyed (deep sea).

In order to arrive at a global diversity estimate it is necessary to convert local condition-determined predictions for diversity (i.e., alpha diversity) into estimates that incorporate spatial turnover of taxa (i.e., beta diversity), without actually having any notion of taxon identity in the model. The way this is accomplished is to identify diversity peaks, connect those peaks to diversity troughs (defined principally by spreading centers) along submerged transects, and then apply an exponentially decaying coefficient of similarity to arrive at a transect-integrated diversity estimate. The slightly more difficult to follow part is that an analogous process is then repeated for all transects ordinated by diversity, with an exponential decay in similarity applied to the distance between transect peaks (the closest peak to a given peak defining that distance and therefore the similarity coefficient).

The result of this exercise is a forward model of global diversity that incorporates hysteresis (in a sense) of the system imposed by evolving paleogeographic boundary conditions. Although considerable emphasis is placed on the resultant global pattern of biodiversity, an equal amount of attention is given to specific geographic regions that are predicted to have high biodiversity (hot spots) and the extent to which local environments are ever able to reach their carrying capacity in

the logistic model.

Given that this is a modeling study of a complex process playing out over the course of the entire Phanerozoic, there is plenty to take issue with. A few examples follow:

1) The method used to define the spatial turnover of biodiversity is defensible, but it largely ignores the fact that there is a distinct difference between longitudinal and latitudinal distance, with the latter often being a source of more significant taxonomic turnover. This is important because paleogeographic boundary conditions can conspire to make it more likely for shelf area to be arrayed latitudinally (i.e., like today's shallow shelf) vs. longitudinally (e.g., the shelf at 400 Ma), and this is presumably quite important to diversity and spatial turnover. It is not clear that the connectedness of the shallow shelf is being considered in the distance-based measure of similarity. Intervening stretches of land are, I believe, used as barriers but not deep ocean basins. Calibrating beta diversity and incorporating the effects of vicariance would seem to be a limitation of the model formulation.

2) The reconstruction of deep sea floor ages prior to the Mesozoic is problematic, to say the least, and this will presumably have some impact on the spatial turnover transects and, therefore, global diversity estimates.

3) The focus on global diversity rather than the underlying origination and extinction rates abstracts process to a very large extent. The approach taken here would seem to diminish the role of extinction in driving diversity patterns (this being implicitly a very origination-oriented model, with the exception of the "Big 5" superimposed, extinction rate is assumed to be a background constant). The decision to focus on origination is perhaps problematic because changes in diversity overall seem to be more tightly coupled to extinction than they are to origination. The end-Devonian "extinction" has a large effect in this model, but the extent to which that was driven by increased extinction rates, as opposed to a decline in origination, has been rightly questioned. The model outcome in the Paleozoic depends very heavily on this event (Fig. E10). It should be noted that everything is framed in terms of net diversification, the difference between origination and extinction, so this comment mainly pertains to the discussion of mechanism.

4) It was unclear to me where estimates of continental flooding were actually coming from in this model (though I could have just missed that). It was also unclear to what extent shorter-term oscillations in continental flooding have impacted the occupancy of shallow shelf environments. The end-Ordovician draw-down in sea level, for example, certainly exposed much of the shelf that here is presumably assumed to be constantly flooded through the Ordovician-Silurian. Presumably these types of higher-frequency oscillations in shelf flooding would serve to further displace local assemblages from equilibrium and so this is a conservative omission for at least some of the ideas considered. The effect of short-duration oscillations in continental flooding could matter particularly in the context of shifts between icehouse and greenhouse worlds. Did the late Paleozoic glaciation have a major effect on shelf persistence and therefore diversity in the Late Paleozoic? It seems not to have had a huge effect in the Pleistocene, though presumably this model would predict an effect if shelf area did indeed change markedly.

Despite a list of these and no doubt many other potential sticking points, this study is well executed and is, to my knowledge, the first such study to attempt a realistic spatially-explicit forward model of biodiversity in the Phanerozoic. It is, quite frankly, the general type of study that I have been hoping to do or see done for quite some time, so it is a very welcome read. The take home results are somewhat unsurprising when considered in the full context of the papers that are cited.

Paleogeographic boundary conditions have evolved asymmetrically during the Phanerozoic, as has been shown before, and continental flooding has varied in a semi-coordinated way with this shift; mass extinctions always seem to have big effects. Perhaps the most interesting insight provided by this modeling study is that the formation of diversity hotspots depends on the integrated history of paleogeographic context and continuity and that the evolving system has conspired to make the Late Cretaceous to Recent somewhat unusual in comparison to earlier times. Identifying an explicit candidate (besides fragmentation) for asymmetry during the Phanerozoic supercontinent cycle is an interesting finding that does help to improve our understanding of Phanerozoic biodiversity and the reason for its long-term trajectory.

Other questions and comments:

Spatial variation in diversity is predicted, and hotspots are highlighted in certain time intervals, but there is no real assessment of the prediction using the empirical fossil record or modern biodiversity. Is recent Mediterranean diversity really that high in comparison to other areas? Is this discrepancy the effect of connectivity crises in the Mediterranean or is something else going on? How well does the predicted spatial structure of diversity in the western Pacific equatorial region align with observations?

One question that comes to mind is whether or not a null geographic model might be useful. That is, what would be the effect of “turning off” plate tectonics and running the forward model on a static earth? This would be an interesting experiment to conduct in some end-member time intervals, maybe 400, 70, and 0. Presumably there is some amount of agreement with the canonical diversity histories that would emerge simply due to the shared mass extinctions operating on an environment template that is finite and structured. This experiment might give some insights into that and the specific role that changing geography has had.

 In Fig. 1 and the analogous supplemental figures K_{min} and K_{max} are selected to emphasize the difference with the exponential model, but these are generally poor fits overall. Is this tradeoff really worth it?

The relationship between temperature, food supply and rates of taxonomic origination is far from clear, though it is defensible in the sense that warm shallow seas (i.e., low latitude seas) have high diversity and appear to be both cradles and museums for biodiversity. Is there any room for discussion?

Supplementary Fig. 1 is cited on line 115, but it is not clear to me exactly what is supposed to be seen in that figure that is relevant to the comments in the text. What am I missing?

There is systematic offset between predicted diversity in the early Mesozoic and observed diversity, no matter the model used (Fig. 2). Is this the effect of differential extinction and a changed baseline origination-extinction balance, the Mesozoic marine revolution or some combination of these and

other factors? I don't expect this to be answered because it is probably unanswerable, but this model-data divergence is interesting and could be noted wrt the fundamentals of the model.

Fig. 4 m-o is truncated at the low end making the distribution appear to be monotonically increasing towards the low end. The full range is presented in Fig. E8, but I'm wondering if the full range shouldn't be shown here. It makes the good point that there is a modal value at around the current break point, which may not be obvious from the truncated results.

Referee #2 (Remarks to the Author):

The authors address the fundamental question of whether or not there are limits to diversity by introducing a regional model that describes diversification dynamics using both logistic and exponential growth over time, coupled with proxies of habitat fragmentation, oceanographic conditions and food availability. They found that diversity dynamics is best described by the exponential growth regime of a logistic function. As such, diversity has remained well below saturation (=upper limit) through time, with saturation being reached only in well-developed hotspots during the late Cretaceous and Cenozoic. Crucially, the authors show that the establishment of post-Paleozoic hotspots can be explained by the long timespan between mass extinctions during high continental fragmentation.

I found this paper fascinating. I think it presents an original and novel methodological approach (that accounts for multiple factors) to address a long standing question in biodiversity. The rationale, results and conclusions are robust and compelling. Importantly, this paper potentially represents a framework that can be applied to similar questions and other systems.

My evaluation of this manuscript is based on a broad familiarity with the subject. As such, it does not serve as an assessment of the intricacies of the methodological approach. Accordingly, I have the following general suggestions for improvement.

1. As stated in the manuscript, the question addressed here has remained open largely because of the biases of the fossil record. While the regional approach here used clearly addresses the resultant spatial heterogeneity of the fossil record, as presented, it remains unclear (or it is only implicit) how exactly is this framework tackling sampling and preservation biases.

2. The link between the establishment of hotspots and the longevity of shelf areas is unclear. On the one hand, the authors state that there is no evidence of an increase in the lifespan of continental margins, and therefore, no connection between the establishment of hotspots and the lifetime of shelf areas. But on the other, that these diversity hotspots were a consequence of the long residence time of shallow water seas within the tropical belt. I found these two statements contradictory and I think the manuscript would benefit of not only clarifying this, but of even developing more the role of habitat fragmentation and persistence of shallow-water habitats. These two factors are critical for the evolution and diversity of not only benthic communities but marine diversity in general.

3. Similarly, I think the authors ought to offer a more complete contrast and/or parallel between the diversity hotspots hypothesis and the Mesozoic marine revolution, especially considering that the latter is entirely driven by biotic interactions and the hypothesis presented here incorporates abiotic factors. This is relevant especially considering that biotic interactions seem to be linked to the diversity saturation/carrying capacity.

4. I consider it necessary to elaborate on the fact that extinction, especially during massive events is selective, but the hotspots hypothesis is presented based on a model that assumes non-random extinction. Of relevance to this issue, I found it interesting the correspondence between Sepkoski's diversity curve and the exponential model.

5. Figure 4 requires a legend on the colour gradient on the right.

Referee #3 (Remarks to the Author):

The manuscript by Cermeno et al details a study that uses a spatially explicit Earth system model to simulate the diversification of benthic marine animals across geological time. By comparing and fitting simulation results to data from the fossil record, the manuscript argues that much of the seafloor remains far below carrying capacity and that mass extinctions are required to account for the pattern of global diversity variation across geologic time.

The question of what factors shape and limit taxonomic diversity across geologic time is a major one that crosses the fields of biology, geology, and Earth system history. The primary strength of the study is that it attempts the most detailed simulation of taxonomic diversification within geological context that I am aware of. The primary weakness is the extent and quality of comparison to fossil data. I am not convinced by the manuscript that its conclusions could not have been reached by simpler methods. I provide a few major comments below, followed by more detailed comments referenced to line numbers (some of which repeat part or all of the major comments).

First, a major strength of the simulations used in the study is that they generate a spatially resolved Earth system model. However, the paper then presents little to no quantitative comparison between the predicted spatial distribution of biodiversity and data from the fossil record that could be used to test these predictions. The quantitative comparison between the model and data comes almost entirely from the record of global diversity, not from its spatial partitioning. Thus, the spatial component of the study serves more as a thought exercise than a test against data. This weakness can be corrected. The Paleobiology Database contains extensive occurrence records for benthic marine animals that could be used to test against the model predictions. In my view, spatially explicit model-data comparison is the key to making this manuscript more convincing.

Second, many of the conclusions of the study, particularly in the absence of spatial tests against the fossil record, could be anticipated prior to the analysis. In particular, the rapid, super-linear increase in taxonomic diversity during the Cretaceous-Recent in the Sepkoski database clearly requires the findings of the study. The other two datasets used for comparison also show rapid increases during the same time interval and diversity maxima at or near the Recent. There is no evidence that either

food supply, continental fragmentation, or temperature have increased exponentially over this time. Extended Data Figure 4 shows that continental configuration effects have varied but not trended overall across the Phanerozoic. Consequently, one must infer, even prior to seeing the modeling results, that most of the seafloor must have been far below carrying capacity for the entire Phanerozoic for diversity to be both at a maximum and increasing rapidly at the present day, after 550 million years of evolution. Stanley and others have already made this argument from global diversity compilations. The argument that this study offers new insight into the underlying dynamics depends on convincing readers that it is accurately representing the spatial dynamics of diversity increase, which would require detailed analysis of the fossil record.

Third, and related to the second point, the study goes to detail in modeling the Earth system but is more perfunctory in its treatment of the fossil data. The global diversity curves were obtained by digitizing published curves for datasets that are readily available. Moreover, because the manuscript simply digitizes these curves, it includes protists (Sepkoski) and non-benthic animals (all three datasets) in a study that is intended to model benthic ecosystems. The underlying data, from Sepkoski and the Paleobiology Database, are readily available online and could be filtered to benthic animals so that the comparisons are correct. Moreover, doing so would allow the authors to use newer data from the Paleobiology Database than were available to Alroy for subsampling. It would also allow them to provide the actual diversity data used in the study as an online appendix, rather than just the summary curves from prior studies. Code to conduct SQS subsampling is available and the manuscript could use the same SQS subsampling routine on more current data. Sepkoski's data are available through Shanan Peters' web portal.

Fourth, the potential role of continental distribution in shaping both global diversity and its spatial distribution has been discussed by James Valentine in a series of influential papers. None that work is cited in this manuscript. This manuscript follows so closely the same questions addressed by Valentine that it is essential not only to cite his previous work but also to discuss how the findings here are similar to or different from his. See detailed comments below for citations to some specific papers.

Line-by-line comments:

25-29: The paper acknowledges some prior fossil work on global diversity by Sepkoski and alternative arguments by Vermeij. However, work by James Valentine is by far the most relevant to the study here. Valentine discussed the combined biological and environmental controls on taxonomic diversification beginning in 1969 and wrote numerous papers addressing the role of plate tectonics in modulating diversity patterns (e.g., Valentine and Moores 1970), including via simulation (e.g., Valentine et al. 1978). Failing to cite and address these prior studies is a major oversight.

Valentine, J.W., 1969. Patterns of taxonomic and ecological structure of the shelf benthos during Phanerozoic time. *Palaeontology*, 12(4), pp.684-709.

Valentine, J.W. and Moores, E.M., 1970. Plate-tectonic regulation of faunal diversity and sea level: a model. *Nature*, 228(5272), pp.657-659.

Valentine, J.W., 1971. Plate tectonics and shallow marine diversity and endemism, an actualistic model. *Systematic Zoology*, 20(3), pp.253-264.

Valentine, J.W. and Moores, E.M., 1972. Global tectonics and the fossil record. *The Journal of Geology*, 80(2), pp.167-184.

Valentine, J.W., Foin, T.C. and Peart, D., 1978. A provincial model of Phanerozoic marine diversity. *Paleobiology*, 4(1), pp.55-66.

34-36: If the Marine Mesozoic Revolution model is not mutually exclusive, how would one determine relative contributions to these two mechanisms? It may be beyond the scope of this study to quantify relative contributions, but some discussion of how this might be done is important, particularly if the study is arguing (as it appears to be) that the spatial model without any particular ecological innovations is sufficient to explain most of the observed pattern.

41: A truly unconstrained, exponential model can be rejected a priori by issues of conservation of mass and constraints of space (see Kowalewski and Finnegan 2010). The only logically coherent argument for the exponential model is that the biosphere remains so far from carrying capacity that it can be safely ignored for purposes of numerical analysis.

Kowalewski, M. and Finnegan, S., 2010. Theoretical diversity of the marine biosphere. *Paleobiology*, 36(1), pp.1-15.

64: "variation" is a better word than "inequality" here

70-71: The study will become more convincing the more that the output is compared in a statistically rigorous way to spatially explicit fossil data. If the manuscript focuses primarily on the fit (or lack thereof) to global diversity curves, it does not advance very much on Sepkoski's kinetic model or Stanley's argument for exponential diversification.

74-78: There is rapidly accumulating evidence that oxygen availability was more limiting to diversification during Paleozoic time than temperature and, probably, food (e.g., Stockey et al., 2021). It is surely too much to include oxygen in this model, but the manuscript needs to be clear not only about what is modeled but also what is assumed to be constant. By not addressing oxygen, I am here assuming that the model treats atmospheric oxygen as a constant input to the cGENIE models. If that is not the case, it should be addressed clearly in the text. Also assumed is that the nature of ecological interactions does not change across geologic time. The manuscript acknowledges this issue with respect to Vermeij's work, but it should also address the work of Bush et al. (2016) on how the evolution of copulation as a reproductive mode could enable more species to co-exist at lower population densities, thus decoupling productivity from total diversity. The Bush argument is particularly relevant because, more than Vermeij, it provides a mechanism that would violate the assumptions of the model used in this study, which is that all taxa are interchangeable or can be modeled as such (following a similar approach to Hubbell's neutral theory). If there is a systematic evolutionary trend toward species existing at lower total population sizes, then assumptions about carrying capacity do not stay constant across time.

Stockey, R.G., Pohl, A., Ridgwell, A., Finnegan, S. and Sperling, E.A., 2021. Decreasing Phanerozoic extinction intensity as a consequence of Earth surface oxygenation and metazoan ecophysiology. *Proceedings of the National Academy of Sciences*, 118(41).

Bush, A.M., Hunt, G. and Bambach, R.K., 2016. Sex and the shifting biodiversity dynamics of marine animals in deep time. *Proceedings of the National Academy of Sciences*, 113(49), pp.14073-14078.

86: The models are intended to simulate the diversity history of benthic marine animals, but then uses datasets that include animals that live in the water column (e.g., most fish and cephalopods) and protists (e.g., foraminifera are a non-trivial component of the Sepkoski dataset). The Sepkoski genus dataset is available online and the Paleobiology Database can be downloaded, filtered, and subsampled to include only benthic marine animals. Given all the effort that went into this study, the study should go to the effort to compare the output to truly corresponding data. There is no excuse to keep the foraminifera and other protists in the Sepkoski dataset or the non-benthic animals in any of the datasets. I doubt that such filtering will have very large effects on the results and interpretation, but the arguments of the paper will be much more convincing if they are made against datasets that have been filtered appropriately.

101-107: This is a big opportunity for model-data comparison. Is there any evidence from the fossil record that the noted Paleozoic (Devonian and Permian) hotspots were unusually diverse, or at least any more diverse than other regions?

165-171: How surprising should this finding be to readers? Visual inspection of the Sepkoski data demonstrates an accelerating increase in diversity from the Cretaceous to the Cenozoic, at values much higher than in previous intervals. These simple facts would argue that any carrying capacity that is uniform across geologic time must be far above present diversity so that the behavior is close to exponential. In other words, is this finding something that could not be determined without this model used in the present study? I am not convinced by the manuscript that the answer here is "yes."

165-171: Perhaps what would help here is an analysis of how much the distribution of Keff changes across geologic time due to variation in temperature and food availability. And plots of how the distributions of food availability and temperatures with time would also help. This information would, I think, help readers to see how much variation in total (and local) diversity varies due to time since the past extinction (diversification dynamics) versus variation in the capacity of the environment to support diversity. My sense from the manuscript is that because most of the planet remains far below carrying capacity, global diversity is modulated mostly by the rates of diversity creation and destruction (frequency and magnitude of mass extinction) and very little by temperature, food availability. In other words, the study ends up arguing that the spatial resolution provided by this study is not needed to create a first-order understanding of Phanerozoic diversity dynamics. (This last argument may be wrong, but the manuscript should address why it is wrong more effectively for the benefit of the readers.) Perhaps the manuscript is also arguing that Sepkoski's "Paleozoic plateau" is not the reflection of carrying capacity for a subset of taxa, as he argues in his kinetic model, but instead simply the result of diversity being reduced by frequent mass

extinction events. This argument has been made previously by Stanley (2007). As noted earlier in this review, the place where this manuscript has more potential to tread novel ground is in the explicit spatial analysis of diversity trends.

Stanley, S.M., 2007. Memoir 4: an analysis of the history of marine animal diversity. *Paleobiology*, 33(S4), pp.1-55.

181-184: This correlation coefficient, like most other correlation coefficients, is designed for data where each x-y pair is independent of each other x-y pair. However, the data analyzed in this study are time series with significant autocorrelation, meaning that x-y pairs are not independent of one another (and are not expected to be so). Ignoring the autocorrelation of the time series will lead to overestimates of the causal connection implied by correlation. The manuscript should not take too much meaning from high values of the correlation coefficient without using methods that account for autocorrelation, such as first-differencing or generalized least squares regression.

189-192: See comment above about whether variations in temperature and food exert strong effects on diversity history in the model. This finding suggests further that they probably don't matter very much. It would be interesting to see what a model based on static modern geography predicts, only varying other parameters. Such an effort could be used to better quantify the relative importance of different factors in generating the Phanerozoic diversity trend in the model.

197: See comment above about autocorrelation and the issue of using correlation coefficients without correcting for autocorrelation. There is no guarantee that runs with higher correlation coefficients in the raw time series will also have higher coefficients in analyses that correct appropriately for autocorrelation, though they might.

231: "accounts for" should be "can account for". Earlier the manuscript acknowledges the possibility that other explanations are not mutually exclusive but here it appears to reject them without showing why they do not contribute to the observed pattern.

475-477: The model presented here uses origination rate to drive diversification and only explicitly addresses extinction in the context of mass extinction. However, we know from the fossil record that both taxonomic origination and taxonomic extinction rates have decline across time. What would happen to the model and its conclusions if extinction rates were assumed to decline across time in a way that at least moderately corresponds to observation? This feels to me like a missed opportunity to match the model to known constraints.

555-565: How good or bad is this approach likely to be? Epeiric seaways are difficult to model and may differ in their dynamics from nearby open oceans. I realize that there's not good way around this issue, but it would help readers to learn more about what the likely direction and magnitude of error might be from this approximation.

666-675: The actual datasets are available (both Sepkoski and Paleobiology Database). They don't need to be digitized from figures. And then they can also be filtered to include only benthic animals,

which is what this study attempts to model. The lack of effort to obtain and filter these data appropriately is puzzling to me given the massive effort that went into other aspects of the study.

Author Rebuttals to Initial Comments:**Referee #1 (Remarks to the Author):**

In principle, there are two approaches to understanding the history of biodiversity on Earth: 1) reconstructing that history based on global tabulations of first and last occurrences of fossil taxa and correlating it with various other records of change, and 2) forward modeling of biodiversity by parameterizing relevant boundary conditions and testing predictions with the fossil record. In this interesting study, the authors attempt the latter, using records of marine fossil biodiversity as benchmarks to compare and calibrate a forward model of biodiversity that is based principally on paleogeography, water mass properties, and persistence of marine environments. The empirical fossil record is also used to identify mass extinctions, perturbations that result in significant negative net diversification, which are also incorporated into the forward model. Two different diversification scenarios are considered: one in which there is unbridled exponential growth in all local geographic regions and one in which local carrying capacities determine local net diversification rates. In both cases, the model is seeded at 541 Ma with a uniform distribution of 1 genus assigned to all areas of sea floor. Diversity then accumulates on the deep sea floor according to its age distribution and water mass properties. On the shelf, where grid cells can oscillate between being emergent or submerged, new initial diversity values are seeded from the nearest neighbor flooded point. The diversification process for each grid remains active for as long as the area is underwater and is reset to 0 whenever it is emergent (shallow shelf) or removed entirely when it is destroyed (deep sea).

In order to arrive at a global diversity estimate it is necessary to convert local condition-determined predictions for diversity (i.e., alpha diversity) into estimates that incorporate spatial turnover of taxa (i.e., beta diversity), without actually having any notion of taxon identity in the model. The way this is accomplished is to identify diversity peaks, connect those peaks to diversity troughs (defined principally by spreading centers) along submerged transects, and then apply an exponentially decaying coefficient of similarity to arrive at a transect-integrated diversity estimate. The slightly more difficult part is that an analogous process is then repeated for all transects ordinated by diversity, with an exponential decay in similarity applied to the distance between transect peaks (the closest peak to a given peak defining that distance and therefore the similarity coefficient).

The result of this exercise is a forward model of global diversity that incorporates hysteresis (in a sense) of the system imposed by evolving paleogeographic boundary conditions. Although considerable emphasis is placed on the resultant global pattern of biodiversity, an equal amount of attention is given to specific geographic regions that are predicted to have high biodiversity (hot spots) and the extent to which local environments are ever able to reach their carrying capacity in the logistic model.

We are grateful for this astute summary of what we did, and how it fits into the wider picture of research on the history of diversity through geological time. We appreciate the time spent reviewing the manuscript.

Given that this is a modeling study of a complex process playing out over the course of the entire Phanerozoic, there is plenty to take issue with. A few examples follow:

1) The method used to define the spatial turnover of biodiversity is defensible, but it largely ignores the fact that there is a distinct difference between longitudinal and latitudinal distance, with the latter often being a source of more significant taxonomic turnover. This is important because paleogeographic boundary conditions can conspire to make it more likely for shelf area to be arrayed latitudinally (i.e., like today's shallow shelf) vs. longitudinally (e.g., the shelf at 400 Ma), and this is presumably quite important to diversity and spatial turnover. It is not clear that the connectedness of the shallow shelf is being considered in the distance-based measure of similarity. Intervening stretches of land are, I believe, used as barriers but not deep ocean basins. Calibrating beta diversity and incorporating the effects of vicariance would seem to be a limitation of the model formulation.

{R} We recognize that the use of a single distance-decay function for the spatial turnover of community composition is a limitation of our model. The reviewer states that this is particularly relevant when comparing estimates from time slices exhibiting differences in the proportion of latitudinally versus longitudinally oriented shelves. In order for us to consider this, it would be necessary to know not only the relative proportion of latitudinally versus longitudinally oriented shelves over time, but also the combined effect that i) the geographic arrangement of shelves and ii) their associated environmental history had on regional diversity, which makes analysis particularly cumbersome. We have re-run the model using three distance decay-functions for Early- and Mid-Paleozoic (541-350 Ma), Late Paleozoic (350-250 Ma) and Mesozoic/Cenozoic (250 Ma to present) (**Figure 1.1**) according to Miller et al. 2009 (figure 8). The analysis shows slight changes with respect to the original curve (using one single distance decay-function) (**Figure 1.2**) and the direction of the change does not alter the conclusions of this study.

- Miller, A. I. et al. Phanerozoic trends in the global geographic disparity of marine biotas. *Paleobiology* 35, 612-630 (2009).

Figure 1.1. Three different distance-decay functions (Variable Jaccard index) according to figure 8 of Miller et al. 2009 for three different time intervals in the Phanerozoic.

Figure 1.2. Effect of using three different distance-decay functions (Variable Jaccard), for the Paleozoic, Mesozoic and Cenozoic, respectively, on global diversity dynamics compared to the default configuration (Fixed Jaccard). The calibrated logistic model is used in these simulations.

➤ **ACTION:** For the sake of simplicity, we maintain our original analyzes in this latest version of the manuscript, but discuss the effect of using a time-varying Jaccard (see new text added below). Following the argument posed by the reviewer; that “*paleogeographic boundary conditions can conspire to make it more likely for shelf area to be arrayed latitudinally vs. longitudinally and that this effect would have influenced the temporal trends in global diversity*”, we would expect greater differences in taxonomic composition among distant regions today (i.e. open-ocean-facing coastal regions) than during the Paleozoic (wide continental shelf seas dominating throughout much of the Paleozoic). If so, this would lead the model to produce estimates of global diversity for the present-day greater than those produced by not assuming a time-varying Jaccard similarity index, further strengthening the idea that the increase in global diversity during the Mesozoic-Cenozoic is real.

The following text has been added to the revised version of the ms (**pages 30-31, lines 706-720**):

“The model considers a single distance-decay function for the spatial turnover of taxonomic composition. However, the degree of provinciality (i.e., the partitioning of life into distinct biogeographic units) varies in space and time as a result of environmental gradients (Miller et al. 2009) and plate tectonics (Valentine & Moores, 1970). In fact, the increase in provinciality has been invoked as the main driver of the increase in global diversity, especially in the Late Cretaceous and Cenozoic (Valentine et al. 1978; Zaffos et al. 2017; Kocsis et al. 2021). This is a deficiency of the model. Unfortunately, information on the extent to which marine provinciality has varied in space and time throughout the Phanerozoic is limited (Miller et al. 2009; Kocsis et al. 2021), and there is no simple (mechanistic) way to implement different distance-decay functions of taxonomic similarity in the model. We speculate that including some degree of provincialism in our model could produce the following. There is a clear difference between longitudinal and latitudinal distance, the latter being a more significant source of taxonomic turnover (Kocsis et al. 2021). This effect would add to the observation that tropical diversity hotspots became more prominent towards the end of the Phanerozoic, offering two complementary explanations for the increase in

diversity during the Cretaceous and Cenozoic: i) favourable conditions for the development of diversity hotspots and ii) a higher degree of provinciality.”

(newly added references are marked with asterisks)

- Kocsis, A.T., Redding, C.J., Scotese, C.R., Valdes, P.J., Kiessling, W.: Increase in marine provinciality over the last 250 million years governed more by climate change than plate tectonics, *Proceedings of the Royal Society B: Biological Sciences*, 288 (1957), doi.org/10.1098/rspb.2021.1342, (2021). **
- Miller, A. I. et al. Phanerozoic trends in the global geographic disparity of marine biotas. *Paleobiology* 35, 612–630 (2009).
- Valentine, J. W. & Moores, E. M. Plate-tectonic regulation of faunal diversity and sea level: A model. *Nature* 228, 657–659 (1970). **
- Valentine, J. W., Foin, T. C. & Peart, D. A provincial model of Phanerozoic marine diversity. *Paleobiology* 4, 55–66 (1978). **
- Zaffos, A., Finnegan, S. & Peters, S. E. Plate tectonic regulation of global marine animal diversity. *Proc. Natl. Acad. Sci. U. S. A.* 114, 5653–5658 (2017).

2) The reconstruction of deep sea floor ages prior to the Mesozoic is problematic, to say the least, and this will presumably have some impact on the spatial turnover transects and, therefore, global diversity estimates.

{R} We agree with the reviewer that reconstructing deep seafloor ages prior to the Mesozoic is problematic. This topic has recently been addressed by Williams and Müller, coauthors of this manuscript, who have developed a method for reconstructing maps of synthetic seafloor age based on global, full-plate topological reconstructions in which seafloor is generated, tracked and subducted. The uncertainties associated with the method are acknowledged in the manuscript. The method applied to the present study constitutes the current state of the art.

➤ **ACTION:** We emphasize this on **page 15, lines 361-367** of the revised manuscript:

“It is important to note that seafloor age maps for most of the Phanerozoic (i.e. pre-Pangea times) are not directly constrained by data due to recycling of oceanic crust at subduction zones. Rather, they are model predictions generated by constructing plate motions and plate boundary configurations from the geological and paleomagnetic record of the continents. Nonetheless, the first-order trends in ocean-basin volume and mean seafloor age are consistent with independent estimates for at least the last 410 Myr (Williams et al. 2021)”.

3) The focus on global diversity rather than the underlying origination and extinction rates abstracts process to a very large extent. The approach taken here would seem to diminish the role of extinction in driving diversity patterns (this being implicitly a very origination-oriented model, with the exception of the “Big 5” superimposed, extinction rate is assumed to be a background constant). The decision to focus on origination is perhaps problematic because changes in diversity overall seem to be more tightly coupled to extinction than they are to origination. The end-Devonian “extinction” has a large effect in this model, but the extent to which that was driven by increased extinction rates, as opposed to a decline in origination, has been rightly questioned. The model outcome in the Paleozoic depends very heavily on this event (Fig. E10). It should be noted that everything is framed in terms of net diversification, the difference between

origination and extinction, so this comment mainly pertains to the discussion of mechanism.

{R} As the reviewer correctly states, the model is framed in terms of net diversification rate and we do so to reduce complexity. The objectives of this study are i) to reconstruct the spatial distributions of diversity throughout the Phanerozoic, and ii) to check if (or to what extent) biological factors have limited the growth of diversity over time. Neither of these goals necessarily requires modeling the rates of origination and extinction.

➤ **ACTION:** We have now discarded the assumption that background extinction rates are assumed to be held constant in the model (this is indeed not necessarily true), thereby allowing for different mechanisms to explain the relationships between temperature, food supply and net diversification rates (new **Supplementary Fig. 5**).

To underpin the robustness of our approach, our analysis considers a broad range of values for the Q_{10} and K_{food} parameters which determine the shape of the Q_{temp} and Q_{food} functions or limitation terms. Q_{temp} and Q_{food} follow an exponential function and a Michaelis-Menten formulation, respectively, and define the temperature and the food dependence of the net diversification rate. Although we find differences in the absolute values of diversity depending on the choice of Q_{10} and K_{food} , overall, the spatial distribution patterns of diversity and the global diversity dynamics remain qualitatively the same for the range of values examined [$(Q_{10}: 1.5-2.5)$, $(K_{\text{food}}: 0.25-1)$] (Extended Data Table 1). In contrast, turning off Q_{temp} and Q_{food} makes the time for speciation the only control on regional diversity, leading to highly unrealistic biogeographic distributions, such as the occurrence of diversity hotspots in high latitudes (new **Supplementary Fig. 4g-1**).

4a) It was unclear to me where estimates of continental flooding were actually coming from in this model (though I could have just missed that).

{R} We apologize for this lack of clarity. We have clarified the method in the revised version of the manuscript.

➤ **ACTION:** In the main text, where the original version said (~line 69): “..., to a global plate motion model that constrains evolutionary time-within-regions”, we write now (**page 4, lines 70-73**):

“..., to a global model of palaeogeography and plate-motion that constrains evolutionary time-within-regions ...”

Additionally, we have modified the main text to more clearly denote where more information on the flooding maps can be found, and added more detail in the Supplementary text (Methods section). Now we write (**page 14, lines 337-341**):

“For continental regions, estimates of paleoelevation and continental flooding rely on a diverse range of geological evidence such as sedimentary depositional environments and the spatio-temporal distribution of volcanic activity. For a full description, see the recent review of Scotese 2021. Together, these data can be used to define the past locations of mountain ranges and paleoshorelines³⁵. For this part of our

reconstruction, we used the compilation of Scotese and Wright³⁴ with updated paleoshorelines based on depositional environment information in current fossil databases³¹. This compilation comprises 82 paleotopography maps covering the entire Phanerozoic. It is important to note that each paleogeographic map is a time-slice (Markwick and Valdes, 2004) representing the concatenation of geological data over several million years. Eustatic sea-level is thought to have varied by ~100 m at timescales much shorter than the duration of the time-slices throughout the Phanerozoic (e.g. Boulila et al, 2018), so that the extent of continental flooding could have varied within each time-slice by a significant amount for our analysis. For this reason, and to assess the uncertainty of our results to continental paleogeography in general, we computed additional maps of continental flooding for the analysis below in which the sea-level was raised or lowered by 100 m compared to the original paleoDEM grids of Scotese and Wright. The curves obtained differ very little from the original curves except for Zaffos et al in which the Cenozoic rise in diversity is more prominent (Extended Data Fig. 4d-f)."

(newly added references to the manuscript are marked with asterisks)

- Boulila, S., J. Laskar, B.U. Haq, B. Galbrun and N. Hara, (2018). Long-term cyclicities in Phanerozoic sea-level sedimentary record and their potential drivers. *Global and Planetary Change*, 165, p128-136 **
- Markwick, P.J. and P.J. Valdes (2004). Palaeo-digital elevation models for use as boundary conditions in coupled ocean-atmosphere GCM experiments: a Maastrichtian (late Cretaceous) example. *Palaeog., Palaeoclim. Palaeoecol.* 213, p37-63. **
- Scotese, C. (2021) An atlas of Phanerozoic paleogeographic maps: The seas come in and the seas go out. *Annual Review of Earth and Planetary Sciences* 49, p679-728. **

4b) It was also unclear to what extent shorter-term oscillations in continental flooding have impacted the occupancy of shallow shelf environments. The end-Ordovician draw-down in sea level, for example, certainly exposed much of the shelf that here is presumably assumed to be constantly flooded through the Ordovician-Silurian. Presumably these types of higher-frequency oscillations in shelf flooding would serve to further displace local assemblages from equilibrium and so this is a conservative omission for at least some of the ideas considered. The effect of short-duration oscillations in continental flooding could matter particularly in the context of shifts between icehouse and greenhouse worlds. Did the late Paleozoic glaciation have a major effect on shelf persistence and therefore diversity in the Late Paleozoic? It seems not to have had a huge effect in the Pleistocene, though presumably this model would predict an effect if shelf area did indeed change markedly.

{R} Changes in continental flooding through time are an integral part of our model and depend on the way ancient topographies and bathymetries are reconstructed (see the previous point on how continental paleoelevation and flooding were estimated).

Regarding mass extinctions and sea level changes, previous studies show that the latest Ordovician (Hirnantian) sea-level drop is relatively poorly constrained (could be up to 1x or 2x times the LGM) (Finnegan et al. 2011, Ghienne et al. 2014), and the same happens with the timing of the post-glacial sea-level rise (Pohl et al. 2016, Pohl & Austermann 2018). Global cooling and glacio-eustatic sea-level drop may have significantly contributed to the Late Ordovician Mass Extinction (phase 1) (Saupe et al. 2019).

Nonetheless, this impact is already (at least in part) accounted for when imposing negative net diversification rates to represent mass extinctions in the model. In fact, because the analysis of global fossil diversity curves is unable to discern the causes of diversity loss during mass extinctions, our imputation of negative diversification rates could have overestimated the loss of diversity in those cases in which sea level fall, a factor already accounted for by our model, contributed to mass extinctions.

- Finnegan, S., et al.: The magnitude and duration of late Ordovician-early Silurian glaciation, *Science*, 331(6019), 903–906, doi:10.1126/science.1200803, (2011). [*clumped isotopes and estimates of land-ice volume*]
- Ghienne, J.-F., et al.: A Cenozoic-style scenario for the end-Ordovician glaciation, *Nat. Comms.*, 5, 4485 (2014). [*reconstruction of land-ice advance-retreat*]
- Pohl, A., et al: Glacial onset predated Late Ordovician climate cooling, *Paleoceanography*, 31, 800–821 (2016). [*simulation of the Ordovician ice sheet*]
- Pohl, A. and Austermann, J.: A sea-level fingerprint of the Late Ordovician ice-sheet collapse, *Geology*, 46(7), 595–598 (2018). [*simulation of the post-glacial sea-level rise*]
- Saupe, E. et al.: Extinction intensity during Ordovician and Cenozoic glaciations explained by cooling and palaeogeography, *Nat. Geosci.*, 13, 65–70, doi:10.1038/s41561-019-0504-6, (2019). [*triggers for the mass extinction*]

The effect of short-term oscillations (e.g. glacial/interglacial cycles) is somewhat more difficult to represent in a model with a time resolution of ca 5-million years. We assume that populations track environmental conditions during marine regressions and transgressions, thus allowing habitat re-colonization. In the model, this process is simulated as follows (**page 24, lines 567-569**):

“the flooded continental points begin to accumulate diversity from the moment they are submerged, starting with a diversity value equal to the nearest neighbour flooded continental point with diversity > 1, thereby simulating a process of coastal re-colonization (or immigration)”.

Short-term oscillations may act to partially re-set accumulated diversity upon on each regression and adding a potential source of error (overestimate in accumulated diversity) as the reviewer recognizes. However, this is only the case for a given 5 Myr tectonic reconstruction reflecting the high sea-level stand of an interval of short-term oscillations; the error would effectively be zero if the tectonic reconstruction reflected the low sea-level stand. On a whole Phanerozoic (541 Ma) scale, we feel that any error is overall relatively minor. This source of error is now acknowledged in the manuscript (**page 10, lines 223-226**):

“The finite, ca. 5 Myr temporal resolution of our time-slices also precludes more rapid changes, such as cycles in sea-level and continental flooding, being explicitly accounted for, creating a potential temporal error in the loss or gain of shelf habitat and associated changes in biodiversity”.

Despite a list of these and no doubt many other potential sticking points, this study is well executed and is, to my knowledge, the first such study to attempt a realistic spatially-explicit forward model of biodiversity in the Phanerozoic. It is, quite frankly, the general type of study that I have been hoping to do or see done for quite some time, so it is a very welcome read. The take home results are somewhat unsurprising when considered in the full context of the papers that are cited. Paleogeographic boundary

conditions have evolved asymmetrically during the Phanerozoic, as has been shown before, and continental flooding has varied in a semi-coordinated way with this shift; mass extinctions always seem to have big effects. Perhaps the most interesting insight provided by this modeling study is that the formation of diversity hotspots depends on the integrated history of paleogeographic context and continuity and that the evolving system has conspired to make the Late Cretaceous to Recent somewhat unusual in comparison to earlier times. Identifying an explicit candidate (besides fragmentation) for asymmetry during the Phanerozoic supercontinent cycle is an interesting finding that does help to improve our understanding of Phanerozoic biodiversity and the reason for its long-term trajectory.

Many thanks for these positive comments; we are glad the referee is so positive about the originality of what we have done.

Other questions and comments:

Spatial variation in diversity is predicted, and hotspots are highlighted in certain time intervals, but there is no real assessment of the prediction using the empirical fossil record or modern biodiversity. Is recent Mediterranean diversity really that high in comparison to other areas? Is this discrepancy the effect of connectivity crises in the Mediterranean or is something else going on? How well does the predicted spatial structure of diversity in the western Pacific equatorial region align with observations?

{R} We recognize that the model cannot simulate the singularities of each of the oceanic regions, and even less so those of relatively small, enclosed seas for which the spatial resolution of the paleogeographic and Earth system models is insufficient to capture relevant features (paleobathymetry, seawater temperature, etc) in detail. Three lines of evidence support the model reconstructions:

- i. We include a new comparison between the regional diversity maps generated by the model and diversity estimates obtained from observations to directly address the reviewer's question, and present this in the latest version of the manuscript (new **Figure 2g-j**, pasted below). In particular, we use observations of two of the most diverse groups of marine invertebrates, crustaceans and molluscs, extracted from the Ocean Biodiversity Information System (OBIS) database. We find that the model reproduces reasonably well modern diversity distributions (i.e. diversity concentrates along shelf seas and decreases towards the deep sea and the high latitudes - this is largely a direct consequence of the outputs of the cGENIE Earth system model).
 - ii. The model reproduces reasonably well the movement of diversity hotspots from the western Tethys to the Indo-Pacific (e.g. **Figure 2e-f, Supplementary Video 3**). This pattern has been previously reported for benthic foraminifera (Fig. 1 in Renema et al. 2008) and coral reefs (Fig. 2 in Leprieur et al. 2016).
 - iii. The dynamics of global diversity produced by the diversification model correlate remarkably well with the dynamics of global diversity derived from the fossil record.
- Leprieur, F., et al.: Plate tectonics drive tropical reef biodiversity dynamics. Nat. Comms., 7, 11461 (2016)

- Renema, W., Bellwood, D.R., Braga, J.C., Bromfield, K.: Hopping Hotspots: Global Shifts in Marine Biodiversity. *Science*, 321(5889):654-7 (2008). DOI:10.1126/science.1155674.

➤ **ACTION:** We add this new comparison between model and observations to **Figure 2 panels g-j**. The figure is commented in the main text (**page 8, lines 169-182**, pasted below) and the methodology used is described in **pages 32-33, lines 754-775**.

“In order to evaluate the model's performance in reconstructing the spatial distributions of diversity, we compare the results of the calibrated logistic model for the recent (i.e., 0 Ma) with observations of two of the most diverse groups of marine invertebrates, crustaceans and molluscs, extracted from the Ocean Biodiversity Information System (OBIS), a global database of occurrence records of marine taxa (Methods). The regional diversity map generated by the model shows reasonable similarities to the observed diversity distributions along the continental margins (new **Fig. 2g-h**). The main discrepancies between the model and the OBIS data occur in the surroundings of Australia and New Zealand, where the model underestimates diversity. Although the model accounts for coastal re-colonization during marine transgressions, it lacks long-distance dispersal, which precludes a more detailed reconstruction of the spatial structuring of diversity in such a highly interconnected ocean region. Despite some regional discrepancies, both observed and modeled diversity decline from the equator towards the poles (new **Fig. 2i-j**), with most diversity concentrated in the Indo-West Pacific, the Atlantic Caribbean-East Pacific, and the Mediterranean (**Fig. 2g-h**).”

Figure 2: Re-diversifying the Phanerozoic oceans. a-f, Spatial distribution of marine animal diversity (# genera / area) in the Cambrian (Guzhangian, 500 Ma), Late Ordovician (Katian, 450 Ma), Early Devonian (Emsian, 400 Ma), Late Carboniferous (Pennsylvanian, 300 Ma), Late Cretaceous (Maastrichtian, 70 Ma) and present generated by the calibrated logistic model after

imposing the pattern of mass extinctions extracted from the fossil diversity curve of Sepkoski. This model run uses the following parameters: $Q_{10} = 1.75$, $K_{\text{food}} = 0.5 \text{ molC m}^{-2}\text{y}^{-1}$, net diversification rate limits ($\rho_{\text{min}} - \rho_{\text{max}}$) = $0.001\text{-}0.035 \text{ Myr}^{-1}$ (per capita), and a K_{min} to K_{max} range between 12 and 123 genera per unit area according to the calibration analysis presented in Extended Data Fig. 6. **g-h**, Current spatial distributions of diversity along the continental margins from model simulations and observations extracted from the OBIS database (genera belonging to Crustacea and Mollusca). For the purpose of comparison, normalized diversities (0-1) bounded between quantiles 0.05 and 0.95 are represented. **i-j**, Zonal mean diversity for 20 degree latitudinal bands

The reviewer also expresses concern about the great diversity of the Mediterranean hotspot compared to tropical hotspots. It is conceivable that the model overestimates the diversity of the Mediterranean due to the coarse spatial resolution of cGENIE and the difficulty of reconstructing the bathymetries of small enclosed seas. In the model, most of the Mediterranean is treated as a shallow-water shelf sea and, as a result, diversification is faster than it should be (at least in some areas). There is also the possibility, as suggested by the reviewer, that the Messinian salinity crisis could have influenced current levels of diversity, a phenomenon that the model cannot account for either.

➤ **ACTION:** We recognize in the manuscript this limitation and add the following sentence to Methods (**page 26, lines 597-605**):

“The model cannot simulate the singularities of relatively small, enclosed seas for which the spatial resolution of the paleogeographic and Earth system models is insufficient to capture relevant features (paleobathymetry, seawater temperature, etc) in detail. The method is also likely to underestimate the diversity of epeiric (inland) seas due to the difficulty of simulating immigration, a process that is strongly influenced by the effect of marine currents and is not considered here. However, the model considers recolonization of recently submerged areas from nearby coastal environments, which partially explains coastal immigration.”

One question that comes to mind is whether or not a null geographic model might be useful. That is, what would be the effect of “turning off” plate tectonics and running the forward model on a static earth? This would be an interesting experiment to conduct in some end-member time intervals, maybe 400, 70, and 0. Presumably there is some amount of agreement with the canonical diversity histories that would emerge simply due to the shared mass extinctions operating on an environment template that is finite and structured. This experiment might give some insights into that and the specific role that changing geography has had.

{R} This is a very interesting point, also raised by Reviewer 3, to test the impact of paleogeography against a static (NULL) geographic model in which plate tectonics is “turned OFF”. To do so, we have carried out new simulations for three static paleogeographic configurations: the Devonian (400 Ma), the Carboniferous (300 Ma) and the present. For each of these three configurations, the model runs for 541 million years in a ‘static mode’, that is, diversity accumulates steadily at a pace determined by the temperature and food assigned to each grid at the selected static configuration. Mass extinctions are imposed in the same way as in the default model with variable paleogeography. The test is performed for the exponential diversification model and the calibrated logistic model and for each of the three mass extinction patterns (aka Sepkoski, Alroy and Zaffos et al). New **Supplementary Figure 2** shows the difference

between the log-transformed normalized diversities produced by the model with static paleogeography (tectonics OFF) and the model with variable paleogeography (tectonics ON). Red and blue colors denote, respectively, the extent to which the static model produces diversity estimates above or below those produced by the model with plate tectonics. Tropical regions are dominated by reddish colors indicating that the static model overestimates diversity in these regions, where high temperatures accelerate diversification.

Supplementary Fig. 2. Testing the impact of plate tectonics on the spatial distributions of diversity. **a-i**, Comparison of the results of the exponential diversification model without plate tectonics and with plate tectonics. The color code represents the difference between the log-transformed normalized diversities (0-1) produced by the model with static palaeogeography (nDiv tectonics OFF) and the model with variable palaeogeography (nDiv tectonics ON) for three time frames (panels row-wise 400 Ma, 300 Ma and 0 Ma) and three extinction patterns (panels column-wise Sepkoski, Alroy and Zaffos et al.). **j-r**, As in panels a-i but for the ‘calibrated’ logistic model.

The absence of plate tectonics leads to a scenario of uncontrolled diversity growth that even mass extinctions cannot dampen (new **Supp. Fig. 3a-c**) and only effective carrying capacities prevent diversity from running away (**Supp. Fig. 3d-f**). These results support the idea that Earth's paleogeographic evolution and sea level changes, by creating, positioning and destroying seafloor habitats, have played a key role in constraining the growth of diversity throughout the Phanerozoic.

Supplementary Fig. 3. Testing the impact of plate tectonics on global diversity dynamics. **a-c**, Global diversity dynamics produced by the exponential diversification model with static palaeogeography (light blue, yellow and red for 400 Ma, 300 Ma and 0 Ma, respectively) and with variable palaeogeography (blue line) for each of the three mass extinction patterns (panels column-wise Sepkoski, Alroy and Zaffos et al). The corresponding fossil diversity curve is superimposed on each panel (grey dashed line). **d-f**, As in panels a-c but for the 'calibrated' logistic model.

➤ **ACTION:** We have added this analysis to the latest version of the manuscript as new Supplementary Figures 2-3 and new text in the main body of the manuscript (**page 10, lines 210-215**):

“...Use of a mechanistic model also enables interrogation of the likely causes of particular patterns in the fossil record that cannot be deduced from inspection of the fossil diversity curve alone (**Fig. 1**). For instance, we find that in the absence of progressive continental reconfiguration that allows continental shelf habitats to reposition along the latitudinal temperature gradient, diversity grows disproportionately in shelf seas lying (permanently) within the tropical belt (Methods, **Supplementary Figs. 2-3**).”

And in a new methods section entitled “Testing a static (null) palaeogeographic model” (**pages 33-34, lines 776-799**):

“In order to evaluate the effect of palaeogeography on global diversity dynamics, we carry out simulations for three static palaeogeographic configurations: the Devonian (400 Ma), the Carboniferous (300 Ma) and the present. For each of these three configurations, the model runs for 541 million years in a ‘static mode’, that is, diversity accumulates steadily at a pace determined by the temperature and food assigned to each grid at the selected static configuration. Mass extinctions are imposed the same way we do in the default model with variable palaeogeography. The test is performed for the exponential diversification model and the ‘calibrated’ logistic model and for each of the three mass extinction patterns (aka Sepkoski, Alroy and Zaffos et al). **Supplementary Figure 2** shows the differences between the log-transformed normalized diversities (between 0 and 1) produced by the diversification models with static palaeogeography (nDiv tectonics OFF) and with variable palaeogeography (nDiv tectonics ON). Red and blue colours denote, respectively, the extent to which the static model produces diversity estimates above or below those produced by the model with plate tectonics. Tropical regions are dominated by reddish colors indicating that the static model particularly overestimates diversity in these regions, where high temperatures accelerate diversification. In the absence of plate tectonics, the model leads to a scenario of uncontrolled diversity growth that even mass extinctions cannot dampen (**Supplementary Fig. 3a-c**) and only effective carrying capacities prevent diversity from running away (**Supplementary Fig. 3d-f**). These results support the idea that Earth’s palaeogeographic evolution and sea level changes, by creating, positioning and destroying seafloor habitats, have played a key role in constraining the growth of diversity throughout the Phanerozoic.”

In Fig. 1 and the analogous supplemental figures Kmin and Kmax are selected to emphasize the difference with the exponential model, but these are generally poor fits overall. Is this tradeoff really worth it?

{R} The reviewer is correct that these figures do not provide information relevant enough to be in the main section of the manuscript. The exponential model and the saturated logistic model are presented as two end-member models for the purpose of introducing the general concept to the readers but their results are now presented as Supplementary Videos 1-2.

➤ **ACTION:** In the new version of the manuscript, we remove the diversity maps produced by the exponential and the saturated logistic models, which are now presented as Supplementary Videos 1-2, and rearrange the set of figures presented in the main manuscript, starting with the global diversity curves (**Fig. 1**), followed by the calibrated logistic model (**new Fig. 2a-f**), its comparison with present-day observations (**new Fig. 2g-j**) and the diversity-to-carrying capacity ratio maps (**Fig. 3**). We feel that this is a more logical sequence of the most relevant figures of this study.

The relationship between temperature, food supply and rates of taxonomic origination is far from clear, though it is defensible in the sense that warm shallow seas (i.e., low latitude seas) have high diversity and appear to be both cradles and museums for biodiversity. Is there any room for discussion?

Indeed, there is much to say, but also many uncertainties and unknowns regarding the relationship between environmental conditions and rates of evolution.

➤ **ACTION:** The following text and references are now added to the Methods section (page 21, lines 488-498):

“The model considers a direct relationship between seawater temperature, food supply and the rate of net diversification based on the theoretical control that temperature and food supply exert on the rates of origination and extinction (Supplementary Fig. 5). Temperature rise is expected to accelerate the biochemical kinetics of metabolism (Allen et al. 2002) and shorten the development times of individuals (Gillooly et al. 2002), leading to higher rates of mutation and origination. Greater food availability increases population sizes, which increases the rates of mutation and reduces the probability of extinction (Pimm et al. 1988). Furthermore, a large body of observations shows the existence of a positive relationship between resource availability (i.e., food supply) and the standing stock of species in marine and terrestrial communities (Tilman, 1999, Costello & Chaudhary 2017). A larger food supply would support a greater number of individuals. A greater diversity of food resources could also lead to a finer partitioning of available resources (Tilman 1982).”

(newly added references are marked with asterisks)

- Allen, A. P., Brown, J. H. & Gillooly, J. F. Global biodiversity, biochemical kinetics, and the energetic-equivalence rule. *Science* 297, (2002). **
- Gillooly, J. F., Charnov, E. L., West, G. B., Savage, V. M. & Brown, J. H. Effects of size and temperature on developmental time. *Nature* 417, (2002). **
- Pimm, S. L., Jones, H. L. & Diamond, J. On the risks of extinction. *Am. Nat.* 132, (1988).
- Tilman, D. Resource competition and community structure. *Monographs in population biology* vol. 17 (Princeton University Press, 1982). **
- Tilman, D. The ecological consequences of changes in biodiversity: A search for general principles. in *Ecology* vol. 80 (1999). **
- Costello, M. J. & Chaudhary, C. Marine Biodiversity, Biogeography, Deep-Sea Gradients, and Conservation. *Current Biology* vol. 27 (2017). **

Supplementary Fig. 1 is cited on line 115, but it is not clear to me exactly what is supposed to be seen in that figure that is relevant to the comments in the text. What am I missing?

{R} Supplementary Figure 1a-d shows how long the continental margins have been underwater throughout the Phanerozoic. Sorry for the confusion, we meant to point to the seafloor age maps in this supplementary figure as support that there has not been a secular trend in the age of the seafloor or in the time that continental shelves have been underwater.

➤ **ACTION:** To be more specific, we add now: “(see seafloor age in Supplementary Fig. 1).”

There is systematic offset between predicted diversity in the early Mesozoic and observed diversity, no matter the model used (Fig. 2). Is this the effect of differential extinction and a changed baseline origination-extinction balance, the Mesozoic marine revolution or some combination of these and other factors? I don't expect this to be answered because it is probably unanswerable, but this model-data divergence is interesting and could be noted wrt the fundamentals of the model.

{R} This is an observation that has been widely discussed during the preparation of the paper without having been able to reach a convincing conclusion regarding the reasons for this discrepancy. We speculate on the possibility that it may have to do with the explosive radiation of specific taxonomic groups. We agree with the reviewer that the discrepancy itself is an interesting result worth exploring in the future, but it is certainly beyond the scope of this study.

➤ **ACTION:** We highlight this discrepancy along with the issue raised by Reviewer 2 regarding the assumption of non-selective extinction in the following new paragraph (**page 22, lines 513-524**):

“The model assumes non-selective extinction during mass extinction events (i.e., the field of bullets model of extinction (Raup, 1991); everything is equally likely to die, no matter the age of the clade and regardless of adaptation). However, there is much fossil evidence supporting extinction selectivity (Peters, 2008; Payne et al. 2016). It could be argued that higher extinction rates at diversity hotspots would have delayed their subsequent recovery, flattening global diversity trends. This is so considering that, in an exponential diversification model, diversity enhances diversification, leading to a non-linear relationship between the magnitude of diversity loss and the subsequent rebuilding time. This argument is difficult to reconcile with Sepkoski's genus-level global diversity curve but could be consistent with Alroy's standardized diversity curve. **Likewise, the model is also not suitable for reproducing the explosive radiations of certain taxonomic groups after mass extinctions, which could explain the offset between the model and fossil observations in the Early Mesozoic (Fig. 1).**”

(newly added references are marked with asterisks)

- D. M. Raup 1991. Extinction. Bad Genes or Bad Luck? xvii + 210 pp. New York, London: W. W. Norton **
- Peters, S.E. 2008. Environmental determinants of extinction selectivity in the fossil record. Nature 454(7204):626-9. doi: 10.1038/nature07032. **
- Payne, J.L., Bush, A.M., Chang, E.T., Heim, N.A., Knape, M.L., Pruss, S.B. 2016. Extinction intensity, selectivity and their combined macroevolutionary influence in the fossil record. Biology Letters 12(10):20160202. doi: 10.1098/rsbl.2016.0202. **

Fig. 4 m-o is truncated at the low end making the distribution appear to be monotonically increasing towards the low end. The full range is presented in Fig. E8, but I'm wondering if the full range shouldn't be shown here. It makes the good point that there is a modal value at around the current break point, which may not be obvious from the truncated results.

{R} Done. We include the full range plots (now Figure 3 m-o).

Referee #2 (Remarks to the Author):

The authors address the fundamental question of whether or not there are limits to diversity by introducing a regional model that describes diversification dynamics using both logistic and exponential growth over time, coupled with proxies of habitat fragmentation, oceanographic conditions and food availability. They found that diversity dynamics is best described by the exponential growth regime of a logistic function. As such, diversity has remained well below saturation (=upper limit) through time, with saturation being reached only in well-developed hotspots during the late Cretaceous and Cenozoic. Crucially, the authors show that the establishment of post-Paleozoic hotspots can be explained by the long timespan between mass extinctions during high continental fragmentation.

I found this paper fascinating. I think it presents an original and novel methodological approach (that accounts for multiple factors) to address a long standing question in biodiversity. The rationale, results and conclusions are robust and compelling. Importantly, this paper potentially represents a framework that can be applied to similar questions and other systems.

We are grateful for the positive comments of the Referee and thank them for the time spent reviewing the manuscript.

My evaluation of this manuscript is based on a broad familiarity with the subject. As such, it does not serve as an assessment of the intricacies of the methodological approach. Accordingly, I have the following general suggestions for improvement.

1. As stated in the manuscript, the question addressed here has remained opened largely because of the biases of the fossil record. While the regional approach here used clearly addresses the resultant spatial heterogeneity of the fossil record, as presented, it remains unclear (or it is only implicit) how exactly is this framework tackling sampling and preservation biases.

{R} Our spatially resolved diversification model is intended to provide a benchmark for the exploration of diversity, with particular emphasis on those regions and/or time intervals for which the fossil record is most biased, incomplete, or non-existent. Specifically - one could treat the model as 'reality', sub-sample its output according to assumptions regarding preservation and geographical bias, and hence explore to what extent the real sampled geological record reflects actual global diversity or not. Likewise, the comparison with the fossil record should provide a good test (validation) of the reliability of the model. Such analysis is beyond the scope of this study, but taken together it could help reconstruct and better understand the biogeography of marine animals since their rise to ecological prominence some 500 million years ago. We are excited with the possibility of conducting this research in collaboration with expert dataset paleobiologists.

➤ **ACTION:** In the revised version of the manuscript, we add the following text (**page 31, lines 721-733**):

“Regarding the comparison of model outputs with the fossil record, because the purpose of this modeling study is to reconstruct the unknown dynamics of diversity within regions, we adopted the strategy of spatially integrating regional diversity from our maps and comparing the resulting global diversity dynamics with the global diversity curves reconstructed from the fossil record. The comparison between the modeled and fossil global diversity curves is justified by the fact that the dynamics of global diversity should be quantitatively less biased than the dynamics of diversity within regions. Ultimately, our spatially resolved diversification model is intended to provide a benchmark for exploring diversity in those contrasting regions and/or time intervals for which the marine fossil record is most biased, incomplete, or non-existent. However, it would be interesting to compare the results of the model against the fossil record, at least for those regions and/or time intervals for which the fossil record is better preserved/sampled. This comparison would allow us to further test the reliability of the model and identify issues from which to improve the model.”

2. The link between the establishment of hotspots and the longevity of shelf areas is unclear. On the one hand, the authors state that there is no evidence of an increase in the lifespan of continental margins, and therefore, no connection between the establishment of hotspots and the lifetime of shelf areas. But on the other, that that these diversity hotspots were a consequence of the long residence time of shallow water seas within the tropical belt. I found these two statements contradictory and I think the manuscript would benefit of not only clarifying this, but of even developing more the role of habitat fragmentation and persistence of shallow-water habitats. These two factors are critical for the evolution and diversity of not only benthic communities but marine diversity in general.

{R} We apologize for the misunderstanding we try to clarify here. To explain the greatest prominence of Late Mesozoic and Cenozoic diversity hotspots, we first propose the possibility that it may be associated with a secular increase in the age of continental margins or the time that continental shelves have been underwater. This hypothesis is rejected because there is no empirical or modeling evidence to support an increase in the age of the seafloor towards the present. The alternative hypothesis is that there are differences in the time that marine faunas have had to diversify between consecutive mass extinctions. The results of the model prevent us from rejecting this hypothesis. Ultimately, it is the combined effect of time (the time elapsed between consecutive mass extinctions) and paleogeography (which controls how long a given continental shelf has remained within the tropical belt - promoting diversification) that control the magnitude of diversity hotspots.

➤ **ACTION:** We have removed the confusing statements from the latest version of the manuscript. The paragraph now is as follows (**pages 7-8, lines 158-168**):

“We argue that the temporal proximity between the Ordovician-Silurian (Hirnantian), Late Devonian (Frasnian-Famennian), and Permian-Triassic mass extinctions, coinciding with a long-lived phase of marine shelves destruction during the assembly of Pangaea, interrupted the full development of diversity hotspots during the Palaeozoic. By contrast, the comparatively long expanse of time that separated the mass extinctions of the end-Triassic and end-Cretaceous extended the time-for-speciation under conditions of increasing continental fragmentation, giving rise to

exceptionally high diversity regions before the Cretaceous-Paleogene mass extinction. The extraordinary diversity of Late Cretaceous hotspots ensured the continuity of relatively high diversity levels in the aftermath of the end-Cretaceous mass extinction, facilitating the subsequent development of diversity hotspots during the Cenozoic. "

3. Similarly, I think the authors ought to offer a more complete contrast and/or parallel between the diversity hotspots hypothesis and the Mesozoic marine revolution, especially considering that the latter is entirely driven by biotic interactions and the hypothesis presented here incorporates abiotic factors. This is relevant especially considering that biotic interactions seem to be linked to the diversity saturation/carrying capacity.

➤ **ACTION:** Following the recommendation of the referee, we have added a paragraph to the discussion section on the differences between the diversity hotspots hypothesis and the Mesozoic marine revolution (**page 11, lines 232-242**):

"It has been hypothesized that the Mesozoic marine revolution (Vermeij, 1977; Vermeij, 2021), that is, the emergence of shell-crushing predators and the consequent ecological restructuring of marine ecosystems, was primarily responsible for the increase in global diversity over the last 150 million years. The fact that our model can reproduce such an increase in diversity without the need to invoke evolutionary innovations like the emergence of new modes of predation (Vermeij, 1977; Vermeij, 2021), defence (Vermeij, 1977; Bush & Bambach 2011), mobility (Bush & Bambach 2011) or reproduction (Bush et al. 2016), among others, raises a new hypothesis based on how Earth's environmental history and palaeogeographic evolution interacted in concert to allow the development of diversity hotspots. We call this the 'diversity hotspots hypothesis', which is proposed as a non-mutually exclusive alternative to the hypothesis that evolutionary innovation and new ecospace occupation led this macroevolutionary trend (Vermeij, 1977)".

(newly added references are marked with asterisks)

- Bush, A. M. & Bambach, R. K. Paleocologic megatrends in marine metazoa. *Annu. Rev. Earth Planet. Sci.* 39, (2011). **
- Bush, A. M., Hunt, G. & Bambach, R. K. Sex and the shifting biodiversity dynamics of marine animals in deep time. *Proc. Natl. Acad. Sci. U. S. A.* 113, (2016). **
- Vermeij, G. J. The mesozoic marine revolution: Evidence from snails, predators and grazers. *Paleobiology* 3, 245–258 (1977).
- Vermeij, G. J. *Evolution and Escalation. Evolution and Escalation* (Princeton University Press, 2021). doi:10.2307/j.ctv18zhf8b. **

4. I consider it necessary to elaborate on the fact that extinction, especially during massive events is selective, but the hotspots hypothesis is presented based on a model that assumes non-random extinction. Of relevance to this issue, I found it interesting the correspondence between Sepkoski's diversity curve and the exponential model.

{R} There is no proper (mechanistic) way to incorporate extinction selectivity into this model, but we agree with the reviewer that this issue is worth elaborating in the manuscript for clarity.

➤ **ACTION:** In the revised version of the manuscript, we speculate on how extinction selectivity would influence the study's conclusions. We add the following text (**page 22, lines 513-524**):

“The model assumes non-selective extinction during mass extinction events (i.e., the field of bullets model of extinction (Raup, 1991); everything is equally likely to die, no matter the age of the clade and regardless of adaptation). However, there is much fossil evidence supporting extinction selectivity (Peters 2008; Payne et al. 2016). It could be argued that higher extinction rates at diversity hotspots would have delayed their subsequent recovery, flattening global diversity trends. This is so considering that, in an exponential diversification model, diversity enhances diversification, leading to a non-linear relationship between the magnitude of diversity loss and the subsequent rebuilding time. This argument is difficult to reconcile with Sepkoski's genus-level global diversity curve but could be consistent with Alroy's standardized diversity curve. Likewise, the model is also not suitable for reproducing the explosive radiations of certain taxonomic groups after mass extinctions, which could explain the offset between the model and fossil observations in the Early Mesozoic (**Fig. 1**).”

- D. M. Raup 1991. *Extinction. Bad Genes or Bad Luck?* New York, London: W. W. Norton **
- Peters, S. E. Environmental determinants of extinction selectivity in the fossil record. *Nature* 454, (2008). **
- Payne, J. L. et al. Extinction intensity, selectivity and their combined macroevolutionary influence in the fossil record. *Biol. Lett.* 12, (2016). **

5. Figure 4 requires a legend on the colour gradient on the right.

{R} Added. Thank you.

Referee #3 (Remarks to the Author):

The manuscript by Cermeno et al details a study that uses a spatially explicit Earth system model to simulate the diversification of benthic marine animals across geological time. By comparing and fitting simulation results to data from the fossil record, the manuscript argues that much of the seafloor remains far below carrying capacity and that mass extinctions are required to account for the pattern of global diversity variation across geologic time.

The question of what factors shape and limit taxonomic diversity across geologic time is a major one that crosses the fields of biology, geology, and Earth system history. The primary strength of the study is that it attempts the most detailed simulation of taxonomic diversification within geological context that I am aware of. The primary weakness is the extent and quality of comparison to fossil data. I am not convinced by the manuscript that its conclusions could not have been reached by simpler methods. I provide a few major comments below, followed by more detailed comments referenced to line numbers (some of which repeat part or all of the major comments).

We appreciate the insightful comments of the Referee and the time spent reviewing the manuscript. We are excited that the spatially resolved model presented here can help address unresolved questions about the macroevolutionary history of marine animals.

The reviewer states that we must provide firm evidence that our conclusions (eg reconstruction of global diversity dynamics) cannot be reached by simpler methods (eg without plate tectonics). We develop this point in a specific comment below and provide evidence that a static geographic model (i.e. one in which plate tectonics is turned off) leads to a scenario of runaway diversity growth that not even mass extinctions can dampen.

First, a major strength of the simulations used in the study is that they generate a spatially resolved Earth system model. However, the paper then presents little to no quantitative comparison between the predicted spatial distribution of biodiversity and data from the fossil record that could be used to test these predictions. The quantitative comparison between the model and data comes almost entirely from the record of global diversity, not from its spatial partitioning. Thus, the spatial component of the study serves more as a thought exercise than a test against data. This weakness can be corrected. The Paleobiology Database contains extensive occurrence records for benthic marine animals that could be used to test against the model predictions. In my view, spatially explicit model-data comparison is the key to making this manuscript more convincing.

{R} We agree that a robust comparison between the regional diversity patterns arising from the model and the fossil record would help to convince readers about the reliability of the model. However, the model was actually developed to overcome the spatial and temporal incompleteness of the fossil data. For instance, how can we convince ourselves/readers that the test is successful if the supposed gold standard (the fossil data) is itself biased? This analysis would require generating taxon accumulation curves (from alpha diversity estimates) within each region (grid cell) for each 1-million-year time interval to obtain estimates of regional diversity comparable with the results of the diversification

model. This would entail an effort far outside of the scope of the paper and the result would still not necessarily be free of uncertainties. Rather, one could treat the model as ‘reality’, sub-sample its output according to assumptions regarding preservation and geographical bias, and hence explore to what extent the real sampled geological record reflects actual global diversity or not – work that could be carried out in the future.

Because the purpose of this modeling study is to reconstruct the dynamics of diversity within regions, we adopted the strategy of spatially integrating regional diversity from our maps and comparing the dynamics of global diversity produced by the model with the global diversity curves reconstructed from the fossil record (reverse engineering). The comparison between the fossil and modeled global diversity curves is justified by the fact that the dynamics of global diversity should be less biased than the dynamics of diversity within regions.

However, we are able to make explicit and meaningful comparisons with observed modern diversity data extracted from the Ocean Biodiversity Information System (OBIS) database, and the model projection for present-day. These comparisons are presented in the new **Figure 2g-j**.

➤ **ACTION:** We agree with the reviewer that testing the spatial distributions arising from the model against the fossil record would be very valuable to support and improve the model and we are excited about the possibility of being able to do it in a future collaboration with expert paleobiologists. We have added the following in the manuscript (**page 31, lines 727-733**):

“Ultimately, our spatially resolved diversification model is intended to provide a benchmark for exploring diversity in those regions and/or time intervals for which the marine fossil record is most biased, incomplete, or non-existent. However, it would be interesting to compare the results of the model against the fossil record, at least for those regions and/or time intervals for which the fossil record is better preserved/sampled. This comparison would allow us to further test the reliability of the model and identify issues from which to improve the model.”

Second, many of the conclusions of the study, particularly in the absence of spatial tests against the fossil record, could be anticipated prior to the analysis. In particular, the rapid, super-linear increase in taxonomic diversity during the Cretaceous-Recent in the Sepkoski database clearly requires the findings of the study. The other two datasets used for comparison also show rapid increases during the same time interval and diversity maxima at or near the Recent. There is no evidence that either food supply, continental fragmentation, or temperature have increased exponentially over this time. Extended Data Figure 4 shows that continental configuration effects have varied but not trended overall across the Phanerozoic. Consequently, one must infer, even prior to seeing the modeling results, that most of the seafloor must have been far below carrying capacity for the entire Phanerozoic for diversity to be both at a maximum and increasing rapidly at the present day, after 550 million years of evolution. Stanley and others have already made this argument from global diversity compilations. The argument that this study offers new insight into the underlying dynamics depends on convincing readers that it is accurately representing the spatial dynamics of diversity increase, which would require detailed analysis of the fossil record.

{R} The fact that we replicate earlier results, such as the rapid rise in diversity in the past 100 Myr, is not surprising, but helps resolve a long-running debate about whether the rapid rise is largely real or not ('pull of the Recent'). The fact that this rise cannot be explained simply by some physical process such as increased food supply, changing temperature, or continental fragmentation is an observation, but not a weakness. We cannot test, or comment upon, the suggestion here (and earlier, by Stanley and others) that the ocean was operating below carrying capacity up to that point. We accept the final point that we cannot provide a final answer here, but we make the case that the Cenozoic global diversity rise is real, using a new approach; as for causes, these require perhaps a detailed study of individual major clades (i.e. those responsible for most of the biodiversity during this interval), and this is beyond the scope of the current paper. In this regard, and in others concerning bias and reality in the fossil record, our paper is likely to be cited as a null hypothesis against which such widespread data exploration can be assessed.

Third, and related to the second point, the study goes to detail in modeling the Earth system but is more perfunctory in its treatment of the fossil data. The global diversity curves were obtained by digitizing published curves for datasets that are readily available. Moreover, because the manuscript simply digitizes these curves, it includes protists (Sepkoski) and non-benthic animals (all three datasets) in a study that is intended to model benthic ecosystems. The underlying data, from Sepkoski and the Paleobiology Database, are readily available online and could be filtered to benthic animals so that the comparisons are correct. Moreover, doing so would allow the authors to use newer data from the Paleobiology Database than were available to Alroy for subsampling. It would also allow them to provide the actual diversity data used in the study as an online appendix, rather than just the summary curves from prior studies. Code to conduct SQS subsampling is available and the manuscript could use the same SQS subsampling routine on more current data. Sepkoski's data are available through Shanan Peters' web portal.

{R} We thank the reviewer for the advice.

➤ **ACTION:** In this new version of the manuscript, we use the Sepkoski dataset without protists (downloaded from <http://strata.geology.wisc.edu/jack/>). Therefore, all simulations have been re-run using the new pattern of mass extinctions (from the new global diversity curve) and the figures modified accordingly. As for the question of including only benthic organisms, our model can account for benthic and nektonic life as long as their diversification is governed by similar evolutionary rules (e.g. links between temperature, biochemical kinetics of metabolism, mutation rate) and there is a link with the physical environment especially on the continental shelves where most diversity accumulates. There is no reason to question that bivalves, cephalopods or other marine invertebrates obeyed similar evolutionary rules. Thus, we eliminate from the manuscript allusions to the modeling of only benthic animals. Note that we use the range of net diversification rates reported in Stanley (2007), which is based on the analysis of the fossil record of marine animals. We opt to keep the original curves from Alroy (2010) and Zaffos et al (2017) as this allows us to give a historical perspective to the paper.

- Stanley, S. M. An Analysis of the History of Marine Animal Diversity. *Paleobiology* 33, 1-55 (2007).

- Alroy, J. The shifting balance of diversity among major marine animal groups. *Science* 329, 1191–1194 (2010).
- Zaffos, A., Finnegan, S. & Peters, S. E. Plate tectonic regulation of global marine animal diversity. *Proc. Natl. Acad. Sci. U. S. A.* 114, 5653–5658 (2017).

Fourth, the potential role of continental distribution in shaping both global diversity and its spatial distribution has been discussed by James Valentine in a series of influential papers. None that work is cited in this manuscript. This manuscript follows so closely the same questions addressed by Valentine that it is essential not only to cite his previous work but also to discuss how the findings here are similar to or different from his. See detailed comments below for citations to some specific papers.

{R} Thank you for urging us to comment on these key documents. We apologize for the omission. Earlier work is now cited and commented on accordingly. Because we ought to limit citations in the manuscript, we add some of them to the following new texts related to the effect of plate tectonics and provinciality on diversity dynamics (**page 10, lines 208-210**):

“Our model corroborates earlier claims that Earth’s environmental history (Stanley 2007; Erwin 2009; Hannisdal & Peters 2011) and the patterns of continental fragmentation and reassembly (Valentine & Moores 1970; Valentine & Moores 1972; Zaffos et al. 2017) have been major determinants of marine animal diversification “.

In **pages 30-31, lines 706-720**, we add the following paragraph to put previous reference into the context of the present study:

“The model considers a single distance-decay function for the spatial turnover of taxonomic composition. However, the degree of provinciality (i.e., the partitioning of life into distinct biogeographic units) varies in space and time as a result of environmental gradients (Kocsis et al. 2021) and plate tectonics (Valentine et al. 1978). In fact, the increase in provinciality has been invoked as the main driver of the increase in global diversity, especially in the Late Cretaceous and Cenozoic (Valentine et al. 1978; Zaffos et al. 2017; Kocsis et al. 2021). This is a deficiency of the model. Unfortunately, information on the extent to which marine provinciality has varied in space and time throughout the Phanerozoic is limited (Miller et al. 2009; Kocsis et al. 2021), and there is no simple (mechanistic) way to implement different distance-decay functions of taxonomic similarity in the model. We speculate that including some degree of provincialism in our model could produce the following. There is a clear difference between longitudinal and latitudinal distance, the latter being a more significant source of taxonomic turnover (Kocsis et al. 2021). This effect would add to the observation that tropical diversity hotspots became more prominent towards the end of the Phanerozoic, offering two complementary explanations for the increase in diversity in the Mesozoic: i) more favourable conditions for the development of diversity hotspots and ii) a higher degree of provinciality.”

(newly added references are marked with asterisks)

- Erwin, D. H. Climate as a Driver of Evolutionary Change. *Current Biology* 19 (2009). **
- Hannisdal, B. & Peters, S. E. Phanerozoic earth system evolution and marine biodiversity. *Science* 334, 1121–1124 (2011). **

- Kocsis, Á. T., Reddin, C. J., Scotese, C. R., Valdes, P. J. & Kiessling, W. Increase in marine provinciality over the last 250 million years governed more by climate change than plate tectonics. *Proc. R. Soc. B Biol. Sci.* 288, (2021). **
- Miller, A. I. *et al.* Phanerozoic trends in the global geographic disparity of marine biotas. *Paleobiology* 35, 612–630 (2009).
- Stanley, S. M. An Analysis of the History of Marine Animal Diversity. *Paleobiology* 33, 1–55 (2007).
- Valentine, J. W. & Moores, E. M. Plate-tectonic regulation of faunal diversity and sea level: A model. *Nature* 228, 657–659 (1970). **
- Valentine, J.W. and Moores, E.M., 1972. Global tectonics and the fossil record. *The Journal of Geology*, 80(2), pp.167-184. **
- Valentine, J. W., Foin, T. C. & Peart, D. A provincial model of Phanerozoic marine diversity. *Paleobiology* 4, 55–66 (1978). **
- Zaffos, A., Finnegan, S. & Peters, S. E. Plate tectonic regulation of global marine animal diversity. *Proc. Natl. Acad. Sci. U. S. A.* 114, 5653–5658 (2017).

Line-by-line comments:

25-29: The paper acknowledges some prior fossil work on global diversity by Sepkoski and alternative arguments by Vermeij. However, work by James Valentine is by far the most relevant to the study here. Valentine discussed the combined biological and environmental controls on taxonomic diversification beginning in 1969 and wrote numerous papers addressing the role of plate tectonics in modulating diversity patterns (e.g., Valentine and Moores 1970), including via simulation (e.g., Valentine et al. 1978). Failing to cite and address these prior studies is a major oversight.

Valentine, J.W., 1969. Patterns of taxonomic and ecological structure of the shelf benthos during Phanerozoic time. *Palaeontology*, 12(4), pp.684-709.

Valentine, J.W. and Moores, E.M., 1970. Plate-tectonic regulation of faunal diversity and sea level: a model. *Nature*, 228(5272), pp.657-659.

Valentine, J.W., 1971. Plate tectonics and shallow marine diversity and endemism, an actualistic model. *Systematic Zoology*, 20(3), pp.253-264.

Valentine, J.W. and Moores, E.M., 1972. Global tectonics and the fossil record. *The Journal of Geology*, 80(2), pp.167-184.

Valentine, J.W., Foin, T.C. and Peart, D., 1978. A provincial model of Phanerozoic marine diversity. *Paleobiology*, 4(1), pp.55-66.

34-36: If the Marine Mesozoic Revolution model is not mutually exclusive, how would one determine relative contributions to these two mechanisms? It may be beyond the scope of this study to quantify relative contributions, but some discussion of how this might be done is important, particularly if the study is arguing (as it appears to be) that the spatial model without any particular ecological innovations is sufficient to explain most of the observed pattern.

{R} This is a very insightful comment, thank you! The model suggests that the overall increase in global diversity during the Late Mesozoic and Cenozoic was made possible by the full development of diversity hotspots in tropical shelf seas. In contrast, the lack of evolutionary innovation in the model prevents us from testing whether or not, or to what extent, the Mesozoic marine revolution contributed to the Mesozoic-Cenozoic global diversity trend. Nonetheless, we find that the calibrated logistic model underestimates global diversity during the last 100 million years with respect to the fossil diversity

curves of Sepkoski and Zaffos et al. Although speculative, it is conceivable that this discrepancy could be attributed to the simultaneous operation (sum of effects) of both mechanisms; (i) evolutionary innovation (by increasing effective carrying capacities) and (ii) the development of tropical diversity hotspots (by extending effective evolutionary time).

41: A truly unconstrained, exponential model can be rejected a priori by issues of conservation of mass and constraints of space (see Kowalewski and Finnegan 2010). The only logically coherent argument for the exponential model is that the biosphere remains so far from carrying capacity that it can be safely ignored for purposes of numerical analysis.

Kowalewski, M. and Finnegan, S., 2010. Theoretical diversity of the marine biosphere. *Paleobiology*, 36(1), pp.1-15.

{R} The reviewer is correct that an exponential diversification model is unsustainable indefinitely. However, it seems crucial to us to determine to what extent mass extinctions and paleogeography have (or have not) prevented regional biota from reaching diversity-dependent constraints. Because diversity varies dramatically between geographic regions, and each geographic region has its own geological and environmental history, addressing this question requires simultaneously reconstructing the dynamics of regional diversity both in space and time. We found that, in general, the dynamics of regional diversity follows a pattern of exponential diversification simply because regional communities have not had enough time to reach ecological saturation during the time between consecutive mass extinctions or between the creation and destruction of seafloor habitats. As far as we know, this has not been addressed before.

64: “variation” is a better word than “inequality” here

{R} Corrected - thanks!

70-71: The study will become more convincing the more that the output is compared in a statistically rigorous way to spatially explicit fossil data. If the manuscript focuses primarily on the fit (or lack thereof) to global diversity curves, it does not advance very much on Sepkoski’s kinetic model or Stanley’s argument for exponential diversification.

{R} We thank the reviewer for this reflection. As previously stated, our spatially-explicit diversity data is tested against (i) modern spatial diversity distributions (new **Figure 2g-j** showing the comparison between the model outputs for the present and observations extracted from the OBIS database) and (ii) fossil global diversity curves.

The study is novel in the sense that (i) it provides spatial distribution patterns resulting from integrating both the effect of paleogeography and environmental variability - the resulting patterns are intended to serve as a benchmark for paleobiologists, (ii) it allows us to calibrate the carrying capacities of the logistic model in order to represent the spatial patterns of ecological saturation in the Phanerozoic oceans and (iii) it provides an alternative explanation for the rapid increase in diversity in the past 100-150 Myr. Although these results have been previously discussed by Sepkoski, Stanley and others, here we present for the first time a spatially-resolved model that sheds light on previous

claims. The model has the potential to raise new hypotheses and test old ones (eg. the relationship between geographic configuration and latitudinal diversity gradient).

74-78: There is rapidly accumulating evidence that oxygen availability was more limiting to diversification during Paleozoic time than temperature and, probably, food (e.g., Stockey et al., 2021). It is surely too much to include oxygen in this model, but the manuscript needs to be clear not only about what is modeled but also what is assumed to be constant. By not addressing oxygen, I am here assuming that the model treats atmospheric oxygen as a constant input to the cGENIE models. If that is not the case, it should be addressed clearly in the text.

Stockey, R.G., Pohl, A., Ridgwell, A., Finnegan, S. and Sperling, E.A., 2021. Decreasing Phanerozoic extinction intensity as a consequence of Earth surface oxygenation and metazoan ecophysiology. *Proceedings of the National Academy of Sciences*, 118(41).

{R} Yes - this was an omission from the cGENIE model description. We do indeed assume a constant value of atmospheric pO_2 in the cGENIE experiments, which we set equal to modern-day. We make this choice in light of the considerable uncertainty in deeper-time pO_2 (Pohl, A. et al. in revision, <https://www.researchsquare.com/article/rs-915282/new/v1>, for instance, lines 157-185 in particular, with references therein) and wish to avoid entraining assumed changes in oxygenation that could imprint potentially unrealistic transitions onto the reconstructed biodiversity curves. In any case, ocean $[O_2]$ is not one of the environmental parameters that is presently included in the model (temperature and organic carbon rain to the seafloor) and we note that even without accounting for ocean $[O_2]$, the model as presented here already reproduces quite well early Paleozoic diversification trends. Arguably, $[O_2]$ could be inherent in the real-world meaning of the upper diversity limit, K_{max} , in our calibrated model although it is not at all obvious how one would relate ambient $[O_2]$ to biodiversity limits. In light of the reviewer's observation, we plan to explore the potential role of changing patterns in ocean $[O_2]$ with time in future work. (We also note as an aside that *Stockey et al.* did not consider the role of ambient $[O_2]$ in absolute biodiversity *per se*, but rather its role in determining the normalized susceptibility (relative magnitude) of extinction in response to warming.)

➤ **ACTION:** We have better clarified the assumptions made in the cGENIE model and the outputs employed in modeling Phanerozoic diversity trends. We have also added to the main text to point out the further possibility of trends in $[O_2]$ modulating the overall Phanerozoic trend in diversity (**Page 10, lines 219-223**):

“However, there are also limits to the potential factors that can be accounted for in our modelling. For instance, in this study, we do not account for variations through time of seawater oxygenation, because of both uncertainty in how oxygenation might set limits on maximum diversity as well as in the Phanerozoic history of atmospheric composition itself (Pohl et al. in revision).”

- Pohl, A., Ridgwell, A., et al. Climate as a Driver of Evolutionary Change. *Nature* (in revision) - <https://www.researchsquare.com/article/rs-915282/new/v1> **

Also assumed is that the nature of ecological interactions does not change across geologic time. The manuscript acknowledges this issue with respect to Vermeij's work, but it should also address the work of Bush et al. (2016) on how the evolution of copulation as a reproductive mode could enable more species to co-exist at lower population densities, thus decoupling productivity from total diversity. The Bush argument is particularly relevant because, more than Vermeij, it provides a mechanism that would violate the assumptions of the model used in this study, which is that all taxa are interchangeable or can be modeled as such (following a similar approach to Hubbell's neutral theory). If there is a systematic evolutionary trend toward species existing at lower total population sizes, then assumptions about carrying capacity do not stay constant across time.

Bush, A.M., Hunt, G. and Bambach, R.K., 2016. Sex and the shifting biodiversity dynamics of marine animals in deep time. *Proceedings of the National Academy of Sciences*, 113(49), pp.14073-14078.

{R} This is a very interesting point to take into consideration. The model does not consider changes in carrying capacity associated with evolutionary innovation and the occupation of previously unexploited habitats. As discussed in a previous comment, accounting for new modes of predation, reproductive strategies or other specific traits could account for the non-explained variability between model and observations. For example, the calibrated logistic model underestimates the increase in global diversity in the Late Mesozoic and Cenozoic when compared to the global diversity curves of Sepkoski and Zaffos et al. Together with Vermeij's Mesozoic marine evolution theory, in this new version of the manuscript, we consider Bush et al's argument about the effects of such a new mode of reproduction on expanding ecospace occupation.

➤ **ACTION:** We add now the following text and references to the discussion (**page 11, lines 235-242**):

"The fact that our model can reproduce such an increase in diversity without the need to invoke evolutionary innovations like the emergence of new modes of predation (Vermeij 1977), defence (Vermeij 1977, Bush & Bambach 2011), mobility (Bush & Bambach 2011) or reproduction (Bush et al. 2016), among others, raises a new hypothesis based on how Earth's environmental history and its paleogeographic evolution interacted in concert to allow the development of diversity hotspots."

(newly added references are marked with asterisks)

- Vermeij, G. J. The mesozoic marine revolution: Evidence from snails, predators and grazers. *Paleobiology* 3, 245–258 (1977).
- Bush, A. M. & Bambach, R. K. Paleoeologic megatrends in marine metazoa. *Annu. Rev. Earth Planet. Sci.* 39, (2011). **
- Bush, A. M., Hunt, G. & Bambach, R. K. Sex and the shifting biodiversity dynamics of marine animals in deep time. *Proc. Natl. Acad. Sci. U. S. A.* 113, (2016). **

86: The models are intended to simulate the diversity history of benthic marine animals, but then uses datasets that include animals that live in the water column (e.g., most fish and cephalopods) and protists (e.g., foraminifera are a non-trivial component of the Sepkoski dataset). The Sepkoski genus dataset is available online and the Paleobiology

Database can be downloaded, filtered, and subsampled to include only benthic marine animals. Given all the effort that went into this study, the study should go to the effort to compare the output to truly corresponding data. There is no excuse to keep the foraminifera and other protists in the Sepkoski dataset or the non-benthic animals in any of the datasets. I doubt that such filtering will have very large effects on the results and interpretation, but the arguments of the paper will be much more convincing if they are made against datasets that have been filtered appropriately.

{R} Please see our comment above. To summarize, we have re-run the model using the new pattern of mass extinctions for Sepkoski (2002) without protists (from the new global diversity curve) and modified the figures accordingly. As for the question of including only benthic organisms, there is no reason to question that bivalves, cephalopods or others obeyed similar evolutionary rules and thus can be accounted for by the model proposed here. We opt to keep the original curves from Alroy (2010) and Zaffos et al (2017) as this allows us to give a historical perspective.

101-107: This is a big opportunity for model-data comparison. Is there any evidence from the fossil record that the noted Paleozoic (Devonian and Permian) hotspots were unusually diverse, or at least any more diverse than other regions?

{R} As mentioned above, what the reviewer proposes is indeed a really exciting project! The model results open up a wealth of opportunities for database paleobiologists and we are excited about the prospect of establishing collaborations in the near future, as well as seeing our paper heavily cited as a starting point for such studies by others.

165-171: How surprising should this finding be to readers? Visual inspection of the Sepkoski data demonstrates an accelerating increase in diversity from the Cretaceous to the Cenozoic, at values much higher than in previous intervals. These simple facts would argue that any carrying capacity that is uniform across geologic time must be far above present diversity so that the behavior is close to exponential. In other words, is this finding something that could not be determined without this model used in the present study? I am not convinced by the manuscript that the answer here is "yes."

{R} The reviewer may be right that our results are not unexpected in some regards. However, all the points made by the reviewer have been hotly contested by database paleobiologists, and for 50 years in fact, since Raup (1972), and so our study seeks to resolve some of the debates, and in such a case, we agree with papers on one side, but not those on the other. Our analysis and result could also be argued to be timely because there have been many recent, high-profile papers in favor of the equilibrium view (e.g. by Alroy, Benson, Close, Rabosky), so the debate is far from resolved and indeed there is at present no generally agreed framework in which it can be resolved. Indeed, some of the recent papers by those four cited authors suggest that it perhaps never can be resolved because of massive bias and heterogeneity in fossil data. These reasons strengthen the purpose of the current contribution.

165-171: Perhaps what would help here is an analysis of how much the distribution of Keff changes across geologic time due to variation in temperature and food availability. And plots of how the distributions of food availability and temperatures with time would also help. This information would, I think, help readers to see how much variation in total

(and local) diversity varies due to time since the past extinction (diversification dynamics) versus variation in the capacity of the environment to support diversity. My sense from the manuscript is that because most of the planet remains far below carrying capacity, global diversity is modulated mostly by the rates of diversity creation and destruction (frequency and magnitude of mass extinction) and very little by temperature, food availability. In other words, the study ends up arguing that the spatial resolution provided by this study is not needed to create a first-order understanding of Phanerozoic diversity dynamics. (This last argument may be wrong, but the manuscript should address why it is wrong more effectively for the benefit of the readers.)

{R} According to our model, the development/interruption of diversity hotspots explains much of the global diversity dynamics. The development of diversity hotspots depends on two main factors: (i) the time-for-speciation between consecutive mass extinctions and (ii) the paleogeographic evolution of the Earth, which controls how long shallow shelves have resided within the tropical belt under conditions favorable for diversification. Thus, according to our model, the spatial structuring of diversity has exerted significant control over the dynamics of global diversity.

➤ **ACTION:** Even if the planet had indeed been operating below carrying capacity, the net diversification rate would still have varied as a result of changes in environmental conditions. In the model, we incorporate seawater temperature and food supply, but recognize that other factors such as $[O_2]$ would have come into play as well. In the revised version of the manuscript, we show the spatial distributions of diversity in the absence of temperature and food dependence of net diversification rate in the model (new **Supplementary Fig. 4**). We also provide the temporal evolution of mean K_{eff} along the continental shelves (**Extended Data Fig. 2g–h**). Regarding the temporal evolution of seawater temperature and food supply, we provide all this information in our github code which will be assigned a permanent DOI with Zenodo upon the acceptance of the manuscript.

Perhaps the manuscript is also arguing that Sepkoski's "Paleozoic plateau" is not the reflection of carrying capacity for a subset of taxa, as he argues in his kinetic model, but instead simply the result of diversity being reduced by frequent mass extinction events. This argument has been made previously by Stanley (2007). As noted earlier in this review, the place where this manuscript has more potential to tread novel ground is in the explicit spatial analysis of diversity trends.

Stanley, S.M., 2007. Memoir 4: an analysis of the history of marine animal diversity. *Paleobiology*, 33(S4), pp.1-55.

{R} We agree that mass extinctions contributed to dampening the growth of global diversity and, perhaps, to give the false appearance that global diversity reached dynamic equilibrium. Our model goes one step further by suggesting that the frequency of mass extinctions and plate tectonics (by creating, positioning and destroying habitats) led to the formation of well-developed diversity hotspots during the Late Cretaceous and Cenozoic. To the best of our knowledge, this is an original result that requires a spatially resolved model like the one presented here.

181-184: This correlation coefficient, like most other correlation coefficients, is designed

for data where each x-y pair is independent of each other x-y pair. However, the data analyzed in this study are time series with significant autocorrelation, meaning that x-y pairs are not independent of one another (and are not expected to be so). Ignoring the autocorrelation of the time series will lead to overestimates of the causal connection implied by correlation. The manuscript should not take too much meaning from high values of the correlation coefficient without using methods that account for autocorrelation, such as first-differencing or generalized least squares regression.

{R} Please note that a causal connection is not intended here, but a plain correlation between fossil data (observations) and model outputs. The use of the concordance correlation coefficient is useful for this purpose as it combines measures of both precision and accuracy (i.e., how close the model relative diversity is to the fossil relative diversity). Accuracy allows us to measure the displacement of the data (within specific time intervals) with respect to the 1:1 line.

189-192: See comment above about whether variations in temperature and food exert strong effects on diversity history in the model. This finding suggests further that they probably don't matter very much. It would be interesting to see what a model based on static modern geography predicts, only varying other parameters. Such an effort could be used to better quantify the relative importance of different factors in generating the Phanerozoic diversity trend in the model.

This same comment has also been raised by Reviewer 1. Here we duplicate the analysis performed and the response given to the other Reviewer.

{R} This is a very interesting point to test the impact of paleogeography against a static (NULL) geographic model in which plate tectonics is "turned OFF". To do so, we have carried out new simulations for three static paleogeographic configurations: the Devonian (400 Ma), the Carboniferous (300 Ma) and the present. For each of these three configurations, the model runs for 541 million years in a 'static mode', that is, diversity accumulates steadily at a pace determined by the temperature and food assigned to each grid at the selected static configuration. Mass extinctions are imposed in the same way as in the default model with variable paleogeography. The test is performed for the exponential diversification model and the calibrated logistic model and for each of the three mass extinction patterns (aka Sepkoski, Alroy and Zaffos et al). New **Supplementary Figure 2** shows the difference between the log-transformed normalized diversities produced by the model with static paleogeography (tectonics OFF) and the model with variable paleogeography (tectonics ON). Red and blue colors denote, respectively, the extent to which the static model produces diversity estimates above or below those produced by the model with plate tectonics. Tropical regions are dominated by reddish colors indicating that the static model overestimates diversity in these regions, where high temperatures accelerate diversification.

Supplementary Fig. 2. Testing the impact of plate tectonics on the spatial distributions of diversity. **a-i**, Comparison of the results of the exponential diversification model without plate tectonics and with plate tectonics. The color code represents the difference between the log-transformed normalized diversities (0-1) produced by the model with static palaeogeography (nDiv tectonics OFF) and the model with variable palaeogeography (nDiv tectonics ON) for three time frames (panels row-wise 400 Ma, 300 Ma and 0 Ma) and three extinction patterns (panels column-wise Sepkoski, Alroy and Zaffos et al). **j-r**, As in panels a-i but for the 'calibrated' logistic model.

The absence of plate tectonics leads to a scenario of uncontrolled diversity growth that even mass extinctions cannot dampen (new **Supplementary Fig. 3a-c**) and only effective carrying capacities prevent diversity from running away (new **Supplementary Fig. 3d-f**). These results support the idea that Earth's paleogeographic evolution and sea level changes, by creating, positioning and destroying seafloor habitats, have played a key role in constraining the growth of diversity throughout the Phanerozoic.

Supplementary Fig. 3. Testing the impact of plate tectonics on global diversity dynamics. **a-c**, Global diversity dynamics produced by the exponential diversification model with static palaeogeography (light blue, yellow and red for 400 Ma, 300 Ma and 0 Ma, respectively) and with variable palaeogeography (blue line) for each of the three mass extinction patterns (panels column-wise Sepkoski, Alroy and Zaffos et al). The corresponding fossil diversity curve is superimposed on each panel (grey dashed line). **d-f**, As in panels a-c but for the ‘calibrated’ logistic model.

➤ **ACTION:** We have added this analysis to the latest version of the manuscript as Supplementary Figures 2-3 and new text in the main body of the manuscript (**page 10, lines 210-215**):

“...Use of a mechanistic model also enables interrogation of the likely causes of particular patterns in the fossil record that cannot be deduced from inspection of the fossil diversity curve alone (**Fig. 1**). For instance, we find that in the absence of progressive continental reconfiguration that allows continental shelf habitats to reposition along the latitudinal temperature gradient, diversity grows disproportionately in shelf seas lying (permanently) within the tropical belt (Methods, **Supplementary Figs. 2-3**).”

And in a new Methods section entitled “Testing a static (null) palaeogeographic model” (**pages 33-34, lines 776-799**):

“In order to evaluate the effect of palaeogeography on global diversity dynamics, we carry out simulations for three static palaeogeographic configurations: the Devonian (400 Ma), the Carboniferous (300 Ma) and the present. For each of these three configurations, the model runs for 541 million years in a ‘static mode’, that is, diversity accumulates steadily at a pace determined by the temperature and food assigned to each grid at the selected static configuration. Mass extinctions are

imposed the same way we do in the default model with variable palaeogeography. The test is performed for the exponential diversification model and the 'calibrated' logistic model and for each of the three mass extinction patterns (aka Sepkoski, Alroy and Zaffos et al). **Supplementary Figure 2** shows the differences between the log-transformed normalized diversities (between 0 and 1) produced by the diversification models with static palaeogeography (nDiv tectonics OFF) and with variable palaeogeography (nDiv tectonics ON). Red and blue colours denote, respectively, the extent to which the static model produces diversity estimates above or below those produced by the model with plate tectonics. Tropical regions are dominated by reddish colors indicating that the static model particularly overestimates diversity in these regions, where high temperatures accelerate diversification. In the absence of plate tectonics, the model leads to a scenario of un-controlled diversity growth that even mass extinctions cannot dampen (**Supp. Fig. 3a-c**) and only effective carrying capacities prevent diversity from running away (**Supp. Fig. 3d-f**). These results support the idea that Earth's palaeogeographic evolution and sea level changes, by creating, positioning and destroying seafloor habitats, have played a key role in constraining the growth of diversity throughout the Phanerozoic."

197: See comment above about autocorrelation and the issue of using correlation coefficients without correcting for autocorrelation. There is no guarantee that runs with higher correlation coefficients in the raw time series will also have higher coefficients in analyses that correct appropriately for autocorrelation, though they might.

{R} Our goal is to compare two time series (observed and modelled), not to establish a causal connection between them. Choosing a CCC greater than 0.70 is a necessary procedure to establish a threshold below which the model results are discarded.

231: "accounts for" should be "can account for". Earlier the manuscript acknowledges the possibility that other explanations are not mutually exclusive but here it appears to reject them without showing why they do not contribute to the observed pattern.

➤ **ACTION:** This sentence has been removed from the manuscript. The first sentence of the discussion reads now as follows (**page 10, lines 208-210**):

"Our model corroborates earlier claims that Earth's environmental history and the patterns of continental fragmentation and reassembly have been major determinants of marine animal diversification."

A little further down in this discussion paragraph, we present other potential explanations for the increase in diversity in the Late Mesozoic and Cenozoic. See also our previous response on this.

475-477: The model presented here uses origination rate to drive diversification and only explicitly addresses extinction in the context of mass extinction. However, we know from the fossil record that both taxonomic origination and taxonomic extinction rates have decline across time. What would happen to the model and its conclusions if extinction rates were assumed to decline across time in a way that at least moderately corresponds to observation? This feels to me like a missed opportunity to match the model to known constraints.

{R} Sorry for this confusing sentence (now deleted) - the model is framed in terms of net diversification rate. Therefore, as it stands now, the model cannot assess for changes in origination and extinction rates over time. A long-term decline in origination and extinction rates, as shown by the fossil record, can lead to positive net diversification rates as long as the rate of origination exceeds the rate of background extinction over time.

555-565: How good or bad is this approach likely to be? Epeiric seaways are difficult to model and may differ in their dynamics from nearby open oceans. I realize that there's not good way around this issue, but it would help readers to learn more about what the likely direction and magnitude of error might be from this approximation.

{R} The method should work for epeiric seas as long as the spatial resolution of the paleogeographic and cGENIE models allow us to properly assign the prevailing environmental conditions within each geographic region (grid cell). Nonetheless, the model is likely to underestimate the diversity of the epeiric seas due to the difficulty of modeling immigration. In the current version of the model, it is stated (**page 24, lines 567-569**):

“the flooded continental points begin to accumulate diversity from the moment they are submerged, starting with a diversity value equal to the nearest neighbour flooded continental point with diversity >1 , thereby simulating a process of coastal re-colonization or immigration”.

The extent to which this method explains the colonization of inland seas and the time evolution of their diversity is uncertain and this is something that should be improved in the future.

➤ **ACTION:** We recognize this potential source of error in the latest version of the manuscript. We write (**page 26, lines 600-603**):

“The method is also likely to underestimate the diversity of epeiric (inland) seas due to the difficulty of simulating immigration, a process that is strongly influenced by the effect of marine currents and is not considered here.”

666-675: The actual datasets are available (both Sepkoski and Paleobiology Database). They don't need to be digitized from figures. And then they can also be filtered to include only benthic animals, which is what this study attempts to model. The lack of effort to obtain and filter these data appropriately is puzzling to me given the massive effort that went into other aspects of the study.

{R} As noted above, we have updated the Sepkoski curve using invertebrates and excluding protists. Regarding the other two curves, we have decided to keep them as they are in the original reports to give our manuscript a historical perspective. In addition, we give more information now in Methods about the groups of animals included in each case.

Reviewer Reports on the First Revision:

Referees' comments:

Referee #1 (Remarks to the Author):

This is my second review of Cermeño et al. I have read the long and detailed response to the previous round of reviews and I have read the revised manuscript. The authors have, in my opinion, satisfactorily addressed most of the substantive issues raised, which has resulted in an improved contribution. My assessment of the overall significance of the paper has not changed markedly from my first review (Reviewer 1) and I will not repeat those comments here. Nor will I provide another summary of the paper, at least as I understood it. Instead, I will point out only a few things from the authors' response and revised manuscript that caught my attention.

Abstract line 25/26: "...the extent to which biological interactions have constrained the growth of diversity over evolutionary time remains an open question, largely because of the incompleteness and spatial heterogeneity of the fossil record. ⁴ [SEP]

Is this true? Let's say for a minute that the incompleteness and spatial heterogeneity of the fossil record could be ignored and the pattern we were left with were "the one." Would this magically resolve the extent to which biological interactions have constrained diversity? As an aside, at least one of the cited papers in this sentence isn't pointing out the flaws in the fossil record, it is making the explicit case that the rock record and the signal in the fossil record are both responding to the same forcing mechanisms for environmental change (linked to tectonics, a recurring theme it seems).

Pg. 21 Response: "The comparison between the fossil and modeled global diversity curves is justified by the fact that the dynamics of global diversity should be less biased than the dynamics of diversity within regions."

This is a rather bold assertion. What are "global diversity dynamics" other than the sum of constituent regional components? More to the point, you don't have a "global fossil record" in any of the diversity results that are used here. Certainly Sepkoski's curve is not truly "global," though it does perhaps have some advantage in that the occurrences defining genus range endpoints were explicitly sought-out (as opposed to being arrived at through some broader sampling approach, like that in the PBDB). Instead, all of these curves present a very regionally-slanted representation of global diversity. To be clear, I am sympathetic to the argument the authors are making, but it is not at all apparent that it is true.

Pg. 21. Response: "However, we are able to make explicit and meaningful comparisons with observed modern diversity data extracted from the Ocean Biodiversity Information System (OBIS) database, and the model projection for present-day."

I am willing to acknowledge that the OBIS database is a somewhat different beast than the Paleobiology Database, but I am very reluctant to conclude that it is somehow a better "gold

standard" that is immune to the vagaries of heterogeneous and incomplete sampling. In the welcome new comparison between model predictions and OBIS, it might be acknowledged that this database, like all such resources, is a geographically heterogeneous sample of the biosphere. It is true that the fossil record has been subject to at least one more filter (geology), but beyond that, the two databases would seem to be much the same. Or am I missing something? Is OBIS really a uniform sample of the marine shelf environment that has no component of variance attributable to sampling effects? It seems to me that my previous comment, repeated in R3's comments, that the spatial information in the fossil record is being underutilized still holds. I do not think that this is a show-stopper, but it is a detraction.

Thank you for adding the static geography experiments. They are interesting and a nice addition to the results.

Shanan Peters (R1 from original submission)

Referee #2 (Remarks to the Author):

The authors introduce a novel regional model that reproduces the diversification trends of marine invertebrates after mass extinctions over the entire Phanerozoic, thereby addressing the fundamental question of whether or not there are limits to diversity.

In this second round of reviews, the authors have addressed all my concerns, which were outlined based on a broad understanding of the subject and which lacked of a deep understanding on the intricacies of the methodological approach. I am overall satisfied with their answers and actions taken to address my concerns. As a palaeo-biologist, I would very much like to see this work published.

Referee #3 (Remarks to the Author):

I have reviewed the revised manuscript from Cermeno et al. In general, I am satisfied that the authors have thoughtfully and sufficiently addressed most of the reviewer comments. I have not reviewed the methods in detail again due to time limitations and the understanding that these were not changed as part of the revision process.

A few comments remain, which I think should be addressed to further improve this interesting study.

30: "is" should be "are"

128-137: I remain quite convinced that a simple correlation coefficient such as Lin's is not appropriate for the analysis of time series data. The fact that it addresses both precision and accuracy is not relevant to the fact that it assumes independent data points rather than an autocorrelated time series as input. A better approach would be to use Lin's CCC on first differences, which would remove the autocorrelation of the time series, or to use a generalized least squares

regression. In the latter case, it would be possible to still compare the slope of the regression to the expected value of 1, as well as to assess the uncertainty on that value, thus addressing both precision and accuracy.

141-144: This logic really applies best to benthic animals. It is fine for pelagic animals in shallow environments, where primary production in the water column is tightly coupled to food supply to the seafloor. In open marine environments, there will be limited food supply to the deep-sea floor due to respiration of sinking organic matter whereas the animals living in the upper water column will have access to much more food. This is the one spot in the manuscript where the decision not to limit the analysis to benthic animals leads to logic that doesn't apply well. I understand that the authors chose to analyze data for all animals rather than to limit to benthic animals, but the text should therefore address the fact that in this particular case, the logic for interpreting how benthic and pelagic animals are affected diverges.

212-215: I am glad to see that the authors tried this modeling experiment. This is a very interesting result.

Author Rebuttals to First Revision:

Referee #1 (Remarks to the Author):

This is my second review of Cermeño et al. I have read the long and detailed response to the previous round of reviews and I have read the revised manuscript. The authors have, in my opinion, satisfactorily addressed most of the substantive issues raised, which has resulted in an improved contribution. My assessment of the overall significance of the paper has not changed markedly from my first review (Reviewer 1) and I will not repeat those comments here. Nor will I provide another summary of the paper, at least as I understood it. Instead, I will point out only a few things from the authors' response and revised manuscript that caught my attention.

Thank you for reviewing the manuscript again.

Abstract line 25/26: "...the extent to which biological interactions have constrained the growth of diversity over evolutionary time remains an open question, largely because of the incompleteness and spatial heterogeneity of the fossil record. ^{SEP}

Is this true? Let's say for a minute that the incompleteness and spatial heterogeneity of the fossil record could be ignored and the pattern we were left with were "the one." Would this magically resolve the extent to which biological interactions have constrained diversity? As an aside, at least one of the cited papers in this sentence isn't pointing out the flaws in the fossil record, it is making the explicit case that the rock record and the signal in the fossil record are both responding to the same forcing mechanisms for environmental change (linked to tectonics, a recurring theme it seems).

We meant to say that the incompleteness of the fossil record confounds our ability to disentangle biotic vs. abiotic factors. We agree with even given a perfect fossil record, there would still be much work ahead to fully understand it. For the sake of overall space, rather than expand the text to help clarify, we have removed the last part of the sentence and associated references in the new version of the manuscript.

Pg. 21 Response: "The comparison between the fossil and modeled global diversity curves is justified by the fact that the dynamics of global diversity should be less biased than the dynamics of diversity within regions."

This is a rather bold assertion. What are "global diversity dynamics" other than the sum of constituent regional components? More to the point, you don't have a "global fossil record" in any of the diversity results that are used here. Certainly Sepkoski's curve is not truly "global," though it does perhaps have some advantage in that the occurrences defining genus range endpoints were explicitly sought-out

(as opposed to being arrived at through some broader sampling approach, like that in the PBDB). Instead, all of these curves present a very regionally-slanted representation of global diversity. To be clear, I am sympathetic to the argument the authors are making, but it is not at all apparent that it is true.

The reviewer is correct that global diversity curves are far from global due to regional biases. This is indeed the reason that led us to pursue this modeling approach. We have removed this sentence in the latest version of the manuscript. An interesting possibility that could be tested next would be to deliberately undersample the model according to biases in the fossil record and compare model and fossil diversities.

Pg. 21. Response: "However, we are able to make explicit and meaningful comparisons with observed modern diversity data extracted from the Ocean Biodiversity Information System (OBIS) database, and the model projection for present-day."

I am willing to acknowledge that the OBIS database is a somewhat different beast than the Paleobiology Database, but I am very reluctant to conclude that it is somehow a better "gold standard" that is immune to the vagaries of heterogeneous and incomplete sampling. In the welcome new comparison between model predictions and OBIS, it might be acknowledged that this database, like all such resources, is a geographically heterogeneous sample of the biosphere. It is true that the fossil record has been subject to at least one more filter (geology), but beyond that, the two databases would seem to be much the same. Or am I missing something? Is OBIS really a uniform sample of the marine shelf environment that has no component of variance attributable to sampling effects? It seems to me that my previous comment, repeated in R3's comments, that the spatial information in the fossil record is being underutilized still holds. I do not think that this is a show-stopper, but it is a detraction.

The reviewer is correct, the OBIS data, like the fossil record data (PBDB), is spatially biased. While we have experience processing OBIS data (and ways to minimize spatial biases in sampling effort, see Methods section OBIS data, Page 34, lines 883-892), we lack similar skills dealing with fossil record data. We hope that our modeling approach will encourage database paleobiologists to test the model's diversity distributions, but also that the model will help fill the fossil gap in those time intervals and paleo-ocean regions in which the fossil record is non-existent or poorly preserved or sampled (a possible way to do this is raised in our response to the previous comment). A spatially resolved synthetic history of biodiversity through geologic time will help support the fossil record and complete the history of biodiversity. We acknowledge limitations in the OBIS database in the main text (Page 8, Lines 195-197):

“Furthermore, the OBIS data are not homogeneously distributed over the global ocean and although our analysis attempts to minimize this bias (Methods), some of the discrepancies between model and observations may be due to database limitations.”

Thank you for adding the static geography experiments. They are interesting and a nice addition to the results.

Thanks for suggesting we try this experiment.

Shanan Peters (R1 from original submission)

Referee #2 (Remarks to the Author):

The authors introduce a novel regional model that reproduces the diversification trends of marine invertebrates after mass extinctions over the entire Phanerozoic, thereby addressing the fundamental question of whether or not there are limits to diversity. In this second round of reviews, the authors have addressed all my concerns, which were outlined based on a broad understanding of the subject and which lacked of a deep understanding on the intricacies of the methodological approach. I am overall satisfied with their answers and actions taken to address my concerns. As a palaeo-biologist, I would very much like to see this work published.

Thank you for reviewing the manuscript again.

Referee #3 (Remarks to the Author):

I have reviewed the revised manuscript from Cermeno et al. In general, I am satisfied that the authors have thoughtfully and sufficiently addressed most of the reviewer comments. I have not reviewed the methods in detail again due to time limitations and the understanding that these were not changed as part of the revision process.

A few comments remain, which I think should be addressed to further improve this interesting study.

Thank you for reviewing the manuscript again.

30: "is" should be "are"

Done.

128-137: I remain quite convinced that a simple correlation coefficient such as Lin's is not appropriate for the analysis of time series data. The fact that it addresses both precision and accuracy is not relevant to the fact that it assumes independent data points rather than an autocorrelated time series as input. A better approach would be to use Lin's CCC on first differences, which would remove the autocorrelation of the time series, or to use a generalized least squares regression. In the latter case, it would be possible to still compare the slope of the regression to the expected value of 1, as well as to assess the uncertainty on that value, thus addressing both precision and accuracy.

The reviewer is correct that our time series are autocorrelated, which inflates the CCCs. It is true that the CCC or any other correlation coefficient applied on non-detrended data will be insensitive to high-frequency variations (i.e., short-term changes) and, therefore, will overestimate the fit when such high-frequency variability exists. However, it should be noted that our model is designed to describe main Phanerozoic trends in diversity and cannot reliably reproduce sub-stage (short-term) variability. This adds to the fact that the fossil data is biased, introducing an additional source of error in the analysis. Moreover, it should also be noted that in this calibration exercise we focus on comparing the general shape of the diversity curves produced by the model and those constructed from fossil data. Thus, our analysis searches for a model output that has an autocorrelation structure similar to that of the fossil time series.

Regardless, we have explored the application of generalized least squares (GLS) regression to the time series data (i.e., fossil data versus model estimates for each combination of K_{\min} and K_{\max}) and calculate Pearson's r and Lin's CCC. Then, we compare these coefficients with those calculated using OLS (i.e., without correcting for autocorrelation as done in the paper). We find that the coefficients resulting from the GLS method are lower than the standard CCC and Pearson's r , but both are highly positively correlated with each other (Figure below). We ought to stress that the combinations of K_{\min} and K_{\max} giving a CCC >0.7 in our fits (i.e. the threshold set to select the K_{\min} and K_{\max} parameters) are still the greatest in the GLS case (i.e. dots on the right side of the dashed line in the left panel).

We include the following paragraph in Methods (Page 29, Lines 758-765) to notice about this methodological question:

“The time series of global diversity generated from the fossil record and from the diversification model exhibit serial correlation and thus the resulting CCCs are inflated. The use of methods for the analysis of non-zero autocorrelation time series data, such as first differencing or generalised least squares regression, allows high-frequency variations along the time series to be taken into account. However, the relative simplicity of our model, designed to reproduce the main Phanerozoic trends in global diversity, coupled with the fact that biases in the fossil data would introduce uncertainty into the analysis, leads us to focus our analysis on the long-term trends, obviating the effect of autocorrelation.”

141-144: This logic really applies best to benthic animals. It is fine for pelagic animals in shallow environments, where primary production in the water column is tightly coupled to food supply to the seafloor. In open marine environments, there will be limited food supply to the deep-sea floor due to respiration of sinking organic matter whereas the animals living in the upper water column will have access to much more food. This is the one spot in the manuscript where the decision not to limit the analysis to benthic animals leads to logic that doesn't apply well. I understand that the authors chose to analyze data for all animals rather than to limit to benthic animals, but the text should therefore address the fact that in this particular case, the logic for interpreting how benthic and pelagic animals are affected diverges.

Thank you for pointing out this relevant issue that we are now commenting on in the new version of the manuscript. We now write (Page 33, lines 866-872):

“The choice to analyze the data for all animals is somewhat in conflict with the fact that, in the open ocean, photosynthetic primary production and the flux of organic matter at the bottom of

the water column are decoupled. This decoupling leads to contrasting differences in the amount of food available to the planktonic/nektonic and benthic communities, yet in the model, we assume that the amount of organic carbon reaching the seafloor is a proxy for food supply. Given that most of the diversity is concentrated in shallow shelf seas, this assumption is likely to be of only relatively minor importance in a global context.”

212-215: I am glad to see that the authors tried this modeling experiment. This is a very interesting result.

Thanks for suggesting we try this experiment.

Reviewer Reports on the Second Revision:

Referees' comments:

Referee #1 (Remarks to the Author):

This is my third assessment of Cermeño et al. I have read the authors' response to the previous round of reviews and the revised manuscript. I believe that they have addressed the questions and concerns that were raised and that the manuscript has been improved. I do not have any more specific comments beyond those that I have raised before in previous reviews (R #1, S. Peters). Notably, I think the treatment of beta diversity (spatial turnover) is defensible but likely to result in significant error that is difficult to assess. I do think the authors underutilize the spatial information in the empirical fossil record, but that is not a fatal flaw in their analysis, it is simply a shortcoming and obvious next step. It is also an open question to what extent the "target" global diversity curves used here are influenced by regional heterogeneity in sampling and the rock record, but that is also kind of the point of attempting to calibrate a forward model of diversity. There is a lot to like about this analysis and I think it is a good contribution. As usual, it is somewhat dissatisfying to have so much of the weight of the analysis and a fair portion of the content relegated to supplemental material, but I understand the motivations here. What is presented in the main text should be accessible to most readers from disparate backgrounds. Thank you for the opportunity to review this manuscript.

Referee #3 (Remarks to the Author):

I have read through this third submission of the manuscript by Cermeno et al. This version addresses the few comments that I had on the second version and I am satisfied with the caveats added to the text to address places where the methods used are not strictly appropriate but probably close enough for the purposes of this study. It also appears that the authors made a good-faith effort to address the remaining comments from Reviewer 1, which I felt were on target in raising some other key points that I had missed. I do not have any further comments and look forward to seeing the manuscript in print.

Reviewer 3 (Jonathan Payne)

Author Rebuttals to Second Revision:

Referee #1 (Remarks to the Author):

This is my third assessment of Cermeño et al. I have read the authors' response to the previous round of reviews and the revised manuscript. I believe that they have addressed the questions and concerns that were raised and that the manuscript has been improved. I do not have any more specific comments beyond those that I have raised before in previous reviews (R #1, S. Peters). Notably, I think the treatment of beta diversity (spatial turnover) is defensible but likely to result in significant error that is difficult to assess. I do think the authors underutilize the spatial information in the empirical fossil record, but that is not a fatal flaw in their analysis, it is simply a shortcoming and obvious next step. It is also an open question to what extent the "target" global diversity curves used here are influenced by regional heterogeneity in sampling and the rock record, but that is also kind of the point of attempting to calibrate a forward model of diversity. There is a lot to like about this analysis and I think it is a good contribution. As usual, it is somewhat dissatisfying to have so much of the weight of the analysis and a fair portion of the content relegated to supplemental material, but I understand the motivations here. What is presented in the main text should be accessible to most readers from disparate backgrounds. Thank you for the opportunity to review this manuscript.

Thank you again for your very insightful and constructive comments, which have greatly improved the manuscript. We fully agree with the reviewer that this modeling study includes a number of assumptions whose impact on the final results is difficult to assess and this is something that needs to be developed and addressed in future versions of the model. In the same way, one of the next steps should be the comparison between the results of the regional diversification model (geographic patterns of diversity) and the fossil record. We hope that the model presented here will serve to build a synthetic history of biodiversity through geological time that, in combination with the fossil record, will allow us to gain a better understanding of the evolution of marine life.

Referee #3 (Remarks to the Author):

I have read through this third submission of the manuscript by Cermeno et al. This version addresses the few comments that I had on the second version and I am satisfied with the caveats added to the text to address places where the methods used are not strictly appropriate but probably close enough for the purposes of this study. It also appears that the authors made a good-faith effort to address the remaining comments from Reviewer 1, which I felt were on target in raising some other key points that I had missed. I do not have any further comments and look forward to seeing the manuscript in print.

Reviewer 3 (Jonathan Payne)

Thank you again for your very insightful and constructive comments, which have greatly improved the manuscript.